



# The role and value of distributed precipitation data for hydrological models

Ralf Loritz[1], Markus Hrachowitz[2], Malte Neuper[1] and Erwin Zehe[1]

[1] Karlsruhe Institute of Technology (KIT), Institute of Water and River Basin Management, Karlsruhe, Germany
[2] Delft University of Technology, Faculty of Civil Engineering and Geosciences, Delft, Netherlands

*Correspondence to*: Ralf Loritz (Ralf.Loritz@kit.edu)

**Abstract**

This study investigates the role and value of distributed rainfall for the runoff generation of a mesoscale catchment (20 km$^2$). We compare the performance of three hydrological models at different periods and show that a
distributed model driven by distributed rainfall yields only to improved performances during certain periods. These periods are dominated by convective storms that are typically characterized by higher spatial and temporal variabilities compared to stratiform precipitation events that dominate the rainfall generation in winter. Motivated by these findings we develop a spatially adaptive model that is capable to dynamically adjust its spatial structure during runtime to represent the varying importance of distributed rainfall within a hydrological model without
losing predictive performance compared to a spatially distributed model. Our results highlight that adaptive modeling might be a promising way to better understand the varying relevance of distributed rainfall in hydrological models as well as reiterate that it might be one way to reduce computational times. They furthermore show that hydrological similarity concerning the runoff generation does not necessarily mean similarity for other dynamic variables such as the distribution of soil moisture.





## 1 Introduction

"*How important are spatial patterns of precipitation for the runoff generation at the catchment scale?*" – This is a key question for the application of hydrological models that has been addressed in several studies over the last three decades (e.g. *Beven and Hornberger, 1982; Smith et al., 2004; Lobligeois et al., 2014*). A frequently drawn
conclusion is that semi-distributed or even lumped models driven by a single precipitation time series often outperform distributed models with respect to their ability to reproduce observed streamflow at the outlet of a catchment (e.g. *Das et al., 2008*). Although such findings are surely constrained by the fact that distributed models have more parameters that need to be identified, which makes model calibration much more challenging (*Beven and Binley, 1992; Huang et al., 2019*), they highlight the ability of the hydrological system to dissipate spatial
gradients efficiently (e.g. *Obled et al., 1994; Berkowitz and Zehe, 2020*)

In contrast to the above-mentioned finding that hydrological systems can efficiently dissipate spatial gradients, several other studies showed that information about the spatial variability of precipitation can significantly improve the predictive performance of hydrological models. For instance, *Euser et al. (2015)* highlighted that distributed
models driven by distributed rainfall could reproduce the observed hydrograph of a 1600 km$^2$ large catchment in Belgium with higher accuracy compared to spatially lumped model structures. Furthermore, *Woods and Sivapalan (1999)* showed that the interplay between spatial patterns of rainfall and soil saturation can substantially impact the runoff generation of a catchment when they analyzed average runoff rates in dependence of the spatial and temporal variability of the meteorological forcing and the catchment state. The relevance of these spatial patterns
is thereby particularly high if the system is close to a threshold where different localized preferential flow processes start dominating (e.g. cracking soils: drying of soil; macropores: occurrence of earthworms) as discussed by *Zehe et al. (2007)*. Spatial averaging of the system state or the meteorological forcing can hence lead to a misrepresentation of relevant spatial patterns, especially at more extreme conditions.

Given the partly contradictory findings present in the literature, it appears reasonable to assume that the relevance of distributed rainfall is changing dynamically over time and depends on the interplay of the prevailing i) system state (e.g. catchment wetness), ii) on the system functional structure, determined by patterns of topography, land-use, and geology, as well as on iii) the strength and spatial organization of the rainfall forcing. In consequence, it seems furthermore rational to hypothesize that also hydrological models should dynamically adapt their spatial
structure to the prevailing context thereby reflecting the inherently dynamic nature of hydrological similarity (*Loritz et al., 2018*).



The idea that hydrological models should dynamically allocate their spatial resolution, as well as the associated representation of natural heterogeneity in time, is motivated by our previous work (*Loritz et al. 2018*). In this study, we highlighted that simulations of a distributed model consisting of 105 independent hillslopes were highly

redundant to reproduce discharge or catchment storage changes of a mesoscale catchment within one hydrological year. Based on the Shannon entropy we identified periods where a rather small number of representative hillslopes was sufficient because most of them functioned largely similar within the chosen margin of error. However, during other periods up to 32 independent representations of hillslopes were required, which underlines that spatial variability of system properties, such as surface topography or soil types among the hillslopes can exert a stronger

influence on the runoff generation at certain times as expected given the findings reported by other studies conducted in the same research environment (e.g. *Fenicia et al., 2016; Loritz et al., 2017*). It can, therefore, be argued that also distributed rainfall and corresponding distributed model structures are only of higher relevance during specific periods, while during other periods a compressed, spatially aggregated model structure may be sufficient. An implementation of such an adaptive spatial model resolution would ensure an appropriate spatial

model complexity, defined based on the least amount of details about the system structure (e.g. the variability of topographic gradients) and catchment states that are sufficient to capture the relevant interactions with the spatial pattern of precipitation. Yet it would be as parsimonious as possible to avoid redundant computations, which again could be used to minimize computational costs (*Clark et al., 2017*).

Moving to the event time scale instead of running continuous simulations is surely one-way to achieve such a dynamical allocation of the model space. This would entail running a set of models that differ with respect to their resolutions in space and time depending on the prevailing structure of the forcing and wetness state of a landscape. Yet, this introduces multiple new problems, for instance, how to infer the initial conditions of a catchment prior to a rainfall event given the degrees of freedom distributed models can offer (*Beven, 2001*). The latter is of

considerable importance particularly during extremes resulting from high-intensity rainfall-runoff events, which can be strongly sensitive to the actual state of the system such as the spatial patterns of macropores (*Zehe et al., 2005*) or of the antecedent soil water content (*Zehe and Blöschl, 2004*).

A different avenue to implement a dynamically changing model resolution is adaptive clustering, as recently

demonstrated for a spatially distributed conceptual (top-down) model by *Ehret et al. (2020)*. This concept allows for continuous hydrological simulations, which use a higher spatial model resolution only at those time steps when it is necessary. The idea behind adaptive clustering is similar to adaptive time-stepping (e.g. *Minkoff and Kridler,*





*2006*). However, instead of reducing the time steps during times when large gradients prevail adaptive clustering increases or decreases the number of independent spatial model elements during times of high functional diversity. The general concept behind adaptive clustering is thereby not entirely new to environmental science and is already used for instance in hydrogeology under the term adaptive mesh here with the main focus to increase the resolution

of gradients during times of high dynamics (*Berger and Oliger, 1984)*. The main difference between the adaptive mesh and adaptive clustering approach is that instead of adjusting the actual numerical model grid during runtime adaptive clustering changes the number of hydrological response units (HRU) that are used (needed) to represent a catchment. This implies that also the degree of spatial heterogeneity of the catchment state (e.g. the wetness state, energy state, etc.) that is covered by the model is dynamically changing.

While the idea of adaptive clustering is promising as it allows a minimum adequate representations of the spatial variability of a hydrological landscape, it has to our knowledge so far only been tested within a simple top-down model (*Ehret et al. (2020))*. It is thus of interest whether such a dynamic clustering is also feasible when using a physically based (bottom-up) model particularly as these models were specifically introduced to explore how

system characteristics and driving gradients control hydrological dynamics (*Freeze and Harlan, 1969)*. Here we will hence test and develop an adaptive clustering approach using straightforward physical reasoning and implement it into a distributed bottom-up model. The underlying objective is to exploit the value of adaptive clustering as a tool to better understand the temporal relevance of distributed precipitation for the runoff generation of a meso-scale catchment and as by-product reiterate that adaptive clustering could potentially be used to reduce

computational times as already discussed in detail by *Ehret et al. (2020)*. High computational times are thereby still one of the many reasons why bottom-up are rarely used on larger scales in an spatial explicit manner (*Clark et al., 2017)*. For instance, *Hopp and McDonnell (2009)* used the HYDRUS 3D model (*Simunek et al., 2016)* and reported computational times ranging from 10 min up to 11 hrs when they simulated water fluxes and state variables at the Panola hillslope (area = 0.001250 $km^2$ (25 m x 50 m); maximal soil depths = 4 m) for a simulation

time of 15 days. A meaningful application of bottom-up models at relevant management scales (around 250 $km^2$ in south Germany e.g. *Loritz, 2019)*, without a violation of important physical constraints (e.g. $10^{-2}$ - $10^1$ m maximum vertical grid size for the Richards equation; *Or et al., 2015; Vogel and Ippisch, 2008)*, would thus imply long computational times. This again strongly limits the number of feasible model runs to examine, for instance, different parameter sets (*Beven and Freer, 2001)*.

In this study, we test the hypothesis if adaptive clustering is a feasible approach to represent the spatial variability of rainfall in a hydrological bottom-up model at the lowest sufficient level of detail without losing predictive





performance compared to a fully distributed model. We test this hypothesis by introducing a clustering approach at the example of the model CATFLOW, which is applied to the 19.4 km$^2$ large Colpach catchment using a gridded radar-based quantitative rainfall estimate by addressing the two following research questions:

1. Does the model performance of a spatially aggregated model improve if it is distributed in space and driven by distributed rainfall?
   2. Can adaptive clustering be used to distribute a bottom-up model in space that it is capable to represent relevant spatial differences in the system state and precipitation forcing at the least sufficient resolution to avoid being highly redundant as a fully distributed model?

**2 Study area, hydrological model and meteorological data**

**2.1 The Colpach catchment**

The 19.4 km$^2$ Colpach catchment is located in northern Luxembourg and is a headwater catchment of the 256 km$^2$ large Attert experimental basin (Fig. 1). The prevailing geology of both catchments are Devonian schists of the Ardennes massif which are characterized by shallow, coarse-grained, and highly permeable soils (> 1 m; e.g.

*Jackisch et al., 2017; Juilleret et al., 2011*). The steep hills of the Colpach are primarily forested and the elevation of the Colpach ranges from 265 to 512 m a.s.l.. Annual runoff coefficients varied around 50 % ± 7 % for the 2011-2017 period. Precipitation is evenly distributed across the seasons (vegetation and winter season), while the runoff generation has a distinct seasonal pattern as around 80 % of the annual discharge is being released between October and March (*Seibert et al., 2017*). The Colpach and its sub-catchments (e.g. Weierbach) have been used as study

area in a series of scientific publications. We refer here to *Pfister et al. (2018)*, *Jackisch (2015)* or *Loritz et al. (2017)* for more detailed system description (mean annual precip: 900 – 1000 mm yr$^{-1}$; mean annual evapotranspiration: 450 – 550 mm yr$^{-1}$; mean annual discharge: 450 – 550 mm yr$^{-1}$; land-use: 65 % forest; 23 % agriculture; 2 % others; mean annual temperature: 9.1 °C).

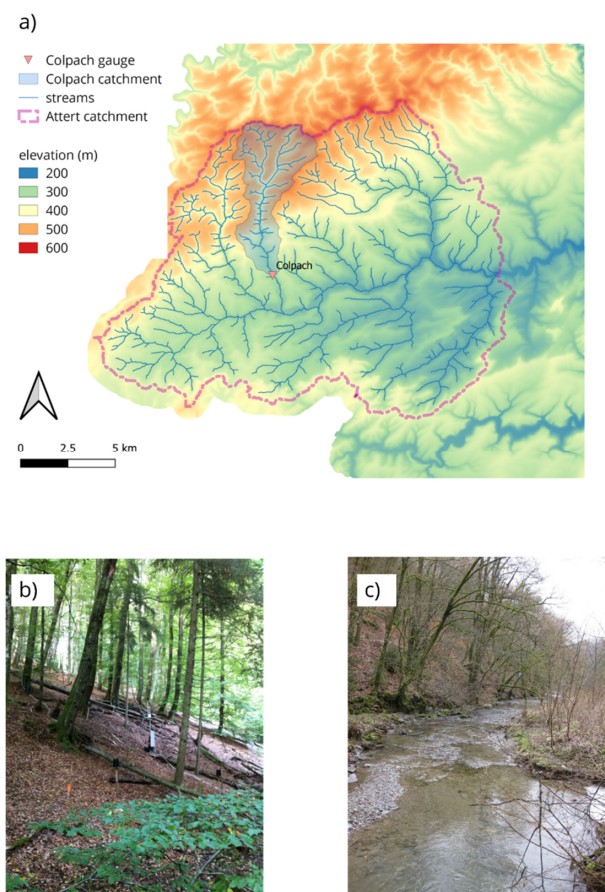

**Figure 1. a) map of the Colpach catchment (location northern Luxembourg), b) picture of a typical forested hillslope within the Colpach catchment, c) the Colpach river around 4 km north of the gauging station.**

## 2.2 The CATFLOW model

5    The key elements of the CATFLOW Model (*Maurer, 1997; Zehe et al., 2001*) are 2d hillslopes which are discretized along a 2-dimensional cross-section using curvilinear orthogonal coordinates. Evapotranspiration is represented using an advanced SVAT (soil–vegetation–atmosphere transfer) approach based on the Penman-Monteith equation, which accounts for tabulated vegetation dynamics, albedo as a function of soil moisture, and the impact of local topography on wind speed and radiation. Soil water dynamics are simulated based on the Darcy-





Richards equation and surface runoff is represented by a diffusion wave approximation of the Saint Venant equations using an adaptive time stepping. Vertical and lateral preferential flow paths are represented as connected pathways containing an artificial porous medium with high hydraulic conductivity and very low retention. The hillslope module is designed to simulate infiltration excess runoff, saturation excess runoff, re-infiltration of surface runoff, lateral water flow in the subsurface, return flow, but cannot handle snowfall or snow accumulation. The latter means that CATFLOW should not be applied if snow is a dominated control, which is not the case in the Colpach catchment. The model core is written entirely in FORTRAN77 and the individual hillslopes can be run in parallel on different CPUs to assure low computation times and high performance of the numerical scheme. Up to date model descriptions can be found in *Wienhöfer and Zehe (2014)* or in *Loritz et al. (2017)*.

### 2.3 Model forcing and observed discharge

Meteorological input data used here are recorded at a temporal resolution of 1 hr at two official meteorological stations by the "*Administration des Services Techniques de l'Agriculture Luxembourg*" at the locations "*Roodt*" and "*Useldange*". The meteorological station "*Roodt*" measures rainfall within the catchment border (Fig. 2 a) and provided the precipitation input to the model of *Loritz et al. (2017)*. The second station "*Useldange*" is located outside the catchment around 8 km west of the Colpach outlet measures air temperature, relative humidity, wind speed, and global radiation. These data are used as meteorological input (except for precipitation) in all model setups in this study. In other words, this means that all model setups in this study are forced by identical meteorological inputs except for the precipitation data (see section 3.1). Therefore, we cannot account for variations of the wind speed or the temperature within the Colpach catchment. A detailed description and analysis of the meteorological data can be found in *Loritz et al. (2017)*.

Quality checked discharge observations of the Colpach are provided by the Luxembourg Institute of Science and Technology (LIST) in a 15 min temporal resolution for the hydrological year 2013/14. The data was aggregated to an hourly temporal resolution and transformed to specific discharge given the catchment area of 19.4 km$^2$.

### 2.4 Spatially resolved precipitation data

Besides the precipitation data from the ground station located in "*Roodt*", we use a gridded quantitative precipitation estimate, which merges weather radar with rain gauge and disdrometer observations (*Neuper and Ehret, 2019*). The two used radar stations are located 40 to 70 km, respectively 24 to 44 km, away from the study site (Neuheilenbach; Germany, Wideumont, Belgium) and are operated by the German Weather Service (DWD) as well as by the Royal Meteorological Institute of Belgium (RMI). Both distances are within a range that the data



can be used at a high-resolution of 1x1 km$^2$ as the signal is neither degraded by beam spreading nor impacted by partial blindness through cone of silence issues (e.g. *Neuper and Ehret (2019)*). The raw data, 10 min reflectivity data from single pol C-Band Doppler radar, were aggregated to hourly averages as well as filtered by static, Doppler clutter filters, and bright-band correction following *Hannesen (1998)*. Second trip echoes and obvious

anomalous propagation echoes were manually removed and the corrected data were used to create a pseudo plan position indicator data set at 1500 m above the ground. A more detailed description of how the reflectivity data was transformed to rainfall data, calibrated as well as validated against rain gauges and disdrometers can be found in the appendix.

The chosen precipitation time series starts on the 1$^{st}$ of October 2013 and ends on the 30$^{th}$ of September 2014. 42 grid cells (1 x 1 km$^2$) of the precipitation field intersect with more than 50 % of their area with the Colpach catchment and are used in this study (Fig. 2 a). The weather radar measured an area-weighted mean of around 900 mm yr$^{-1}$ in the Colpach catchment for the selected period. This is in accordance with the reported climatic averages (900 - 1000 mm yr$^{-1}$) of this region (*Pfister et al., 2017*). The maximum hourly precipitation difference between

the grid cells in the study period is 14 mm hr$^{-1}$ (August 2014) and the maximum annual precipitation difference between the grid cells is 95 mm yr$^{-1}$ (Fig. 2 b). Temporally, the precipitation is evenly distributed over the year with around 50 % of rainfall in winter and 50 % of rainfall in summer with a short dry spell from mid-March to the end of April. There is a weak correlation between the mean elevation of the grid cells and the annual precipitation sums of 0.43. This implies that precipitation tends to be slightly higher in the northern parts of the

catchment that are also characterized by higher altitudes (Fig. 2 a).



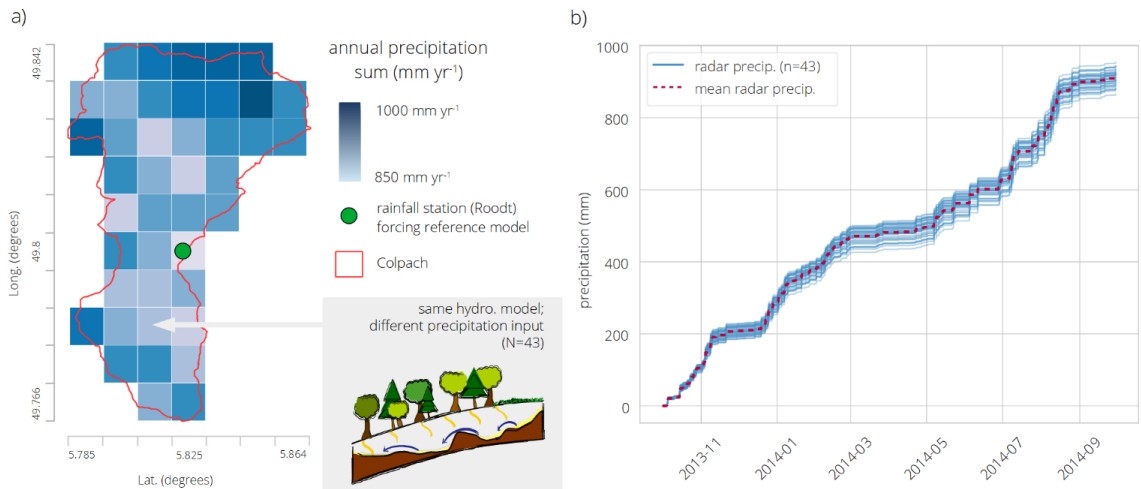

**Figure 2. a) annual sums of the gridded precipitation field over the Colpach catchment for the hydrological year 2013/14 as well as the location of the rainfall station "*Roodt*" which is used as precipitation input for the *reference model* (spatial resolution: 1 km², coord. system WGS84), b) cumulated precipitation for each grid cell for the hydrological year 2013/14**

**of the precipitation field (blue lines) and the corresponding mean of the precipitation field (dashed red line).**

## 3. Modeling approach

In the following section, we give a short introduction to the different model setups we use in this study and refer to the corresponding subsection for more detailed descriptions of each setup.

*Reference model*

The spatially aggregated *reference model* (section 3.1) was designed and intensively tested in the Colpach catchment in a previous study (*Loritz et al., 2017*). This model serves as benchmark here to a) evaluate the other three models and b) provides the structural basis for them. Moreover, are the model deficits to simulate streamflow in the summer months of the *reference model* discussed in *Loritz et al. (2017)* one of the main motivations of this

study (see section 3.2) apart from the finding of *Loritz et al. (2018)* that a suitable model structure needs to adapt its resolution in time.

*Model a*


*Model a* (section 3.2*)* is identical to the *reference model* and hence also spatially aggregated. The only difference between the models is that it is driven by different precipitation data. This precipitation data is the area-weighted mean of the spatially resolved precipitation product described in section 2.4 and measured by a weather radar. The main reason for running *model a* is to exclude that already the quantitative differences between the precipitation

data measured at the ground station "*Roodt*" and by the weather rainfall data result in a performance increase and not the spatial variability of the rainfall field.

*Model b*

The third model (*model b*; section 3.2) is a distributed version of the *reference model*. *Model b* is thereby distributed

based on the resolution of the spatially resolved precipitation data and was designed to examine the role of distributed rainfall on the runoff generation in the Colpach catchment. It represents the Colpach with 42 spatial grids (1 x 1 km²). In each of these grids, we run a model similar to the *reference model*, however, driven with the specific precipitation data measured at this location by the weather radar.

*Model c*

Other than the three above mentioned non-adaptive models (*reference model*, *model a*, *model b*) we develop a third, spatially adaptive, model (*model c*; section 3.3). This model is capable to dynamically adapt its spatial model structure in time. To dynamically allocate its structure, it uses the spatial variability and the strength of the rainfall forcing as well as its fingerprint the catchment (model) state. The main goal is to show that we can achieve similar

simulation results compared to *model b*, however, with a coarser dynamically adapting spatial model structure. We test this model at two selected rainfall-runoff events.

**3.1 Non-adaptive models – The reference model of *Loritz et al. (2017)***

All simulations in this study are based on a spatially aggregated model structure (*reference model*), developed and extensively tested in the Colpach catchment in a previous study (*Loritz et al., 2017*). The general idea behind the

proposed model concept (*representative hillslope*) is that a single bottom-up hillslope model reflects a meaningful compromise between classical top-down and bottom-up models (*Hrachowitz and Clark, 2017; Loritz, 2019*). This is the case as it allows that macroscopic model parameters can still be derived from available point measurements. The parameters of the model of *Loritz et al. (2017)* were hence, for the most part, derived directly from a large amount of field data, and the model was only afterward manually fine-tuned by further exclusively adjusting the

spatial macropore density within a few trial and error runs to simulate the seasonal water balance of the Colpach catchment. The model simulations were tested against hourly discharge observations on an annual and seasonal


time scale (as well as against a sub-basin of the Colpach) and against hourly soil moisture observations (38 sensors in 10 and 50 cm depth), and hourly normalized sap flow velocities (proxy for transpiration; 30 sensors). The developed model structure agreed well with the dynamics of the observables and showed higher model performances as reported in other studies working with different top-down model setups in the same environment

(*Wrede et al., 2015*).

### 3.2 Non-adaptive models – Model a and b

Despite the acceptable annual model performance of the *reference model*, it showed deficits to simulate the runoff response to a series of summer rainfall-runoff events. As discussed in *Loritz et al. (2017)*, one possible explanation for the unsatisfying performance is that summer precipitation in the Colpach catchment is mainly driven by

convective atmospheric conditions. These convective precipitation events are characterized by a much smaller spatial extent as well as by higher rainfall intensities compared to the stratiform and frontal precipitation events that dominate during winter (*Neuper and Ehret, 2019*). The insufficient model performance in summer could therefore likely be a consequence of the larger spatial gradients of the rainfall field compared to the winter season that cannot be accounted for in the original model of *Loritz et al. (2017)*. In other words, this entails that a

hydrological model, distributed at a sufficiently high spatial resolution, is required to capture the spatial variability of the precipitation field to satisfactorily simulate the runoff generation of the Colpach. One goal of this study is hence to test the hypothesis whether the performance deficiencies of the representative hillslope model (*reference model*) in summer are mainly caused by the inability of the setup to account for the spatial gradients of the precipitation field, rather than a result of important structural differences (e.g. soil, land-use, topography) within

the Colpach catchment.

To address the first research questions of this study: "*Does the model performance of a spatially aggregated model improve if it is distributed in space and driven by distributed rainfall*" we analyze simulations of two alternative model setups (*model a and b*) additional to the *reference model* from *Loritz et al. (2017)*:

*Model a* is identical to the *reference model*, however, driven by the area-weighted mean of the spatially resolved precipitation data described in section 2.4 (Fig. 2 b). We added *model a* to test if the performance difference between the *reference model* and our distributed *model b* is merely a result of quantitative differences between the different precipitation products measured either by a single ground station or by a weather radar.





M*odel b* is a spatially distributed version of the *reference model.* This means that all model parameters of the representative hillslope (*reference model*), as well as all other meteorological variables such as temperature or wind speed, are similar and the only two differences between the *reference model* and *model b* is that *model b* is spatially distributed as well as driven by different rainfall data. *Model b* is thereby distributed on the spatial

resolution of the precipitation field similarity as done for instance by *Prenner et al. (2018)* and not following the traditional spatial discretization strategy of CATFOW based on a fixed number of hillslopes, inferred from surface topography or land-use. We justify this assumption based on the model validation in *Loritz et al. 2017* and on a study conducted in the same research environment (*Loritz et al., 2019*) where we showed that different sub-basins of the Attert basin (the Colpach is a headwater catchment of the Attert catchment) have similar specific discharges

as long as they are located in the same geological setting and are driven by a similar meteorological forcing (see also section 3.3.2).

### 3.2.1 Model analysis

We analyze the simulation performances of *model a* and *b* by calculating the Kling-Gupta efficiency (KGE; *Kling and Gupta, 2009*) between the hourly discharge simulations of the individual models against hourly observed

discharge at different time scales (annual, seasonal, event scale). M*odel a* and *b* are hence run for the hydrological year 2013-2014 with hourly printout times and differ only concerning the precipitation data they are driven with:

- *Model a*: driven by an area-weighted mean of the spatially resolved precipitation data.
- *Model b*: driven by 42 precipitation time series each reflecting a grid cell of the precipitation field shown in Fig. 2.

To be able to compare the discharge of the spatially aggregated *model a* and the distributed *model b* with the observed discharge of the Colpach catchment and to account for the routing of the water from a specific location to the outlet, we added a simple lag function acting as channel network. The latter is based on the average distance of each grid cell to the outlet of the Colpach assuming a constant flow velocity of 1 m s$^{-1}$. For *model a*, we simply average all distances to the outlet and shift the single discharge simulation accordingly.

### 3.3 Spatially adaptive model – Model c

To address the second research question of this study: "*Can adaptive clustering be used to distribute a bottom-up model in space that it is capable to represent relevant spatial differences in the system state and precipitation forcing at the least sufficient resolution to avoid being highly redundant as a fully distributed model?*" we develop a third model setup (*model c*). This spatially adaptive model setup is based on the distributed *model b*, however, is

capable to dynamically adjust its spatial structure in time, as detailed in section 3.2.1 to 3.2.3. The underlying



adaptive clustering approach is based on straightforward physical arguments on how the spatial and temporal patterns of rainfall control the spatial pattern of the wetness state of a structural similar catchment. By structural similar, we mean that time-invariant properties of the catchment (time-invariant on the scale we are working on) like geology, topography or land-use that constrain the state space of a catchment are similarly distributed within potential hydrological sub-units of our catchment (e.g. sub-basins or hillslopes; see also section 3.2.2). We discuss the spatially adaptive *model c* for two selected rainfall-runoff events, which are characterized by distinctly different precipitation properties. By that, we examine the dynamic relationship between the spatio-temporal patterns of the rainfall forcing and its fingerprint the catchment state and show how they can be represented in a model. Full automation of the adaptive clustering approach and a test on a longer time scale is, however, beyond the scope of this study. The latter would provide only little more scientific inside (besides being technically challenging) how the variability of rainfall influences the state of a catchment and how this phenomenon can be used to dynamically allocate a model structure in time.

### 3.3.1 Spatially adaptive modeling

Spatially adaptive modeling or adaptive clustering is an approach to dynamically adjust the spatial structure of a hydrological model in time offering the possibility to reduce computational times as well as to find an appropriate, time-variant spatial model resolution *(Ehret et al. 2020)*. The basic idea of adaptive clustering has been motivated within the work of *Zehe et al. (2014)* who stated that functional similarity in a catchment (or in a model) can emerge if different sub-units are structurally similar (e.g. topography, geology, land-use, etc.), are driven by a similar forcing and are at a similar state. The latter implies that the concept of hydrological similarity, which is frequently used as the basis to discretize a catchment in space (e.g. *Wagener et al., 2007*), cannot be time-invariant but needs to dynamically change in time as corroborated by *Loritz et al. (2018)*. This is the case as the relevance and interaction of different spatial patterns of the catchment structure, state and forcing also vary in time (*Woods and Sivapalan, 1999*). A suitable discretization of a catchment into similar functional units needs hence to be time-variant and one way to achieve such a dynamic model resolution is spatially adaptive modeling.

Implementing adaptive clustering into a distributed model requires specific decision thresholds that define whether spatial differences in the structure, forcing and state of potential sub-units are so large, that they need a distributed representation. This entails that if differences between the structure, forcing, or state of two or more distributed model elements (here gridded models) are below these thresholds they are by definition similar which means that they can represent each other's hydrological function. The entire idea that certain spatial model elements can represent other model elements and hence other areas of a catchment is not new and has been used frequently in





Hydrology since at least *Sivapalan et al. (1987)* where they introduced the concept of representative elementary areas. The main novelty of adaptive clustering is that hydrological similar model elements are dynamically grouped and split in the runtime of the model instead of running a constant number of functional similar elements for the entire simulation period (*Ehret et al., 2020*).

**3.3.2 Spatially adaptive modeling – similarity assumption**

Identifying periods when a given model element or hillslope can represent another one because it functions hydrologically similar is the main challenge of adaptive clustering. In this study, we subdivide the precipitation field, and the model states at each time step into equally distant bins (groups) and define those as similar if different precipitation grid cells (forcing) or different gridded hillslope models (states) occupy the same bin. This implies

that they function similarly and can thus represent each other. To give an example, imagine if 50 % of the catchment area receives rainfall of around 1 mm hr$^{-1}$ and 50 % around 2 mm hr$^{-1}$. In this specific case, we would have two occupied forcing bins (precipitation groups). In the following, we explain our time-invariant similarity assumptions for the system structure as well as our time-variant similarity assumption of the catchment (model) state and the precipitation forcing.

*Time invariant similarity of the system structure*

The first step of our adaptive clustering approach requires the identification of hydrological response units (HRUs) that potentially act similar. A sufficient criterion for this is that their structural setup (e.g. geology, land-use, etc.) and their actual state (e.g. storage) are similar at a given time step. As already mentioned in section 3.2, our

previous studies showed that different hydrological sub-units, in this case hillslopes, of the Colpach catchment, can be characterized by similar subsurface characteristics (integral filter properties). This implies a potential similar rainfall-runoff transformation when they are in a similar state. This is supported by our previous work (*Loritz et al., 2017, 2019*) which revealed that a sub-basin of the Colpach catchment (0.45 km$^2$) and a neighboring catchment (30 km$^2$) located in the same geological setting have almost identical specific discharges as long as they

are at similar states and forced by comparable amounts of precipitation. This implies that the spatial variability of the system structure within the Colpach can be represented by a single spatially aggregated model and all grid cells of the precipitation field can thus be represented by the same model with the same model parameters as long as they are in the same state and driven by the same forcing. This entails, however, also that if we extend our research area to a catchment that is divided, for instance, into two geological settings that function hydrologically differently

(regarding their filter properties) we would always need to run at least two structural different models where each of these models represents one of two geological settings.



*Time variant similarity of the precipitation forcing*

The second decision threshold we need to identify defines the minimum difference at which we consider differences in the precipitation field as relevant for the runoff generation. Simply speaking, two structural similar

hydrological units that are in the same state will only respond differently to an external forcing if the variability in the forcing has exceeded this threshold. Here, we picked a threshold of 1 mm hr$^{-1}$ upon we consider differences between precipitation observations (grid cells) as relevant. We chose this threshold as it represents a reasonable difference upon which we expect that a hydrological landscape element might function differently than another in a humid environment. This means that only if the spatial differences in the precipitation field are above 1 mm hr$^{-1}$

do we drive *model c* with different precipitation inputs.

*Time variant similarity of the catchment state*

The third assumption is to identify a threshold upon which we consider that two model elements are in the same state. This means that we need to select a point in time after a spatially variable rainfall event (> 1 mm hr$^{-1}$) when

two or more models in the individual grid cells have "*forgotten*" the differences between them introduced by the interplay of the previous precipitation signal with drainage and evaporation dynamics. Here, we use the change in discharge over time (dQ dt$^{-1}$; slope of the simulated hydrograph) to infer similar model states. By that, we expect that two or more gridded models are again in the same state if their runoff simulations change in a similar range (0.05 mm hr$^{-1}$). As soon as this is the case and two or more gridded models are in the same state, we average their

states (average saturation of each grid cell of the CATFLOW hillslope grid) and by that, aggregate the models back again into a single hillslope. The value of 0.05 mm hr$^{-1}$ was picked as it reflects the desired precision of the adaptive model.

### 3.3.3 Spatially adaptive modeling - model implementation

As stated in section 3.3.2, we classified the entire Colpach catchment as hydrologically similar concerning the

runoff generation as long as the different hydrological sub-units of the catchment are in the same state and receive a comparable forcing. This means that we start the simulation with one gridded hillslope to represent the entire catchment and continue in this mode as long as we have not detected a spatial difference in the precipitation field above the selected threshold of 1 mm hr$^{-1}$ (Fig. 3, t=0). At each time step, we bin the precipitation input of the next time step and determine the number of allocated bins (*P = number of precipitation bins*). If more than one

precipitation bin is occupied (*P > 0*) we increase the number of gridded models (*M = no. of running gridded models*) by running the same model in the same initial state, however, driven by different precipitation inputs.





Imagine a scenario where the Colpach catchment is represented by one hillslope ($S = 1$) and we observe a precipitation event where 50 % of the catchment receives no precipitation, 20 % 7 mm hr$^{-1}$ and 30 % 8 mm hr$^{-1}$ (Fig. 3, t=1). This would mean that three precipitation bins are allocated ($P = 3$) and hence we need to increase

the number of running models also to three ($M = 3$). After running these three models for one time step with the different precipitation inputs, we bin the model states (dQ dt$^{-1}$). Let us assume we would identify two occupied model state bins, which means that two different model states ($S = 2$) are needed to represent the variability of catchment states. This could happen if the differences between the 7 mm hr$^{-1}$ and 8 mm hr$^{-1}$ rainfall intensity did not result in a significant difference in the discharge simulation of the two corresponding models. Following our

approach, we aggregate the two models that are driven by 7 mm hr$^{-1}$ and 8 mm hr$^{-1}$ by averaging their states. We do this by averaging the relative saturation of the corresponding CATFLOW hillslope grids, which is straightforward in our study as they have the same width as well as lateral and vertical dimensions. In case that the hillslopes would not be structural similar this requires a weighted averaging of soil water contents to avoid a violation of mass conservation. After the aggregation of the models, we have two model states ($S = 2$) each

representing 50 % of the catchment area.

If there is no further rainfall occurring we wait until the gradients in system states have been depleted and the two running models have "*forgotten*" the difference in the past forcing and both predict similar dQ dt$^{-1}$ values and aggregate the two models again two one gridded model. If rainfall continues in the next time step ($P > 1$) we need

to check which model states ($S$) receive which forcing. For instance, given our hypothetical example, we know that after the last simulation step we needed two model states ($S = 2$) to represent our catchment. Each of these two states represents 50 % of the area of the catchment. At the next time step, we observe a precipitation event where 50 % of the catchment receives 8 mm hr$^{-1}$ and the other 50 % 3 mm hr$^{-1}$ (Fig. 3, t = 2). In this case, we have to check if the two model states ($S = 2$) receive both precipitation inputs of 8 and 3 mm hr$^{-1}$. Let us assume that

one model state is receiving 80 % of the 8 mm hr$^{-1}$ and 20 % of 3 mm hr$^{-1}$ rainfall. The other model 20 % of the 8 mm hr$^{-1}$ and 80 % 3 mm hr$^{-1}$. In this specific setting, we would need to run four models ($M = 4$) to account for the spatial variability of the model states and precipitation input, while each of those reflect a different combination of the model state and forcing in different parts of the catchment. At this stage, we again either wait until the internal differences have been dissipated to reduce the number of models or we increase the number of models in

case that precipitation with larger spatial variability of P = 1 is continuing (Fig. 3, t = n). The maximum number of models we could require in our adaptive clustering approach depends on the maximum resolution of the precipitation input upon we divided the Colpach catchment and is 42 in our study.



### 3.3.4 Spatially adaptive modeling - model analysis

To test our spatially adaptive *model c* against the observed discharge of the catchment, we route the simulated runoff contributions according to their location to the outlet by assuming a mean flow velocity of water within the channel network of 1 m s$^{-1}$. However, as the same model can represent different grids with different locations we

additionally need to calculate the average flow distances to the outlet of all grids a model is representing and shift the simulation by the average distance accordingly. We then take the area-weighted mean of every simulation at each time step. The performance of the adaptive *model c* is then measured by the KGE against the observed discharge and the area-weighted average of the distributed *model b*. The latter addresses our second research question and follows the logic that an appropriate adaptive model should lead to similar simulations as a fully

distributed model, however, with fewer model elements. While we use CATFLOW as a model here, the proposed approach is not restricted to this model and can be used in any hydrological model that distributes a catchment into independent spatial units.

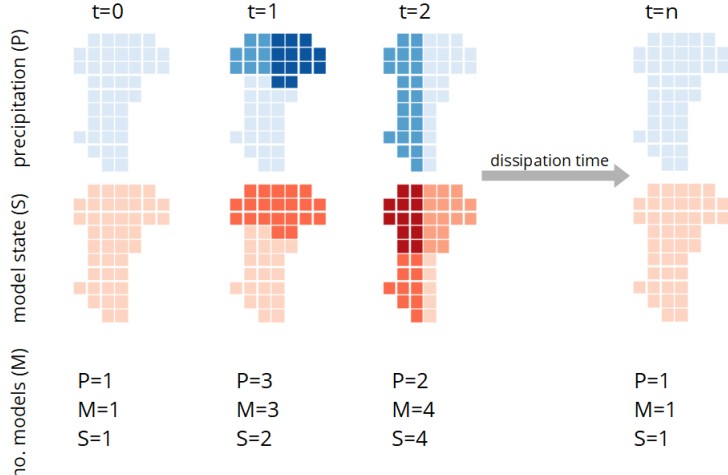

**Figure 3. Sketch of the spatial adaptive modeling described in section 3.3.3. The upper panel shows the precipitation forcing (blue) and the lower panel the model states (red). The numbers below the figures indicate how many precipitation (P), model state (S) bins (groups) are occupied and how many models (M) are running at the given time step.**





## 4. Results

In the following section, we investigate the precipitation field and compare the performance of the discharge simulations of the *reference model, model a* and *b* at the annual, seasonal, and event scale by comparing the simulations against hourly observed discharge data. We, furthermore, present the results of the adaptive modeling

for two selected rainfall events and the spatial distribution of the precipitation forcing as well as the model states of *model c* for rainfall event I.

### 4.1 Precipitation characteristics

While rainfall sums are equally distributed between the winter (Oct. – Mar.) and vegetation season (Apr. – Sep.) in the selected hydrological year 2013/14 (Fig. 2 b), the rainfall intensities and the associated standard deviation

(here used to measure the spatial variability of the precipitation field) of the precipitation field are in general higher in summer (Fig. 4 a & b). For instance, the five rainfall events with the highest rainfall intensities as well as the highest standard deviation in space were all observed in the summer season. Rainfall intensity and spatial variability are thereby strongly linked to each other which is reflected in their linear correlation of 0.82. The latter is no surprise as convective storms, which dominate the precipitation generation in summer, are typically

characterized by higher spatial variabilities and higher rainfall intensities. This finding is surely neither surprising nor limited to the chosen research environment (e.g. *Hrachowitz and Weiler, 2011; Wilson et al., 1979*) but it confirms one of our initial assumptions that rainfall is spatially more diverse in the summer season compared to the winter months in the Colpach catchment.

We selected two rainfall-runoff events to test the adaptive *model c* (Fig. 4, *time of the events are indicated by the red horizontal bars*). We chose the first event as it has the highest rainfall intensity of 19 mm hr$^{-1}$ and the third-highest spatial variability measured by the standard deviation of 3.8 mm hr$^{-1}$ in the time series as well as a distinct runoff reaction. Rainfall event I was observed at the beginning of August, lasted for about 5 hrs and the highest spatial differences between the grid cells of 14 mm hr$^{-1}$ was reached right at the beginning of the event (Fig. 5 and

6). The rainfall event moved from west to east over the catchment and reached its maximum rainfall intensity after approximately 3 hrs. No rainfall had occurred before the event for a period of 102 hrs. We can hence assume that the catchment was in a moderately dry state before the event which is also indicated by soil moisture measurements presented in *Loritz et al. (2017).*



The second rainfall event was selected as it has distinctly different properties (low spatial variability, low intensity, longer duration) when compared to the first event. Event II has a maximum rainfall intensity of 5.8 mm hr$^{-1}$ and a maximum spatial difference between the grid cells of 4 mm hr$^{-1}$. The event lasted for around 15 hrs, there was no rainfall observed 20 hrs before the event but more than 36 mm of rainfall in the previous three days. We can hence

5     assume that the soils in the catchment where rather wet which is again supported by the soil moisture measurements presented in *Loritz et al. (2017)*.

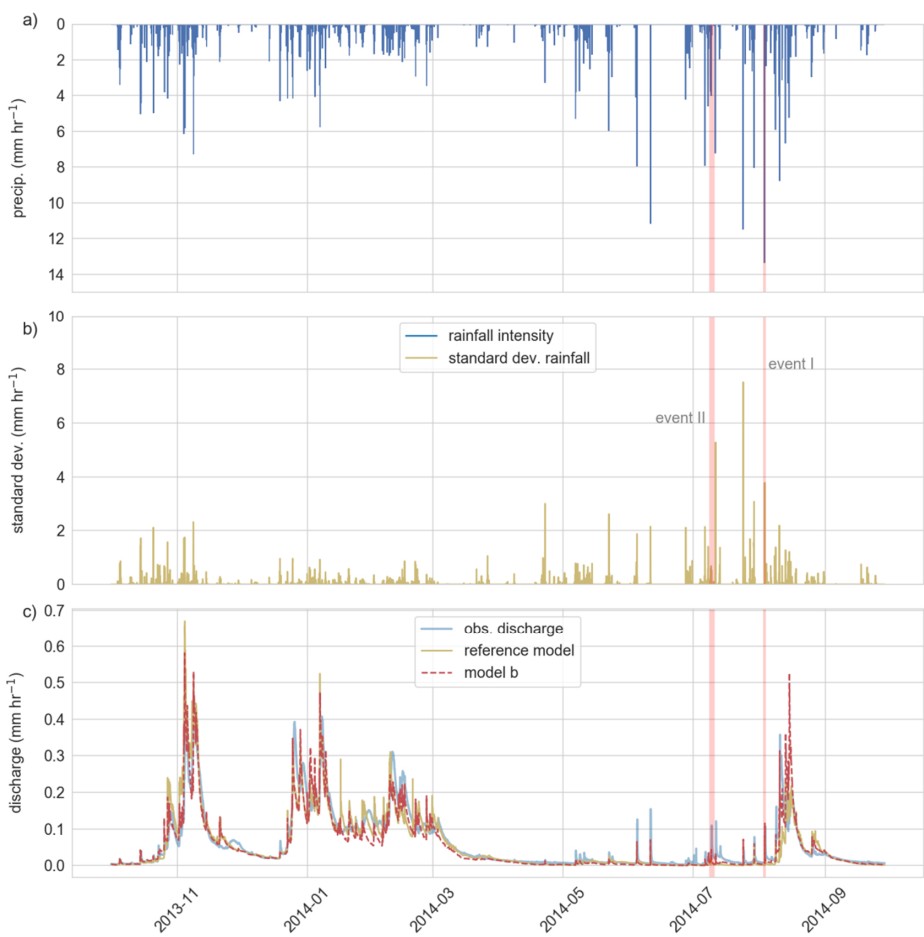

**Figure 4. a) average rainfall intensity of the precipitation field (mm hr⁻¹), b) corresponding standard deviation of the precipitation field (mm hr⁻¹), c) observed discharge of the Colpach catchment and the discharge simulation of the reference model as well as of the distributed model b. The two red bars highlight the location of the two selected rainfall-runoff events used to test the adaptive clustering approach.**

**4.2 Temporal dependency of the model performance**

The performances of the four model setups (*reference model, model a, b* and *c*) to simulate the observed discharge of the Colpach catchment measured by means of the KGE are shown in Tab. 1. If one compares the two spatially aggregated models that differ only with respect to their rainfall forcing the *reference model* outperforms *model a*





during the winter season and on the annual time scale while m*odel a* has a higher performance in the vegetation season (Apr. – Sep.). Both models are characterized by KGE values larger than 0.8 in the winter season and for the entire simulation period while the predictive performance drops in summer and is particularly low for the two rainfall-runoff events resulting even in negative KGE values. The differences between the KGE values (ΔKGE)

between the two spatially aggregated models (*reference model* and *model a*) are low in winter, increase in summer, and are the highest for the convective rainfall event I. Here does the *reference model* only have a similar performance as the mean of the discharge time series indicated by a KGE value of -0.41 (please note that it is not zero as in the case when using the Nash-Sutcliff efficiency as shown by *Knoben et al. (2019)*).

The observed discharge of the Colpach catchment, the discharge simulations of the *reference model* as well as the discharge simulation of the distributed *model b* are presented in Fig. 4 c. The visual comparison of the two models shows that the *reference model* has a lower runoff production during summer, which is particularly visible in August and September. Interestingly, the latter cannot be explained by the annual or seasonal precipitation sums as both models are driven by on average similar precipitation sums of around 900 mm yr$^{-1}$ for the entire year and

around 450 mm 6 month$^{-1}$ in the summer season. Overall, *model b* has the highest predictive performance measured by means of the KGE in all five test periods (annual, winter, summer, and the two selected rainfall events) if compared to the two spatially aggregated models. The absolute differences between the model performances depend again on the selected period. For instance, for the entire simulation period, the *reference model* and *model b* have close to equal KGE values around 0.9 while the differences between the KGE values are in summer ΔKGE

= 0.2 and for the spatially variable rainfall event I around ΔKGE = 0.7.

Although *model b* has the highest KGE values for the two selected rainfall-runoff events, the general model performance is, given the KGE values of 0.29 and 0.1, still relatively low for both runoff events. The low performance can be explained by a general underestimation of the total runoff volume at both events (Fig. 7), while

it seems that the shape of the hydrograph is simulated acceptable. The latter is supported by the fact that the distributed *model b* is capable to simulate the observed double peak at event I. Furthermore, we tested the addition of a direct runoff component by assuming that 10 % of the rainfall is directly added to the channel network instead of falling on the hillslopes. This model extension could be justified by sealed areas within the catchment, by precipitation that directly falls into the stream or on saturated areas like the riparian zone and increases the KGE

of *model b* from 0.29 to 0.48 at event I. However, we do not update our model here as the main goal of this study is not to perform the best possible rainfall-runoff simulation but to investigate the role of the spatio-temporal patterns of the rainfall for the runoff generation of a mesoscale catchment.





**Table 1. Model performances of the four model setups to simulate the observed discharge of the Colpach catchment, which are measured by using the Kling-Gupta efficiency (KGE) based on the hourly simulation and observation time steps. Performances are shown for the entire hydrological year (2013/2014), for the winter (Oct. – Mar.) and summer**
**season (Apr. – Sep.) as well as for two selected summer rainfall-runoff events in July and August.**

| | annual performance (KGE) | winter performance (KGE) | summer performance (KGE) | rainfall event I (KGE) | rainfall event II (KGE) |
|---|---|---|---|---|---|
| reference model from Loritz et al. (2017) | 0.88 | 0.88 | 0.52 | -0.41 | -0.09 |
| model a (spatially-aggregated) | 0.85 | 0.84 | 0.65 | -0.16 | -0.05 |
| model b (distributed model) | 0.91 | 0.89 | 0.73 | 0.29 | 0.1 |
| model c (adaptive model) | - | - | - | 0.29 | 0.1 |

### 4.3 Spatially adaptive modeling - simulation results

The upper panel of Fig. 5 shows the binned precipitation field of rainfall event I. The precipitation field was binned based on the chosen bin width of 1 mm hr$^{-1}$. The rainfall field allocates 0 bins (precipitation groups) at t = 0 (P = 
0), 12 bins at t = 1 (P = 12), 12 bins at t = 2 (P = 12), 3 bins at t = 3 (P = 3) and 2 bins at t = 4 (P = 2). The number of occupied bins indicates the spatial variability of the rainfall event at a given time step and would reach maximum spatial complexity if P equals 42. This means that if a high number of bins is allocated the forcing is spatially variable and respectively a higher number of models is needed to represent the spatial variability of the precipitation. The number of bins does, however, not specify how large the gradients are within the spatial 
precipitation field. For instance, if 50 % of a precipitation field is characterized by a rainfall amount of 20 mm hr$^{-1}$ and the other 50 % by 1 mm hr$^{-1}$ the number of allocated bins is two although the absolute difference between the bins is large.

The lower panels of Fig. 5 and Fig. 6 show the binning of the model states (*S*) of the adaptive model for each time 
step of event I. At t = 0, we run a single model representing the entire catchment with a single model state. At t = 1, the precipitation starts and the spatial field is classified into 12 bins (P = 12). Following our approach, this




entails that we need to run 12 models (M = 12) at t = 1 to account for the spatial variability of the rainfall. After one simulation step, we estimate the number of model states by binning the slope of the discharge simulations of the 12 models resulting in two different model states (as two model state bins are occupied). Each of these states represents now a different part of the catchment with a different area (Fig. 6, lower panel). For instance, at t = 1 around 76 % of the catchment area is represented by a model in a state where discharge changes below 0.05 mm hr$^{-1}$ and 14 % between 0.05 and 0.1 mm hr$^{-1}$. At t = 2, the precipitation field has again been classified into 12 bins but at this time step, the catchment is represented by two model states from the time step before. This means we need to check which combinations of states and precipitation input occur. In other words, which grids are represented by which state and are forced by which precipitation input. In this specific setting, we need to run 16 models which is lower as the theoretical maximum (2 model states (*S*) x 12 precipitation bins (*P*) = 24 running models (*M*)) as not all model states are driven by all grouped precipitation inputs. Afterward, we again group the model states (*S = 4*) and continue until t = 4 after which no rainfall occurs and we again represent the entire catchment by a single model. In total, we were able to reduce the maximum number of gridded models from 42 to a maximum of 16 at rainfall event I and at the second event from 42 to 4 without a predictive performance loss in comparison to the distributed *model b* (Tab. 1). The latter is, besides the comparison of *model c* with the observed discharge, also shown by the high KGE values between the distributed *model b* and the adaptive *model c* of around 0.98 at both events.

### 4.4 Spatially adaptive modeling – dissipation of differences

The dissipation timescale (*memory timescale*) at both events until the different hillslope models have "*forgotten*" the last forcing and are again in the same "*runoff generation state*" is relatively short. More specifically, already after 1 hr of no precipitation at event I and II the differences between the runoff generation of the hillslope models in *model c* are below the picked threshold of 0.05 mm hr$^{-1}$. This means that our *model c* would represent the entire catchment with a single hillslope model until a new rainfall event (P > 1) occurs. This picture is quite different for the soil moisture distribution between the hillslopes, at least in deeper soil layers. For instance, Fig. 8 shows the soil moisture distribution of two hillslope models in 10 to 20 and 60 to 100 cm depth which either has received the highest amount of rainfall measured at a given grid cell at event I (30 mm, 5 hr$^{-1}$) or the lowest (15 mm, 5 hr$^{-1}$) for two different time steps during and after event I (t = 3 and t = 24; see Fig. 5). Both hillslope models started in a similar initial model state and Fig. 8 only shows the wetness of the soil matrix. The memory time scale of the topsoil correlates thereby quite well with the runoff generation and we observe the largest difference between the "wettest" model which has received the highest amount of rainfall and the "driest" model which has received the lowest amount of rainfall at t = 3 after the highest rainfall intensity (see Fig. 5). After 24 hrs, this difference persists



but it slowly dissipates and has almost completely disappeared after 48 hrs. In the deeper soil layer, the picture is different. During the event, we see no reaction to the rainfall forcing of the soil matrix and water bypasses these areas through preferential flow paths. However, 24 hrs after the first rainfall of event I the difference between the models regarding their soil moisture distributions in deeper layers is slowly increasing although there was no

5 further rainfall. The latter means that by aggregating the different hillslope models, as done in our adaptive *model c* after only one hour of no rainfall, we delete the difference between the soil moisture distributions. As we use the mean to aggregate our models, we are, however, still conserving mass. The question remains of how important these differences are on longer time scales or for the root water uptake.

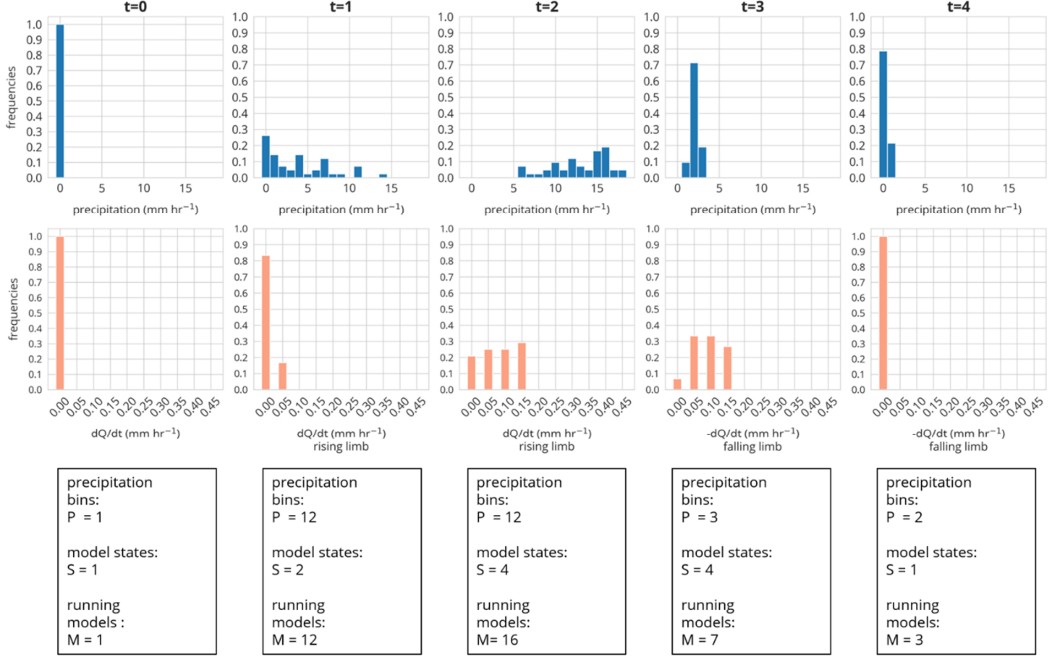

**Figure 5. Binned precipitation field (blue) and binned model states (orange) of the adaptive model (t = 0; August 3rd 2014 15:00 CET); P = no. of allocated precipitation bins, S = no. of allocated model space bins, M = no. of running models at the given time step. The spatial distribution of the precipitation and the model states for event I are displayed in Fig. 6.**





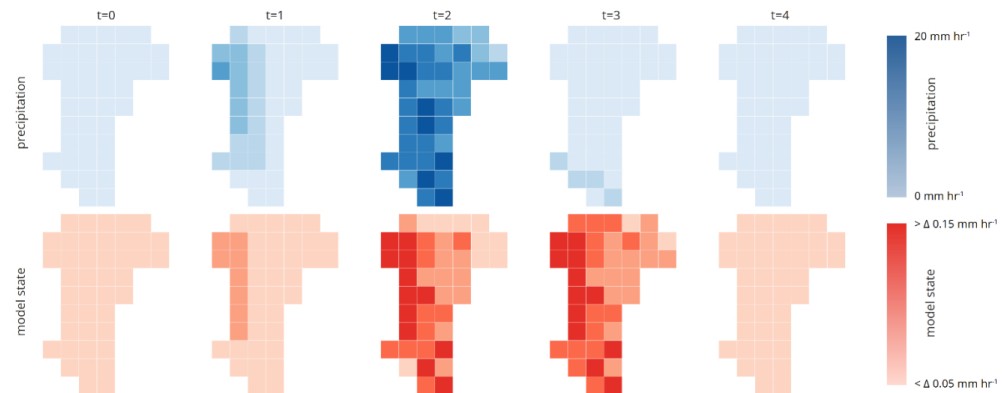

**Figure 6. Spatial and temporal distribution of the precipitation field (upper panel) and the corresponding states of the actual model grids used by the adaptive model c (lower panel). The model state is estimated by the slope of the simulated discharge. The corresponding bins (groups) of the precipitation and model states are shown in Fig. 5.**

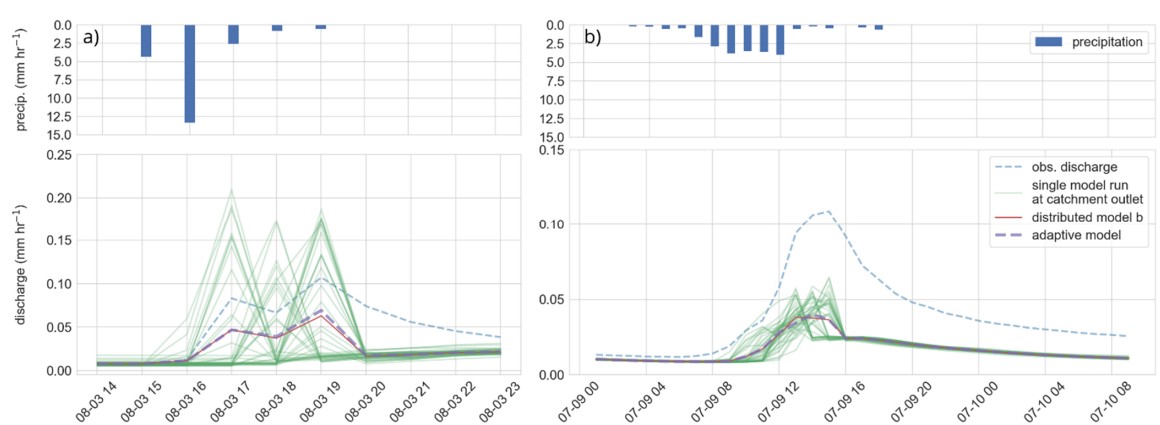

**Figure 7. a) rainfall-runoff event I and b) rainfall-runoff event II. Blue bars in the upper panel show the average precipitation of the precipitation field for each time step (mm hr⁻¹). The green curves in the lower panel represent a single gridded model of the distributed model b; red line the area-weighted mean of the distributed model; purple dashed line the area-weighted mean of the adaptive model and dashed blue line the observed specific discharge of the Colpach.**



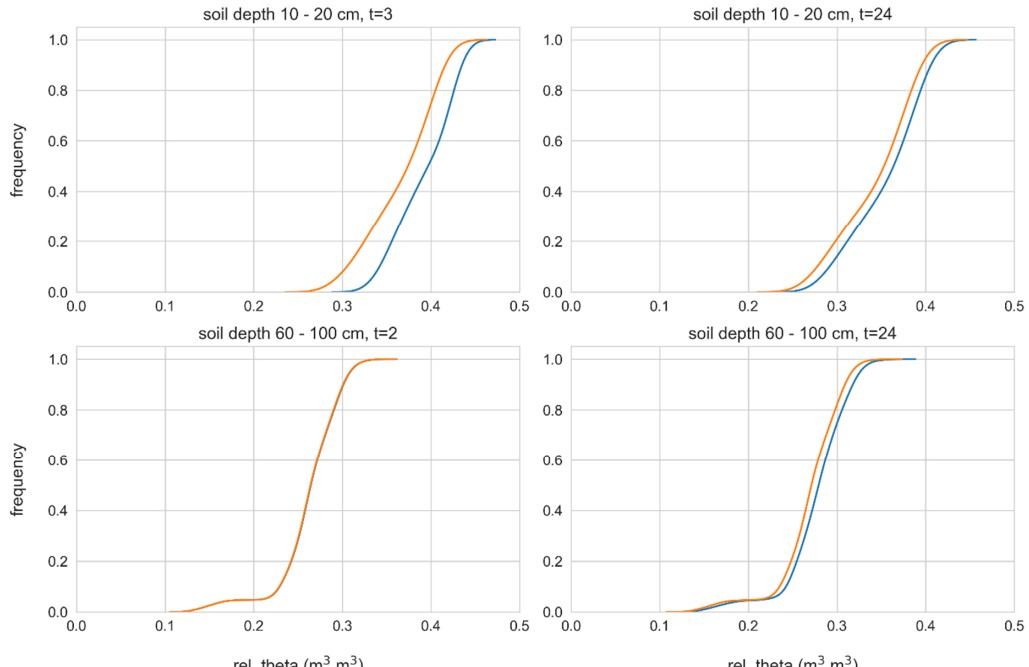

**Figure 8. Relative soil moisture distributions for two gridded hillslope models of *model b* that received the lowest (orange curve) respectively the highest (blue curve) amount of rainfall during event I (15 mm 5 hr⁻¹ and 30 mm 5 hr⁻¹). Presented for time step t = 3 (during the event) and t = 24 (after the event).**

## 5. Discussion

### 5.1 The role and value of distributed rainfall for hydrological models

While the three non-adaptive model setups (*reference model, model a,* and *b*) perform equally well with respect to simulate the discharge of the Colpach catchment in the winter season, this is not the case in the summer season where the distributed model has higher KGE values as both spatially aggregated models. This corroborates our hypothesis stating that the predictive performance of the spatially aggregated *reference model* introduced by *Loritz et al. (2017)* increases if the model is distributed in space and driven by distributed rainfall. However, *model b* has still several deficiencies especially at the two selected rainfall-runoff events where it does underestimate the total





observed runoff volume. This indicates that there is still room for further improvements of the hydrological model to increase its predictive performance.

Although the model comparison in this study is rather heuristic (e.g. we used only a single performance metric etc.), the findings highlight that the use of distributed rainfall is recommended during the summer season for simulations of the runoff generation of the Colpach. This insight is consistent with the findings of *Euser et al. (2015)* and *Wilson et al. (1979)* who showed similar results in a 1600 and 42 km$^2$ large catchment. On the other hand, the winter performance of *model b* did not improve in comparison to the *reference model*. In line with these results, *Obled et al. (1994)* and *Das et al. (2008)* concluded that "*the spatial variability of rainfall, although important, is not sufficiently organized in time and space to overcome the effects of smoothing and dampening when running off through this rural medium-sized catchment*.". Given the rather small size of the Colpach catchment and the fact that the use of distributed rainfall increased the performance during the summer, it seems that catchment size might not be the best indicator to decide if or if not a distributed hydrological model driven by distributed rainfall is needed. The higher relevance of spatially distributed precipitation for hydrological modeling in the summer months surely reflects the circumstance that also the average rainstorm size as well as the average rainfall intensities change between the seasons and are on average much higher during summer (*Neuper and Ehret, 2019*). Given the changing meteorological regime, it seems reasonable to link the increasing relevance of distributed rainfall to these changes. The fact that average storm size over a catchment is a key indicator to identify the role of distributed precipitation on the runoff generation of a catchment was also pointed out by *Nicótina et al. (2008)*. As the dominant rainfall generation mechanisms change during the hydrological year in many humid catchments from frontal to convective, it comes from a physical perspective, not as a surprise that also the relevance of distributed rainfall differs between the seasons or even between different rainfall-runoff events.

The evaluation of the model performances highlights that the necessary spatial model structure does not only change between seasons but more importantly from rainfall event to rainfall event. This idea has already been highlighted by *Watts and Calver (1991)*. They concluded that "*the finest available definition of rainfall may be desirable for modeling…*" of convective rainfall events while higher spatial model resolutions did not increase the predictive performance of their models during stratiform rainfall events. At first glance contradicting, *Lobligeois et al. (2014)* reported that the distribution of rainfall is in general of higher relevance in certain regions of France when they analyzed 3620 rainfall-runoff events of 181 mesoscale catchments. However, they also discussed that a substantial amount of rainfall-runoff events contradicted this general pattern, which shows that the distribution of rainfall can be of high importance in regions, even if the spatial precipitation patterns are usually not a dominant


control on the runoff formation. As such "rare" events are frequently linked to extremes, which are in turn beyond the realms of experience on what these landscapes have adapted to, they are of considerable importance despite their low occurrence in time (*Loritz, 2019*).

### 5.2 Spatially adaptive modeling – as a tool to reduce redundant computations

The proposed adaptive modeling approach is promising because the spatial adaptive *model c* performed similarly as the distributed *model b,* although it used a much smaller number of hillslopes. The maximum number of gridded model elements that were necessary to represent the variability of catchment states and precipitation was reduced by a factor of 2.5 for event I. The total gain in computational efficiency is however larger as most of the time less than 16 models are required to represent the catchments runoff generation.

*Clark et al. (2017)* recognized computational times as mayor obstacle when using physically-based models for practical applications, as proposed in the landmark publication of *Freeze and Harlan, (1969)*. The discussion about saving computational times with adaptive clustering is, however, challenging (*Ehret et al., 2020*) as the gain depends on the chosen model approach (e.g. numerical scheme), on the used hardware, the programming language,

the compiler or on the number of printout times of a model to the hard-drive. Furthermore, the relevance of saving computational times of, for instance, 10 % depends on the absolute calculation time of a model and hence whether a model run needs 100 min or 100 d to be completed. A fair comparison would mean to setup a virtual environment and work under similar conditions, e.g. by using a virtual machine as well as using a fully automated adaptive clustering approach. Both is, however, beyond the scope of this study and we would like to point toward the study

of *Ehret et al. (2020)* which discusses the potential of adaptive clustering with respect to computational times in detail.

### 5.3 Spatially adaptive modeling – as a tool to better understand the dissipative nature of a hydrology

In this study, we focus on the potential of adaptive modeling to examine when interactions between a variable precipitation forcing and a variable catchment state cause a variable runoff response and when these differences

get "*forgotten*" due to the dissipative nature of hydrological systems. Our results show or reiterate that the relevance of distributed rainfall for hydrological modeling is dynamically changing in space and time. One way to account for this dynamically changing relevance is to run distributed models driven by distributed rainfall the entire time at the highest possible spatial resolution. Such an approach, sometimes referred to as hyper-resolution



modeling (e.g. *Bierkens et al., 2015*), would avoid cases in which we unnecessarily underestimate the needed (spatial) model complexity of a hydrological model (e.g. *Fenicia et al., 2011b; Höge et al., 2018*). However, this procedure may lead to a strong increase of uncertainty due to an increased number of model parameters (e.g. *Beven, 1989*), result in a general overestimation of the simulated spatial variability due to error propagations within

the model as well as increase the number of redundant computations in a majority of the simulation period (*Clark et al., 2017*). The latter implies a vast amount of computations as the natural length scale (grid size) of water flow in the critical zone, which is frequently simulated by using the Darcy-Richards equation, should not exceed a lateral grid size of 10 m and vertical grid size below 1 m in homogeneous soils (*Vogel and Ippisch, 2008*). The same is true for simulating surface runoff with different diffusive wave approaches where typically much higher

flow velocities occur compared to the subsurface which again requires high resolutions and small calculation time steps. Hyper-resolution modeling without a delineation of the underlying system in independent sub-units for parallelization is hence up-to-date constrained to rather small length scales, at least if applications shall not compromise the underlying physics.

Physical constraints of small grid sizes and calculation time steps must not be a dead-end for applying bottom-up models on larger scales. This is because it is often found that different catchments in the same hydrological landscape function similarly despite the overwhelming small scale variability we often observe with point-scale measurements (e.g. *Loritz et al., 2017*). This entails a large potential to transfer information about model states from one catchment or hillslope to another (e.g. *Hrachowitz et al., 2013*) and offers the possibility to aggregate

structurally similar sub-units of a system and simulate their functioning by a single representative, as long as they are in a similar state and driven by a similar forcing (*e.g. Sivapalan et al., 1987; Zehe et al., 2014*). The fact that hydrological systems are highly dissipative but constrained by there structure is thereby the key to explain the feasibility of this dynamic grouping as the unique characteristics of the forcing over an area do not prevail but are depleted or "*forgotten*" in a relatively short time, at least if the focus is on the runoff generation. Specifically, we

found during both events that already after 1 hr of no rainfall the spatially adaptive model required only a single hillslope model to represent the diversity in the runoff generation between the models. While this finding is surely constrained by the chosen threshold, the picture is nevertheless quite different in deeper soil layers where the diversity of the rainfall forcing leads even after 24 hrs to increasing differences between the "driest" and "wettest" models. A part of the information about the different meteorological forcings between the two models is hence

still stored in the model state after 24 hrs and has not yet been dissipated. The importance of those differences likely depends on the dominant runoff generation process. In the present case, they have a minor impact as *model*



*b* and *c* show similar average baseflow simulations 24 hrs after the rainfall event I and II although *model c* uses only a single hillslope model (difference smaller than 0.001 mm hr$^{-1}$).

While the structure of a catchment constraints its state space, its actual position therein is controlled by the
meteorological forcing and by an attracting local thermodynamic equilibrium, a point where all driving gradients are depleted. As larger gradients dissipate faster than smaller ones if they are controlled by the same integral resistance properties, structural similar parts of a landscape will converge to the same state and thereby "*forget*" differences between their forcing and state. This convergence leads to the emergence of hydrological similarity in time (*Loritz et al., 2018*) and explains the changing relevance of distributed rainfall. This again is the theoretical
ground that explains why adaptive modeling works in hydrological systems and not necessarily in meteorological systems as their chaotic nature can amplify state differences on longer time scales, instead of dissipating those (e.g. *Lorenz, 1963*). Our developed adaptive modeling approach is using this straightforward physical reasoning of the causal interplay between the precipitation forcing and the catchment state to dynamically allocate its model structure. It is built upon a well-established concept in hydrology, which states that individual observations or
model states can represent each other if they are allocated to the same group (e.g. *Wood et al., 1990*). The related bin widths (grouping) can be selected either based on our physical understanding (*Loritz et al., 2018*) or identified based on a statistical analysis of the underlying distribution (of for instance the precipitation data; e.g. *Gong et al., 2014; Scott, 1979*). The general approach is strongly motivated by the idea that a spatially homogeneous field can be compressed to a single time series without losing information about the spatial pattern of rainfall. This is,
however, not the case if the spatial field is highly variable where a compression to a single observation reduces the information provided to a hydrological model and hence can average out extremes and potentially relevant spatial constellations (e.g. *Loritz et al., 2018; Weijs et al., 2013*). Spatially adaptive modeling can, therefore, be used as a tool to analyze the relevance of certain spatial detail in a hydrological model as well as to better understand the dissipative nature of hydrology.

**6. Conclusions**

In this study, we try to better understand the role and value of distributed precipitation data for the runoff generation of a mesoscale catchment. We compare the model performances of three hydrological models at different periods and show that a distributed model driven by distributed rainfall yields only to improved performances during certain periods. We then step beyond this finding and develop a spatially adaptive model that is capable to
dynamically adjust its spatial model structure in time. This model is capable to represent the varying importance





of distributed rainfall within a hydrological model without losing performance compared to a spatially distributed, gridded model. Our results confirm that spatially adaptive modeling might be a) one way to reduce computational times as already shown by *Ehret et al. (2020), b)* can be used to better understand the varying importance of spatial state and forcing differences in hydrological models and c) highlight that similarity between the runoff generation of two hillslopes does not necessarily mean similarity between other state variables (e.g. soil moisture in deeper soil layers).

The main findings of this study are:

1)  The importance of distributed rainfall on hydrological modeling is given by the natural variability of rainfall dynamically changing in time. In consequence, there cannot be a time-invariant answer to the question "*How important are spatial patterns of precipitation for the runoff generation at the catchment scale?*" nor to any related question which deals with an "*optimal*" spatial discretization of a hydrological landscape within a model.

2)  Spatially adaptive modeling is a feasible way to account for the changing importance of distributed rainfall within a hydrological model and at the same time can be used to better understand the interplay of the rainfall forcing, the catchment structure, and its state.

3)  The tested catchment is organized in a manner that spatial differences between the precipitation forcing are effectively "*forgotten*". This entails that gradients, which drive runoff, are effectively dissipated in a relatively short period. This period might, however, be quite different for other fluxes and state variables depending on the dominant runoff generation process.





**Appendix A: Detailed description of the distributed rainfall data.**

The distributed precipitation data used in this study is based on single-polarization C-band Doppler radar measurements. The mainly used radar data is from the radar located in Neuheilenbach, Germany and operated by the German Weather Service (DWD). The raw volume data set has an azimuthal resolution of 1° and a radial

resolution of 500 m. The -3dB beamwidth of the antenna is 1°. The radar site is between 40 and 70 km away from the study area. This means that the resolution is yet neither significantly degraded by the beam spreading, nor partial blinded through cone of silence issues. During the period from the 1st of October 2013 to the 27th of March 2014, the radar in Neuheilenbach was out of service due to maintenance issues. We hence used data from a radar located in Wideumont, Belgium in this period. The radar in Wideumont is operated by the Royal Meteorological

Institute of Belgium (RMI) and is also a C-band Doppler radar with the same technical specifications as the radar of the DWD. The distance between radar site in Wideumont and the study area is between 24 to 44 km. Thus, the same statements about the resolution, which were made in the case of the data from Neuheilenbach, also apply to the radar data of Wideumont.

The data was quality controlled and a correction was performed. The particular raw data was at first filtered by a static clutter filter and then also by a Doppler clutter filter. Subsequently, a bright-band correction (*Hannesen, 1998*) was applied. Occasional contamination of the data by second trip or anaprop echoes was removed by using approaches of *Bückle (2009)* and *Neuper (2009)*. Specific attenuation corrections were not applied. Furthermore, the data was carefully quality checked by an experienced radar meteorologist and operational weather forecaster,

who even spends his spare time watching radar pictures. From the corrected data a pseudo PPI (plan position indicator) data set at 1500m above ground was created and afterward an adequate (based on the synoptic situation) reflectivity-rain rate relation (Z-R relation) was applied to compute the precipitation rate (e.g. *Fabry, 2015*). In the last step, the distributed precipitation fields were checked against quality-controlled rain gauges and if necessary manually corrected.



*Data availability.* The hydrological model CATFLOW and all simulation results are available from the leading author on request. The rainfall data were provided by fourth author Malte Neuper from the Karlsruhe Institute of Technology. The discharge observations were provided by the Luxembourg Institute of Science and Technology within the "Catchments As Organized Systems (CAOS)" research group (FOR 1598) funded by the German Science Foundation (DFG). Please contact Laurent Pfister or Jean-Francois Iffly.

*Competing interests.* The authors declare that they have no conflict of interest.

*Acknowledgements.* This research contributes to the "Catchments As Organized Systems (CAOS)" research group (FOR 1598) funded by the German Science Foundation (DFG ZE 533/11-1, ZE 533/12-1). Laurent Pfister and Jean-Francois Iffly from the Luxembourg Institute of Science and Technology (LIST) are acknowledged for organizing the permissions for the experiments and providing discharge data and the digital elevation model. We also thank the whole CAOS team of phase I & II.



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
