# Peer review of "The role and value of distributed precipitation data for hydrological models"

_Hydrology and Earth System Sciences, 2020_

## Referee Comment (RC1) · Daniel Wright (Referee) · 20 Aug 2020

The authors present a framework for dynamically adapting the level of spatial detail resolved within a physics-based rainfall-runoff model depending on the spatial variability in precipitation. I found this the be one of the most interesting manuscripts that I've ever reviewed, and commend the authors on this innovative work. Nonetheless, there are some issues that should be addressed before the manuscript is suitable for publication in HESS, and that could help maximize the impact of the work.

Regards, Daniel Wright, University of Wisconsin-Madison

Major comments:

1. I believe the discussion could be strengthened by deeper consideration of how this

approach would "scale up" to larger watersheds or regions. Part of my reason for encouraging this is that the land surface modeling (LSM) community is at least as concerned as the rainfall-runoff community about model computational demands of long-term/ensemble simulations, and are seeking ways of representing fine-scale (e.g. hillslope and below) over continental-to-global domains. In fact, land surface modeling was the focus of the well-known Wood et al. (2011) hyperresolution modeling opinion piece. In addition, there has been relevant progress in LSM development that the authors should cite. I will mention these below. But in terms of scaling up, the key aspects seem to be acknowledgement that heterogeneity of model parameters will increase with modeled area, while the rainfall spatial coverage will, on average, decrease.

2. I believe the discussion could also be strengthened by some discussion of how well this approach might fit with specific types of spatial discretizations. It fits quite naturally with hillslope-based models. The fit is less clear with gridded or TIN-based models-or at least with high-resolution gridded models in which individual model grids must "communicate" with each other to transmit water via overland or subsurface flow to channels. It seems that the computational advantages of the approach might be limited in that case. In addition, models such as GSSHA in which overbank river flow can return to the land surface would have some limits here too. These issues are worth discussing because such models constitute important current directions in physics-based model development.

3. While there may be other relevant LSM developments, the one that I am aware of is Hydroblocks (Chaney et al. 2016). While I recommend reading that paper, the basic approach is similar to this manuscript's in that spatial units are grouped into hydrologically similar clusters to reduce the computational demand. The difference is that in Hydroblocks, these clusters are not dynamically reassigned according to time-varying characteristics (unless the developers have recently added that capability). So in fact, your approach appears to be superior in some respects. Specifically, within Hydroblocks, since there is no dynamic reassignment, you can never have a cluster

that extends beyond the spatial extent of a single precipitation grid cell, which means that their approach loses computational efficiency with higher-resolution precipitation datasets. Your approach thus seems to hold more promise in terms of flexibility to advances in precipitation inputs.

4. More clear description of what each model does and does not do is needed in Section 3. Specifically, I found it confusing the way that the models are briefly introduced at the beginning of the section, and then discussed further in various subsections. I also find it strange that you have text that is not assigned to specific subsections. It isn't clear why section 3.2.1 is needed... convention is that you don't include subsections unless you have at least 2 or 3 (i.e. 3.2.2, 3.2.3). This section structuring needs rethinking. More important, I really couldn't figure out from the descriptions what the differences between some models were. I also don't understand the motivation for using a different rainfall dataset for the reference model and model a; this seems unnecessary. I think one think that would really help is to not use "model a", "model b", etc. but some brief descriptive names that actually help the reader understand and recall the differences. Also, a table that compares the key features and differences of all the models could be effective.

5. Zhu et al. (2018) and Peleg et al. (2017) both highlight how distributed rainfall structure is really important in determining flood frequency across a range of scales. Though I normally refrain from suggesting that authors cite my own work, in this case it seems appropriate to highlight these studies, since they do show that for extreme events, rainfall space-time structure is extremely important in determining hydrologic response even at very small scales (see Peleg et al. in particular), and that this importance varies with rainfall magnitude and basin size. Along with this, I disagree with the statement on pg. 27: "it seems that catchment size might not be the best indicator to decide if" a distributed model is needed. It probably is the best single indicator, but is still insufficient. I draw a somewhat different conclusion from your work: that a distributed approach is always needed to reap the full benefit of spatially distributed

rainfall (at least in locations in which convective rainfall can occur), and that provides motivation for continued developments such as this into ways of handling this need in computationally-efficient ways. Likewise, I disagree with the statement on pg. 30 line 18-19: compressing rainfall into a single time series isn't so important as the ability to only use as much computational power as is truly needed to solve the problem at hand.

6. Some discussion of implications for calibration would be interesting. Is it necessary to calibrate using a fully distributed model? This would limit the usefulness of this approach in some respects such as automated calibration procedures.

7. There are a number of minor grammatical issues that nonetheless cause some distraction from the overall high quality of the manuscript. I will point out some of these below, but it could be worthwhile to have a native English speaker perform a careful proofreading.

Minor comments:

1. It may be worth defining more carefully what you and others mean by hyperresolution. There seem to be important differences-Wood et al. (2011) mention 1 sq. km or smaller, while you seem to refer to much finer scales than that. This isn't trivial since a 1 sq km gridded Richard's equation based model can, in my experience, run quickly (of course, whether such models make physical sense is another issue. . .).

2. Figure 2a: what is the small conceptual diagram to the lower left? No explanation is given. Consider deleting.

3. It may be worth more carefully explaining in the introduction what is meant by "dissipate spatial gradients efficiently"

4. Pg. 16 line 30-32: change to ". . . maximum number of precipitation grid cells (42 in this study)."

5. Pg. 1 line 9: delete "at different periods"

6. Pg. 1 line 10: change "yields… performances" to "only improved performance"

7. Pg. 1 line 12: delete "the"

8. Pg. 1 line 17: change "They furthermore" to "We also"

9. Pg. 2 line 7: Usage of "constrained" seems strange here. Consider rewording this sentence.

10. Pg. 2 line 18-19: change to "they analyzed the dependence of average runoff rates on the spatial and temporal variability…"

11. Pg. 3 line 3: change "this study" to "that study" unless you mean the specific manuscript that I am reviewing right now.

12. Pg. 3 line 12: "higher" than what? Consider changing to "… are only important during…"

13. Pg. 5 line 2: delete "large"

14. Pg. 7 line 2: delete "an"

15. Pg. 8 line 3: change "pol" to "polarimetric" and "as well as" to "and"

16. Pg. 8 line 5: add a comma after "removed"

17. Pg. 9 line 13: sentence starting with "Moreover" has some grammatical problem related the usage of "are". I'm not sure how to fix it.

18. Pg 10 line 27: delete "still"

19. Pg. 11 line 7: change to "… deficits in simulated runoff response…"

20. Various places, including Pg. 12 line 27: replace "capable to" with "able to"

21. Various places, including Pg. 13 line 2: replace "structural similar" to "structurally similar"

22. Pg. 13 line 10: replace "inside" with "insight into"

23. Various places, including Pg. 13 line 16: replace "time-variant" with "time-varying" or "temporally varied"

24. Pg. 18 line 4: delete comma after "We"

25. Pg. 18 line 5: "... the spatial distribution of the forcing..." I'm not quite sure what you're saying, perhaps because I'm still confused on the differences in the different models.

26. Pg. 21 line 6-7: The sentence beginning with "Here" is awkward. At least need to change "have" to "has", but probably needs other changes too.

27. Pg. 21 line 11: Delete "the" before "visual"

28. Pg. 21 line 12: delete "a" before "lower"

29. Pg. 21 line 25: change to "the shape of the simulated hydrograph is acceptable" and delete "the" at the end of the line

30. Pg. 23 line 10: replace "as" with "than"

31. Pg. 23 line 19: replace "at" with "for"

32. Pg. 23 line 20: delete "More"

33. Pg. 26 line 11-12: change "has still several" to "still has several"

34. Pg. 27 line 13: delete "or if not"-not needed in English.

35. Pg. 27 line 28: replace "at first glance contradicting" with "In contrast"

36. Pg. 27 line 31: replace "amount" with "number"

37. Pg. 28 line 6: replace "although it used" with "using"

38. Pg. 28 line 8: replace "less" with "fewer"

39. Pg. 29 line 12: strange usage of "up-to-date"

40. Pg. 30 line 4: at least change "constraints" to "constrains", but I'm not clear on what exactly the authors mean... probably related to subcatchment heterogeneity.

41. Pg. 30 line 28: change to "... only yields improved performance..."

42. Pg. 31 line 20: I suggest adding to this sentence "... and the specific model discretization used." This goes back to my earlier point about high-resolution gridded models, etc.

References:

Chaney, N. W., Metcalfe, P., and Wood, E. F. (2016) HydroBlocks: a field‐scale resolving land surface model for application over continental extents. Hydrol. Process., 30: 3543– 3559. doi: 10.1002/hyp.10891.

Peleg, N., F. Blumensaat, P. Molnar, S. Fatichi, and P. Burlando. "Partitioning the Impacts of Spatial and Climatological Rainfall Variability in Urban Drainage Modeling." Hydrol. Earth Syst. Sci. 21, no. 3 (March 14, 2017): 1559–72. https://doi.org/10.5194/hess-21-1559-2017.

Wood, Eric F., Joshua K. Roundy, Tara J. Troy, L. P. H. Van Beek, Marc F. P. Bierkens, Eleanor Blyth, Ad de Roo, et al. "Hyperresolution Global Land Surface Modeling: Meeting a Grand Challenge for Monitoring Earth's Terrestrial Water." Water Resources Research 47, no. 5 (2011): 1–10. https://doi.org/10.1029/2010WR010090.

Zhu, Zhihua, Daniel B. Wright, and Guo Yu. "The Impact of Rainfall Space‐Time Structure in Flood Frequency Analysis." Water Resources Research 54, no. 11 (2018): 8983–98. https://doi.org/10.1029/2018WR023550.

---

## Referee Comment (RC2) · Anonymous Referee #2 · 8 Sep 2020

This was the first time I was involved as a reviewer for this manuscript. The manuscript introduces an adaptive spatial clustering of hydrologic response units (HRU) to cope with the dynamics of the intermittent rainfall by keeping the model as parameter parsimonious (=model states) as possible in terms of reduction of similar-reacting HRUs. The manuscript is well-written and I enjoyed reading it. The introduced clustering is innovative from and fits into the scope of the journal. I have a few moderate and a number of minor comments, which are stated below. My overall recommendation would be a moderate revision to give the authors enough time to solve the open issues. Since I can only choose between minor and major revision, major revision it is.

Moderate comments:

The manuscript is about the reduction of the spatial model resolution based on the

variety of precipitation as input signal. I'm wondering if there is not an adaption of the temporal resolution required as well since scales in space and time are not independent of each other (see Melsen et al. (2015) and references therein)? Maybe it's not an issue for the small catchment studied here, but for larger catchments with a small hydrologic variability the numeric stability can be questioned due to the large spatial discretization and the high temporal resolution (e.g. in terms of the Courant-Friedrichs-Lewy condition, Courant et al., 1928). The authors should proof this condition for their model setup and discuss possible issues in the manuscript. An alternative would be to reduce the temporal resolution as well, which would lead to an additional reduction of parameters/computational costs.

The authors have selected two events to show the ability of adaptive clustering. The choice of both events seems to be very arbitrary. From Fig. 4 it seems that the resulting runoff peaks are not representative for runoff mechanisms of the catchment. As far as I understand from P13l8-10 the clustering is carried out manually and not automatically so far, which is the reason why the authors decided for two small events covering only a few time steps. However, I disagree with the hypothesis that "a test on a longer time scale . . . would provide only little more scientific inside" (P13l9-10), which is also not proven by the authors. I rather expect that the reduction of model parameters due to the adaptive spatial resolution is reduced significantly for long-lasting rainfall events causing a direct runoff response over several days as e.g. in Nov 2014, Jan 2014-Mar 2014 and Aug 2014. Another point that can be questioned is snow, which does not cause runoff immediately, but when snow melt begins. How will this be affected/can be incorporated by the adaptive clustering? The impact of more complex events than those analysed in the current study has at least to be discussed sufficiently in the manuscript, although an analysis of more events is encouraged to represent the effect of the adaptive clustering on the variety of runoff responses.

P15l20 The model states are identified by the slope of the resulting runoff curve. However, the slope can be more or less identical for one time step independent of the
current runoff situation, e.g. if runoff is reduced in one tile from 25mm to 20mm and in another tile from 10mm to 5mm (which could be the case in a stratiform event with a convective cell inside), the resulting slope is the same, right? So the soil moisture and other storage elements is then "averaged" due to the same model state of both tiles, although both tiles are in completely different hydrologic situations. It would be useful if the authors would comment on that issue or, if I understood it not correctly, clarify the part where I got lost.

Specific comments:

P4l5-8 The difference is not clear formulated at this point. It becomes clearer while reading the manuscript, but should be communicated concisely at this point.

P7l27 Where are the disdrometers located? Can they be used to improve the rainfall input for the reference model to achieve a more realistic uniform areal rainfall? If not, could be an increase of rainfall amounts with altitude improve the areal rainfall estimate? The Roodt station is situated in the raster field with the lowest rainfall amounts (Fig 2) and not representable for the catchment. So any correction has to be done to enable a fair comparison between reference model and model a.

Fig2 Please add rain gauge data to Fig2b) to enable a comparison of all rainfall inputs

P10l2, p11l26 area-weighted -> As I understand the areal mean is estimated by the arithmetic mean of the satellite data. How do weights for different areas affect this estimation? This is not clear for me, please rephrase/add the explanation.

P11l2 sap flow -> Do the authors mean by sap flow the flow in plants? I can't imagine at this point how the authors applied observations like that in the current study. If so, please describe a bit more detailed, since it is not a conservative measure for model validation and hence of great interest for the community.

P12l23 "average distance of each grid cell to the outlet" -> Should it not be the distance along the flow path/flow direction? So it would be possible that the runoff is assumed

to stream upwards in some areas of the catchment- Please rephrase or reconsider.

P12l30 "3.2.1 to 3.2.3" -> "3.3.1 to 3.3.3"

P13l2 "wetness state" Please define this term. It sounds as only soil moisture is included without any additional information, but there is more included, right? If not, why not using the term soil moisture? Section 3.3 and 3.3.1 There are repetitions among the paragraphs, please remove them.

P15l4&21 Both thresholds are catchment size-dependent (as the authors state also later). For other applications it would be useful to introduce a catchment size-dependency to derive these thresholds. This is beyond the scope of the study since it demands a multi-catchment analysis, but the authors should add a small sensitivity analysis by e.g. using $\Delta P > \{0.5, 1, 2\}$ mm/hr as thresholds. This is along with a comment I have for the results section later, but I want to state it already here. In the results discussion it is often mentioned, that the number of parameters is reduced between model b and c, there is no figure illustrating it, although I would imagine it would be an impressive plot with y as KGE over x as the summarized number of model parameters per time step (or on average) for one event. Model reference, a, b, c($\Delta P > 1$mm), c($\Delta P > 0.5$mm) and c($\Delta P > 2$mm) would be the points to show in the diagram. I assume model c would represent a break in the curve (KGE not increasing, while number of model parameters do) and the different thresholds would represent the uncertainty of this approach.

Table 1: As fa as I understood the calibration was done only for the reference model, right? Although that seems to be done in a former publication, a brief information about calibration and validation period, objective function and so on is required to interpret the table. For model a, b and c no parameters were changed, so the same parameter set was used throughout the study to enable comparisons? If there was a re-calibration for model c, the reference model and models a and b should be re-calibrated for the events only as well to enable a fair comparison

Fig. 5: I'm a bit confused here. The authors state P=12 for t=2, but from counting it is P=13 – please double-check (also the number of entries in the following text referring to t=2). Additionally, for t=4 M=3 results from P=2 and S=1 – from my understanding the maximum of model states is max(M)=2 in this case, please double-check.

P27l4-22 This paragraph provides already a good overview of related references. However, from my understanding the reference of Nicotina et al. (2008) concluded that spatial patterns of rainfall are only important for large catchments (8000km$^2$ in their study) for hourly time steps, the correct estimation of areal rainfall is sufficient for smaller catchments. The authors should review this reference again and check their implementation in the current manuscript. Also, Ogden and Julien (1993) state that only for rainfall with durations shorter than the concentration time of a catchment the spatial distribution of the rainfall matters, for longer rainfall events only the temporal distribution matters. To highlight the importance of distributed models the authors could also look at Krajewski et al. (1991), Bardossy & Das (2008) or Müller-Thomy et al. (2018).

Technical corrections:

P7l23 aggregated -> transformed

P7l27 merges -> is a merged product of

P10l29 afterward -> afterwards

P12l12 Thee is a 3.2.1, but no 3.2.2.

P15l15 ">1 mm/hr" -> "$\Delta P$ >1 mm/hr" Please add a variable name here. However, "P" is already used in the manuscript for precipitation bins. P is a general abbreviation for precipitation, please consider e.g. "PB" for precipitation bins.

P15l30 (P>0) -> (P>1)

Citation syntax errors: p4l13, p8l2

References: Fenicia et al. (2011a) is identical to Fenicia et al. (2011b), please doublecheck.

References:

Bárdossy, A., Das, T. (2008): Influence of rainfall observation network on model calibration and application, Hydrology and Earth System Science 12, 77–89

Courant, R., Friedrichs, K., Lewy. H. (1928): Über die partiellen Differenzengleichungen der mathematischen Physik, Mathematische Annalen, 100, 32-74 (in German)

Krajewski, W. F., Lakshmi, V., Georgakakos, K. P., Jain, S. C. (1991): A Monte Carlo study of rainfall sampling effect on a distributed catchment model, Water Resources Research 27 (1), 119–128

Melsen, L. A., Teuling, A. J., Torfs, P. J. J. F., Uijlenhoet, R., Mizukami, N., Clark, M. P. (2015): Hydrology and Earth System Science Opinions: The need for process-based evaluation of large-domain hyper-resolution models, Hydrology and Earth System Science 20, 1069–1079

Müller-Thomy, H., Wallner, M., Förster, K. (2018): Rainfall disaggregation for hydrological modeling: Is there a need for spatial consistence?, Hydrology and Earth System Sciences, 22, 5259-5280

Nicotina, L. E., Celegon, E. Alessi, Rinaldo, A., Marani, M. (2008): On the impact of rainfall patterns on the hydrologic response, Water Resources Research 44, W12401

Ogden, F. L., Julien, P. Y. (1993): Runoff sensitivity to temporal and spatial rainfall variability at runoff plane and small basin scales, Water Resources Research 29 (8), 2589–2597
* * *

---

## Referee Comment (RC3) · Wouter Knoben (Referee) · 11 Sep 2020

**Summary**

The authors develop and test a hydrological model that is able to change its spatial complexity in time. In its most simple state, the model represents the Colpach catchment in Luxembourgh as a single representative hillslope. In its most complex state, the model would be able to use 42 hillslope elements to simulate the catchment's response to extremely spatially variable rainfall inputs. The model adds hillslope elements based on the spatial complexity of incoming precipitation and removes hillslope elements based on the change of runoff over time. Both processes use a threshold to decide when upscaling or downscaling the model is needed or possible. The authors

show that the adaptive model reaches the same KGE scores as a fully distributed model that uses 42 hillslope elements all the time, while the adaptive model needs 16 representative hillslopes at most. This is shown for two short-duration event that occurred during summer.

I have read this paper with much interest and found it generally easy to read and understand. As models grow more complex, computation times go up and studies such as this could open up great opportunities to reduce computation costs by avoiding redundancy in model calculations. However, I have some questions about the tests and metric the authors use to show that the adaptive model is as good as the fully distributed one. These are outlined below. I've provided additional requests for clarification in the line-by-line comments in the hopes that these are helpful.

Kind regards,

Wouter Knoben

**Comments**

My main concern is the choice of using dQ/dt to reduce the number of model elements. Using the change in discharge over time to measure similarity of states can only work if there is a unique relationship between model state and dQ/dt. Given the equifinality in the fluxes-discharge relation that's typically visible in hydrological models (see e.g. Khatami et al., 2020), I think the section that introduces this concept (P16, l17) is not quite clear about why this dQ/dt assumption can be used together with CATFLOW.

Reading further, the authors address this concern to some extent in section 4.4 (P23, l18). This section however seems to show that CATFLOW does not exhibit such a unique relationship and the model reduces the number of model elements before the groundwater states reach similarity. This does apparently not affect the quality of the simulations much, because the KGE scores in Table 1 seem to indicate the adaptive model is as good as the fully distributed model for the two testing events.

[Figure]

Fig. 4 shows that both testing events are selected in the middle of summer, when presumably the catchment is in quite a dry state (catchment state is not mentioned when selection of the two events is discussed on P18, l20 to P19, l6). The fact that both events are selected during the dry summer could mean that the model can reset itself to mostly empty between the events and as such the long term (seasonal) impacts of not keeping the groundwater states separate can not be investigated with the current two testing events.

Equally, the events concern high flows so the impact of differences in slow groundwater states probably do not register in the dQ/dt values during the falling limb of the hydrograph (and thus the adaptive model simplifies itself).

There is the compounding issue that the KGE scores used to calculate the performance of model c are only calculated during the high flow event and that metrics such as KGE are typically not very sensitive to errors in low flows. This means that the parts of the simulation time series where the differences in groundwater states could be seen are both not included in calculation of the KGE score of model c and if they were, the KGE metric might not be able to pick up on any differences.

Summarizing the above, I'm not sure that the dQ/dt criterion is entirely appropriate to determine when the adaptive model can reduce its complexity, and I'm equally unsure if the current two testing events would be able to show if the dQ/dt criterion is or is not appropriate. The straightforward solution would be to run model c for the year, add these results to Table 1 and briefly investigate for example the relative contributions of different fluxes to the overall water balance and the model's response to a few precipitation events during winter. Given that the adaptive model should be faster than the fully distributed one, this should not be a large computational burden and it will provide a much more complete impression of the capabilities of the adaptive model.

**Line-by-line**

P5, l5. This question seems quite strongly related to the contrasting results in the

literature that the authors discuss in the first and second paragraph of the introduction, where they conclude that the impact of using a distributed model and/or distributed forcing data is conditional on the catchment under investigation. This research question seems a bit generic in that light, given that only a single model and catchment are being investigated in this work. As is, question 1 seems more like a formality to me (it must be answered with "yes" before Q2 can be investigated) and the main focus of the manuscript seems to be on Q2. Perhaps the manuscript can gain a bit in focus if only the current research question 2 is specified, and the work done to answer the current Q1 is presented as a prerequisite to address the current question 2. For example, "We test this hypothesis by first showing that the model CATFLOW applied to the 19.4 km2 Colpach catchment using a gridded radar-based quantitative rainfall estimate improves in performance when it is distributed in space and driven by distributed rainfall. We then address the following research question: "Can adaptive clustering be used to distribute a bottom-up model in space that it is capable to represent relevant spatial differences in the system state and precipitation forcing at the least sufficient resolution to avoid being highly redundant as a fully distributed model?"

P5, l14. Assuming that "> 1 m" refers to soil depth, should it be "< 1 m"?

P7, l9. Which numerical scheme is used by CATFLOW?

P7, l20. If possible without using too much space, it might be helpful to the reader to briefly summarize the main findings of Loritz et al. (2017).

P7, l21. What are the outcomes of this quality control?

P7, l28. I'm not quite sure I understand why these distances are given as a range if only a single station is concerned. Does this indicate minimum and maximum distance of the catchment bounds to each radar station?

P9, l13. I find this sentence a bit hard to follow. Is the part from "apart from . . ." onwards necessary here? This is already discussed in the introduction.

P10, l21. Why is the model tested during two events instead of over the full year? How were these events selected?

P11, l14. The conclusion that a distributed model is needed to account for runoff driven by convective precipitation would be stronger if the authors can (briefly) list which processes are represented at too coarse a scale in the reference model for it to properly deal with convective precipitation.

P11, l14. I believe this sentence would be more complete if it also explicitly mentioned that distributed instead of catchment-averaged precipitation data is needed to properly simulate the result of convective precipitation events.

P11, l27. It would be helpful for the reader to repeat that the only difference between reference model and model a is the choice of precipitation data.

P12, l3. Are these variables similar or identical to those used in the reference model?

P12, l4. To clarify, does this mean that model b is run in a gridded fashion with the catchment divided into 42 grids (matching the precipitation grid)? If not, it would be good to clarify this in the text and mention the number of model elements that the precip field similarity approach gives. Line 18 on this page could benefit from a similar clarification.

P12, l23. Are there some observations that could help support the choice for 1 m/s?

P14, l29. It might be good to extend this line of reasoning to soil types and vegetation cover, as these are commonly used as model inputs/parameters.

P15, l7. This sentence is quite general (referring to humid environments) and could use a reference. However, if the authors chose 1 mm hr-1 based on their expertise and knowledge about this catchment, then I think it's more accurate (and in no way worse) to phrase this decision along those lines, e.g.: "We chose this threshold as a reasonable value upon which we expect differences in hydrologic behavior, based on our collective understanding of the Colpach catchment."

P17, l10. I think it's import to repeat the similarity condition of dQ/dt here, because for a model that has no unique relation between model state and dQ/dt values this method cannot be applied without accounting for this difference.

P20, l6. The authors use KGE values in this section and Table 1. I'm not sure to what extent the aggregated value is a useful metric for events that last only a handful of time step. It would be good to at least disaggregate the KGE into its correlation, variability and bias components (e.g. quantify what can be qualitatively estimated from Figure 7) to see if the total KGE scores of the individual models are generated by (roughly) the same types of errors in the simulations.

P21, l25. "acceptable" is somewhat subjective because no standard of acceptability has been defined. It might be cleaner to simply report the correlation component of the KGE to quantify to what extent the hydrograph shape is simulated.

P21, l26. This trial of a direct runoff component seems somewhat ad-hoc to me. I don't think this adds anything to the manuscript and that it will take more space than is available to properly justify this change. I suggest to remove these sentences.

P30, l4-24. These sentences seem as if they would be better placed in the introduction or methodology sections.

**Editorial**

P1, l13. "capable to dynamically adjust" > "capable of dynamically adjusting"

P5, l3. "by addressing" > "to address"

P7, l6. "dominated" > "dominant"

P9, l13. "Moreover, are the model . . ." > "Moreover, the model deficits . . . (2017) are . . ."

P12, l29. "model b, however, is" > "model b but is"

P12, l30. "3.2.1 to 3.2.3" > "3.3.1 to 3.3.3"

P13, l10. "inside' > "insight"

P14, l18. "similar" > "similarly"

P15, 5. "structural" > "structurally"

P15, l15. "models" > "model elements"

P21, l24. Figure 7 should be moved and renumbered as Figure 5 if it is mentioned here

P21, l25. "acceptable" > "acceptably"

P27, l21. "perspective, not" > "perspective not"

P29, l22, "there" > "their"

**References**

Khatami, S., Peel, M. C., Peterson, T. J., Western, A. W. (2019). Equifinality and flux mapping: A new approach to model evaluation and process representation under uncertainty. Water Resources Research, 55, 8922–8941. https://doi.org/10.1029/2018WR023750

---

## Referee Comment (RC4) · Anna E. Sikorska-Senoner (Referee) · 16 Sep 2020

This paper proposed an adaptive modelling as an alternative to a distributed model for representing spatial variability of the catchment and forcing input (precipitation). Such an adaptive modelling should be able to run faster than a distributed model but should provide a similar model performance as its fully distributed version. The manuscript is generally well written and it is easy to follow. The idea of a spatially adaptive model that dynamically adjusts its spatial structure during runtime is indeed very interesting and has a potential for being applied in many (hydrologic) modelling approaches. Yet, I have few major issues that should be addressed before considering this manuscript for a publication in HESS. Thus, I recommend a major revision.

[Figure]

**General comments**

1. The adaptive model (model c) is tested here only on two rainfall events, which I see as the major weakness of this manuscript. As the strength of this approach should lie in the possibility to apply it to a continuous modelling and not to an event-based modelling. Thus, I think it would be important to demonstrate how the model c works on continuous time series. As this is missing in the current manuscript, we still do not know at the end whether it is a good or a bad option to be used.

2. The performance metrics of the calibrated (tuned) models should be provided so that the model ability to predict rainfall events could be assessed.

3. A model set-up between the model a and b could be very didactical, i.e. having a structure as the model b but using the precipitation input as the model a (the same for each grid cell).

4. It is not quite clear how the switch between different model setups (i.e. the number of model run in the model c) affect the setup of initial conditions for next runs, which is important to be considered for continuous model simulations but also for simulations of events. More details should be provided on that.

5. It would be also very didactical to see the comparison of the precipitation records from the ground station with the precipitation fields obtained from the gridded data. This is never done in the manuscript and no reason for not doing that is given.

6. The (rather) poor model performance of all tested models' set-ups for two selected events requires some discussion. It appears that none of these model can really capture the dynamics of these two events even if using the distributed

model and the distributed rainfall information (with KGE<0.3). Hence, it is even more important to verify the model performance (pkt. 2). An addition of other metrics that focus entirely on the flood event such as peak or time to peak could be here very informative. Given a rather poor models' performance, it is difficult to justify the need of developing the adaptive model based on the distributed model if the latter does not provide acceptable simulation results.

7. A fair comparison of all presented models should involve the same metrics, i.e. computation over the same time at a continuous time scale. In this study, different model setups are compared at different scales that makes it difficult to get an overview of their performance.

**Detailed comments**

1. Abstract: 'a mesoscale catchment'; a 20-km$^2$ catchment appears rather small to me than meso-scale.

2. Abstract: 'three hydrological models', the model is actually the same but different set-ups are used that span from the averaged model until the distributed model. Please clarify that here.

3. L. 20-21 p. 3: It is not always possible and justified to switch from a continuous model to an event based model. Hence, continuous modelling is often required in many applications.

4. L. 19 p. 4: The tested catchment appears rather small to me. How do you define the cut here for a small/meso-scale catchment?

5. L. 7-9 p. 5: consider restructuring this sentence.

6. L. 11-13 p. 7: could you add the location of these meteorological stations to the map in fig 1?

7. L. 15 p. 7: it should be 'and measures...'

8. L. 16 p.7: is there any weighting applied here and what kind of?

9. L. 28 p. 7: could the locations of radars be also placed in the fig. 1?

10. L. 12-14 p. 8: Why do you compare these values with literature and not with the ground station records from your catchment? Is there any reason that you are not using the precipitation records from the ground station?

11. Fig. 2: One would expect that the radar values would be compared with the ground station values as a corresponding mean (Fig. 2b). Could you add these values to the figure?

12. L. 10-17 p. 9 till 21 p. 10: I am not quite sure if this text is really helpful. After reading these lines, we still do not know how the reference model and other models look like. Maybe you could merge these lines with the sections 3.1-3.3.

13. L. 19-20 p. 10: I would say that the main goal is to test or verify whether similar model performance can be achieved with the adaptive model as compared to the model b. However, by the comparison that you did we still do not know the answer to this question as the comparison is done only based on two pre-selected events both having rather a poor model performance. Please comment on that and also state why these events were chosen for the comparison (and not others)?

14. L. 21-22 p. 10: Why do you compare the adaptive model with model b using only these two events and not the entire simulation period? In my opinion, the greatest potential of the adaptive modelling lies in continuous modelling and not in the event-based.

15. L. 31 p. 10 – l. 1 p. 11: why do you test the model only based on the annual assessment and not on hourly simulations? It is quite surprising because you use the model for assessing the model performance at an event-based scale in the second step, i.e. when comparing different models. I think it is important to report here how the model behaves at an hourly time scale so that one knows what can be expected from the model.

16. L. 3 p. 11: Which metrics were used here for assessing that the model performance agreed well with the dynamics of observed values? can you give some more details on that?

17. L. 7-8 p. 11: It is not surprising that the model performs poorly at time series scale if it was evaluated only on an annual basis. Some insights should be given here; why was the model tested at an annual basis if its intention is to predict events?

18. L. 3 p. 12: similar to what?

19. L. 12. p. 12: the model analysis should go after the introduction of all models.

20. L. 16-19 p. 12: an additional model between the models a and b would be here very useful, i.e. a model that has a structure as the model b but uses precipitation as in the model a (so it uses the same precipitation for each grid cell). This inclusion could nicely show the added value (or no value) of including a spatial distribution of i) the model and ii) of the precipitation input data.

21. L. 8-10 p. 13: why exactly? In my opinion, the strength of this approach lies in the possibility to apply it to a continuous modelling and not to an event-based modelling. Thus, I think it would be nice to demonstrate how the model works on continuous time series in terms of the model performance and computational efforts. As such a test is missing in the current manuscript, we still do not know

at the end whether it is a good or a bad option to use the adaptive modelling approach... Based on the two events selected, we cannot say much about the value of the adaptive approach as the model performance remains poor for these events (as seen from the Table 1 and fig. 7). If the full simulation is not possible (could you give more details why exactly?), already a simple test with shorter but continuous time series of few months or weeks could provide some more insights on how this approach is really working.

22. Tab. 1: as the initial idea is to improve the model performance for rainfall events, it appears from the table that the model c and model b have still rather poor model performance for the event I and II. In addition, all models perform poor for these events. Yet, an inclusion of the spatial variability does not improve much the model performance that is still not so good. Thus, it calls a question of the need of such an adaptive inclusion to this spatially distributed model which performance is rather low. . .. Could you comment on that? A decomposition of KGE into its components would bring more insights on the models' behaviour.

23. L. 9-10 p. 15: How many grid cells need to have a difference higher than this threshold to use the model c?

24. L. 30-31 p. 15 fig.3: is the re-arranged model running with the same initial conditions of the original model or how do you decide on these initial conditions if you want to increase or decrease the number of M in the subsequent time intervals particularly if a continuous simulation is performed?

25. L. 4-6 p. 18: for a fair comparison of different models, you should use the same metrics and the same time periods for evaluation. It is not clear why this is not the case here.

26. Fig. 4: could you add simulations with the model a?

27. L 8. P. 20: The reference Knoben et al. (2019) is missing in the literature list.

28. L. 6-7 p. 21: the performance of KGE below 0 is still rather very poor, which requires some further explanations. According to Knoben et al. (2019), simulations can be considered as behavioural if KGE>0.3 (with KGE $\approx -0.41$ for a mean flow benchmark).

29. Table 1: the performances (KGE) of all models is rather poor for the events here selected (with KGE between -0.41 and 0.29). As already the model b (distributed) cannot simulate the two events in a good way (as also seen from the fig.7), why would you spend time on developing the adaptive model based on the model b instead if improving the model b or testing different models here? Could you comment or justify that?

30. Fig. 7: the simulations with the model a and the reference model should also be added here. Moreover, for both events, all models largely underestimate the events. Could you comment on that?

31. L 10 p. 26 – l. 2 p. 27: do you have any idea where this large underestimation may come from and how it could be improved?

32. Discussion: I missed some recommendations for other works. When and how would such an adaptive modelling be recommended? How one can set up the adaptive process? And why it is really needed to implement such an adaptive modelling?

**Reference**

Knoben, W. J. M., Freer, J. E., and Woods, R. A.: Technical note: Inherent benchmark or not? Comparing Nash–Sutcliffe and Kling–Gupta efficiency scores, Hydrol. Earth Syst. Sci., 23, 4323–4331, https://doi.org/10.5194/hess-23-4323-2019, 2019.

---

## Author Comment (AC1) · 18 Sep 2020

Reply to Referee #1 Daniel Wright:

**Daniel Wright** *(DW): **Summary and Recommendation:** The authors present a framework for dynamically adapting the level of spatial detail re-solved within a physics-based rainfall-runoff model depending on the spatial variability in precipitation. I found this the be one of the most interesting manuscripts that I've ever reviewed, and commend the authors on this innovative work. Nonetheless, there are some issues that should be addressed before the manuscript is suitable for publication in HESS, and that could help maximize the impact of the work.*

**Ralf Loritz (RL):** We would like to thank Daniel Wright for his positive comments and the time he invested to review our Manuscript (MS). The revised MS will follow the reviewer's recommendations and include among other things a re-structured section 3 (model introduction) as well as a more extensive discussion about its connection to the land surface modeling community. Furthermore, will we carefully check the references the reviewer recommended and see whether they help us to improve our argumentation.

Major comments:

*1. **DW**: I believe the discussion could be strengthened by deeper consideration of how this approach would "scale up" to larger watersheds or regions. Part of my reason for encouraging this is that the land surface modeling (LSM) community is at least as concerned as the rainfall-runoff community about model computational demands of long-term/ensemble simulations, and are seeking ways of representing fine-scale (e.g. hillslope and below) over continental-to-global domains. In fact, land surface modeling was the focus of the well-known Wood et al. (2011) hyperresolution modeling opinion piece. In addition, there has been relevant progress in LSM development that the authors should cite. I will mention these below. But in terms of scaling up, the key aspects seem to be acknowledgement that heterogeneity of model parameters will increase with modeled area, while the rainfall spatial coverage will, on average, decrease.*

**RL:** Thank you for raising this point. We mentioned in our MS in the method section page 14 line 28 to 31: "*This entails, however, also that if we extend our research area to a catchment that is divided, for instance, into two geological settings that function hydrologically differently (regarding their filter properties) we would always need to run at least two structural different models where each of these models represents one of two geological settings.*"

We agree with the reviewer that this section is rather short and will provide a more extensive discussion in a revisited MS (see also the following points). We will also carefully read the proposed references by Woods et al. (2011).

**2. DW**: *I believe the discussion could also be strengthened by some discussion of how well this approach might fit with specific types of spatial discretizations. It fits quite naturally with hillslope-based models. The fit is less clear with gridded or TIN-based models-or at least with high-resolution gridded models in which individual model grids must "communicate" with each other to transmit water via overland or subsurface flow to channels.*

**RL:** This is an important point. Indeed our approach is limited to hydrological models that are based on a division of the landscape into partly independent spatial units (similar to the work of *Chaney et al. 2016*). However, at least in theory, there is no limit on how complex the interaction between these independent sub-units are as long as there is redundancy/similarity when different model elements "*communicate*" with each other. However, the question at what point of model complexity we would still save computational times by reducing redundancy depends on a series of factors (e.g. model, resolution, no. of processes and state variables). We will discuss this in a revisited MS.

**DW**: *It seems that the computational advantages of the approach might be limited in that case. In addition, models such as GSSHA in which overbank river flow can return to the land surface would have some limits here too. These issues are worth discussing because such models constitute important current directions in physics-based model development.*

**RL:** From our perspective, it makes much sense to divide a landscape into different building blocks such as hillslopes, sub-basins, etc. (*e.g. Zehe et al., 2014*). This is the case as current physically-based models are still constrained to small areas if they are set up on an appropriate grid size. We see hence no way around dividing a landscape into some kind of independent sub-units and either run models in parallel or/and group similar model elements (dynamically or time-invariant) if we want to work on larger scales. That said, we also believe that it would be rather difficult to implement a spatial adaptive modeling approach in a current model like Delft2d or GSSHA.

We wrote in our MS (Pg. 16 line 10): "*While we use CATFLOW as a model here, the proposed approach is not restricted to this model and can be used in any hydrological model that distributes a catchment into independent spatial units.*". To underpin this point we will discuss the limitation of our approach in more detail (please also see the discussion with the second referee and the third). Again we thank DW for this valuable comment.

**3. DW**: *While there may be other relevant LSM developments, the one that I am aware of is Hydroblocks (Chaney et al. 2016). While I recommend reading that paper, the basic approach is similar to this manuscript's in that spatial units are grouped into hydrologically similar clusters to reduce the computational demand.*

**RL:** Thank you very much for pointing us to the study of *Chaney et al. (2016)*. We will examine it carefully.

**DW**: *The difference is that in Hydroblocks, these clusters are not dynamically reassigned according to time-varying characteristics (unless the developers have recently added that capability). So in fact, your approach appears to be superior in some respects. Specifically, within Hydroblocks, since there is no dynamic reassignment, you can never have a cluster that extends beyond the spatial extent of a single precipitation grid cell, which means that their approach loses computational efficiency with higher-resolution precipitation datasets. Your approach thus seems to hold more promise in terms of flexibility to advances in precipitation inputs.*

**RL:** Interesting comment. We would like to highlight that we only showed that our approach is theoretical and practically feasible. It remains an open question if we could actually save more computational times in comparison to HydroBlocks or similar time-invariant approaches. The question is open for discussion as our approach is also more complicated. We will discuss this in a revised discussion section.

**4. DW:** *More clear description of what each model does and does not do is needed in Section 3. Specifically, I found it confusing the way that the models are briefly introduced at the beginning of the section, and then discussed further in various subsections. I also find it strange that you have text that is not assigned to specific subsections. It isn't clear why section 3.2.1 is needed...convention is that you don't include subsections unless you have at least 2 or 3 (i.e. 3.2.2, 3.2.3). This section structuring needs rethinking. [...] Also, a table that compares the key features and differences of all the models could be effective. I think one think that would really help is to not use "model a", "model b", etc. but some brief descriptive names that actually help the reader understand and recall the differences. More important, I really couldn't figure out from the descriptions what the differences between some models were. I also don't understand the motivation for using a different rainfall dataset for the reference model and model a; this seems unnecessary.*

**RL:** We stated in our MS on page 11 line 26-29: "*We added model a to test if the performance difference between the reference model and our distributed model b is merely a result of quantitative differences between the different precipitation products measured either by a single ground station or by a weather radar.*"

However, we understand the comment of Daniel Wright and we will restructure section 3 entirely and remove the short introduction of the different models. We will also follow your advice and add a table with the key features of each model.

**5. DW:** *Zhu et al. (2018) and Peleg et al. (2017) both highlight how distributed rainfall structure is really important in determining flood frequency across a range of scales. Though I normally refrain from suggesting that authors cite my own work, in this case it seems appropriate to highlight these studies, since they do show that for extreme events, rainfall space-time structure is extremely important in determining hydrologic response even at very small scales (see Peleg et al. in particular), and that this importance varies with rainfall magnitude and basin size.*

**RL:** We have carefully read both publications and they fit nicely into our revisited discussion. Thank you for pointing us towards these two references.

**DW:** *Along with this, I disagree with the statement on pg. 27: "it seems that catchment size might not be the best indicator to decide if" a distributed model is needed. It probably is the best single indicator, but is still insufficient. I draw a somewhat different conclusion from your work: that a distributed approach is always needed to reap the full benefit of spatially distributed rainfall (at least in locations in which convective rainfall can occur), and that provides motivation for continued developments such as this into ways of handling this need in computationally-efficient ways.*

**RL:** What we wanted to convey here is that it is the combination of the drainage area and the average size of a typical rainstorm, which is important and not the drainage area alone. For instance, if you wanted to predict the runoff formation in the Colpach catchment in the winter season a spatially aggregated model driven by a single precipitation time series might be sufficient as our results show. This means also that you could invest your limited time and improve for instance the groundwater representation in your model instead of setting up a distributed model. However, if you wanted to make predictions in the summer months our results highlight that you need some sort of a distributed model to be able to capture the spatial variability of the rainfall. This means that only because the Colpach is 20 km$^2$ we cannot decide if we need a spatially distributed model as the catchment size does not explain how variable its meteorological forcing is. *Nicotina et al. (2008)* argued along these lines and stated that the "*total residence time of a water parcel is often controlled by the travel time within hillslopes, we find that when typical hillslope size is smaller than the characteristic size of rainfall structures (say, a correlation length of rainfall intensity), the rainfall pattern effectively samples all possible residence times and the response of the catchment does not depend on the specific rainfall pattern.*" and the second referee pointed us towards the study of *Ogden and Julien (1993)*. The second reviewer also nicely summarized their key finding: *"only for rainfall with durations shorter than the concentration time of a catchment does the spatial distribution of the rainfall matters, for longer rainfall events only the temporal distribution matters.".* Following these two studies and our own results we would argue that our first research question in our MS: "How important are spatial patterns of precipitation for the runoff generation at the catchment scale?" can only be answered if we combine information about the catchment size (e.g. average hillslope length, concentration time) with

information about the meteorological forcing (e.g. intensity, correlation length, velocity). In a revisited MS we will rephrase this paragraph and explain in more detail what we meant by this statement.

*DW: Likewise, I disagree with the statement on pg. 30 line 18-19: compressing rainfall into a single time series isn't so important as the ability to only use as much computational power as is truly needed to solve the problem at hand.*

**RL:** Again an interesting point you raise here. In our specific setting, compression of precipitation and saving computational power are the same. By compressing the precipitation field to a single time series we also compress our model, minimize redundant calculations, which again means that we save computational power. So, we would argue that we first need to understand (test) how far we can compress our rainfall field without losing predictive performance before we can save computational times in a meaningful manner. The data-based / machine learning community most likely would disagree ☺

**6. DW:** *Some discussion of implications for calibration would be interesting. Is it necessary to calibrate using a fully distributed model? This would limit the usefulness of this approach in some respects such as automated calibration procedures.*

**RL:** Typically, one run of the reference model (a single CATLOW hillslope) for a simulation period of one year and hourly printout times takes about 2 - 3 hrs. Assuming that you run your code on a workstation with 32 cores you can run about 400 model setups in 24 hrs. As structurally similar areas are represented by the same model in our approach, testing different model parameters sets should be feasible even in larger areas if the structural properties are not too variable/complex. We will discuss this in a revisited MS.

**7. DW:** *There are a number of minor grammatical issues that nonetheless cause some distraction from the overall high quality of the manuscript. I will point out some of these below, but it could be worthwhile to have a native English speaker perform a careful proofreading.*

**RL:** We will carefully proofread the MS once more and would like to highlight that there will be another professional language check by Copernicus if the MS is accepted for publication.

References:

Nicótina, L., Alessi Celegon, E., Rinaldo, A. and Marani, M.: On the impact of rainfall patterns on the hydrologic response, Water Resour. Res., 44(12), 1–14, doi:10.1029/2007WR006654, 2008.

Ogden, F. L. and Julien, P. Y.: Runoff sensitivity to temporal and spatial rainfall variability at runoff plane and small basin scales, Water Resour. Res., 29(8), 2589–2597, doi:10.1029/93WR00924, 1993.

Zehe, E., Ehret, U., Pfister, L., Blume, T., Schröder, B., Westhoff, M., Jackisch, C., Schymanski, S. J., Weiler, M., Schulz, K., Allroggen, N., Tronicke, J., van Schaik, L., Dietrich, P., Scherer, U., Eccard, J., Wulfmeyer, V. and Kleidon, A.: HESS Opinions: From response units to functional units: a thermodynamic reinterpretation of the HRU concept to link spatial organization and functioning of intermediate scale catchments, Hydrol. Earth Syst. Sci., 18(11), 4635–4655, doi:10.5194/hess-18-4635-2014, 2014.

---

## Author Comment (AC2) · 18 Sep 2020

Reply to Anonymous Referee #2:

**Anonymous Referee #2** *(AR2): Summary and Recommendation: The manuscript introduces an adaptive spatial clustering of hydrologic response units (HRU) to cope with the dynamics of the intermittent rainfall by keeping the model as parameter parsimonious (=model states) as possible in terms of reduction of similar-reacting HRUs. The manuscript is well-written and I enjoyed reading it. The introduced clustering is innovative from and fits into the scope of the journal. I have a few moderate and a number of minor comments, which are stated below. My overall recommendation would be a moderate revision to give the authors enough time to solve the open issues. Since I can only choose between minor and major revision, major revision it is.*

**Ralf Loritz (RL):** We would like to thank the second referee for the time and the effort she/he put into his review. The points she/he raises are relevant and addressing them will help to improve our manuscript. We hope that after this discussion (as well as after we revised our manuscript) all issues she/he raises can be clarified.

Moderate comments:

**1. AR2**: *The manuscript is about the reduction of the spatial model resolution based on the variety of precipitation as input signal. I'm wondering if there is not an adaption of the temporal resolution required as well since scales in space and time are not independent of each other (see Melsen et al. (2015) and references therein)? Maybe it's not an issue for the small catchment studied here ...*

**RL:** Important comment. The results of the study of *Zhu et al. (2018;* recommended by the first reviewer) highlight that the timing of the precipitation is more important in smaller catchments while it is the spatial pattern in larger catchments. We will discuss this in a revisited MS and carefully read the study of *Melsen et al. (2015).*

**AR2**: *... but for larger catchments with a small hydrologic variability the numeric stability can be questioned due to the large spatial discretization and the high temporal resolution (e.g. in terms of the Courant-Friedrichs-Lewy condition, Courant et al., 1928). The authors should proof this condition for their model setup and discuss possible issues in the manuscript. An alternative would be to reduce the temporal resolution as well, which would lead to an additional reduction of parameters/computational costs.*

**RL:** CATFLOW uses an adaptive time-stepping, which means that time steps can be reduced down to seconds depending on the numerical solver. In the presented study the Darcy Richards equation is solved implicitly while the surface runoff is solved explicitly (for more details see also *Zehe et al., 2001*). As the horizontal grid resolution of the CATFLOW hillslope (reference model) is below 1 m, the vertical below 10 cm and time steps are small we have no issue to fulfill the Courant criteria in our model.

Nevertheless, you mention an important point here and the fulfillment of physical and numerical constraints were the main motivations of our former "representative hillslope" study (*Loritz et al. 2017*).

CATFLOW hillslopes are typically interconnected by a river network and runoff is routed downstream with a diffusion wave approach (explicitly solved) assuming a prismatic river cross-section and roughness that changes with changing Strahler order. However, the combination of a river network with the raster layout of our adaptive model is not straightforward (although not impossible, for instance by linking the centroid of a raster cell to the closest node of the river network). To make things not more complicated as necessary we decided to use a lag function in this study. This lag function is not solved numerically but shifts the simulated hydrographs in time by a constant velocity. Again we have no issue with the Courant criteria here. The latter is different if we would have used an adaptive mesh approach where the numerical grid is changed during runtime. Here we carefully need to check the courant criteria when we increase the size of the grids. We will discuss this in a revisited MS.

**2. AR2**: *The authors have selected two events to show the ability of adaptive clustering. The choice of both events seems to be very arbitrary. From Fig. 4 it seems that the resulting runoff peaks are not representative for runoff mechanisms of the catchment. As far as I understand from P13 l8-10 the clustering is carried out manually and not automatically so far, which is the reason why the authors decided for two small events covering only a few time steps. However, I disagree with the hypothesis that "a test on a longer timescale...would provide only little more scientific inside" (P13 l9-10), which is also not proven by the authors.*

**RL:** Respectfully, we do not agree with the assessment of the reviewer regarding the selection of our rainfall events. We chose event I as it has the highest intensity and third-highest spatial variability in the chosen period. We chose event II because we wanted to test our adaptive model at a rainfall event with a longer duration and lower intensity). From examining the rainfall-runoff events in summer we believe that both events represent the runoff generation in summer well as long as subsurface storm flow is dominant. We agree with the reviewer that our statement is a bit misleading and we will explain better what we mean here. Please see also the discussion with the third and fourth reviewers.

*AR2*: *I rather expect that the reduction of model parameters due to the adaptive spatial resolution is reduced significantly for long-lasting rainfall events causing a direct runoff response over several days as e.g. in Nov 2014, Jan 2014-Mar2014 and Aug 2014.*

**RL:** You are correct. The needed spatial model resolution in winter is much lower compared to the chosen summer rainfall-runoff events. This is indicated by the fact that the distributed *model b* and the *reference model* perform almost identical with respect to simulate the observed discharge in the winter season. We would argue that is is difficult to justify the use of a spatially distributed model over a

spatially aggregated model if they perform similarly as long as the focus is on an integral response of a system.

*AR2*: *Another point that can be questioned is snow, which does not cause runoff immediately, but when snow melt begins. How will this be affected/can be incorporated by the adaptive clustering? The impact of more complex events than those analysed in the current study has at least to be discussed sufficiently in the manuscript, although an analysis of more events is encouraged to represent the effect of the adaptive clustering on the variety of runoff responses.*

**RL:** Interesting point. Snow is rare in the Colpach catchment which is fortunate as CATFLOW has no internal snow routine. However, let's assume we would have used a model with a snow routine in an area where snow is a dominant control on the runoff generation. In this specific scenario, we would indeed have to adapt our definition of similarity between the model states. In other words, instead of using the only slope of the simulated hydrograph alone to define similarity, we would also need to check the snow cover before we would group models as functional similar based on their state. Two similar hillslopes would then have the "same" snow cover (given a threshold) as well as the same slope of the hydrograph. We very much like the idea of testing the approach in an area where snow is an important factor. However, for now, we will discuss the limits of choosing a single variable to group model states in a revisited MS.

*3. AR2*: *The model states are identified by the slope of the resulting runoff curve. However, the slope can be more or less identical for one time step independent of the current runoff situation, e.g. if runoff is reduced in one tile from 25mm to 20mm and in another tile from 10mm to 5mm (which could be the case in a stratiform event with a convective cell inside), the resulting slope is the same, right? So the soil moisture and other storage elements is then "averaged" due to the same model state of both tiles, although both tiles are in completely different hydrologic situations. It would be useful if the authors would comment on that issue or, if I understood it not correctly, clarify the part where I got lost.*

**RL:** You are correct. In an earlier version of our spatial adaptive model, we used the absolute discharge to identify similar model states. The issue here was that two models could produce the same discharge at a given time step but one model would simulate a rising hydrograph while the other a declining. We hence decided to take the slope of the hydrograph assuming that the model differences would be small given the size of the Colpach catchment, the focus on the summer season and because we only simulate shallow subsurface stormflow. In the case of a stratiform event with a convective cell inside or if we have snow in a catchment our assumption might be violated. Thank you for raising this issue and along your lines, we will add another criterion to our spatially adaptive model. In a revisited MS only model elements which share a similar $dQ \, dt^{-1}$ (0.05 mm $hr^{-1}$) as well as Q (0.05 mm $hr^{-1}$) will be grouped together.

Specific comments:

*AR2*: *P4 l5-8 The difference is not clear formulated at this point. It becomes clearer while reading the manuscript, but should be communicated concisely at this point.*

**RL:** We will rephrase this sentence.

*AR2*: *P7 l27 Where are the disdrometers located? Can they be used to improve the rainfall input for the reference model to achieve a more realistic uniform areal rainfall? If not, could be an increase of rainfall amounts with altitude improve the areal rainfall estimate? The Roodt station is situated in the raster field with the lowest rainfall amounts (Fig 2) and not representable for the catchment. So any correction has to be done to enable a fair comparison between reference model and model a.*

**RL:** When we were setting up the reference model for our proceeding study (*Loritz et al., 2017*) the only rainfall measurement available at that time was the ground station in "Roodt". As we are aware that the comparison between the *reference model* and *model b* (the distributed model) is not entirely fair as they used different rainfall data we added *model a* to the model ensemble. In a restructure section 3 we will clarify this as well as add the location of the distrometers to the appendix A1.

*AR2*: *Fig 2 Please add rain gauge data to Fig 2b) to enable a comparison of all rainfall inputs.*

**RL:** We will add the rainfall data from "Roodt" to Fig2b.

*AR2*: *P10 l2, p11 l26 area-weighted -> As I understand the areal mean is estimated by the arithmetic mean of the satellite data. How do weights for different areas affect this estimation? This is not clear for me, please rephrase/add the explanation.*

**RL:** As not all of the 42 raster cells of the distributed rainfall data are entirely within the borders of the Colpach catchment their weight was reduced when we calculated the average precipitation for *model a*. We will rephrase this sentence accordingly.

*AR2*: *P11 l2 sap flow -> Do the authors mean by sap flow the flow in plants? I can't imagine at this point how the authors applied observations like that in the current study. If so, please describe a bit more detailed, since it is not a conservative measure for model validation and hence of great interest for the community.*

**RL:** By sap flow we indeed mean sap flow in plants. In our proceeding study (*Loritz et al. 2017*) we compared normalized sap flow velocities against normalized transpiration simulations of CATFLOW to evaluate the transpiration simulation. Sap flow measurements where thereby one of the keys for a successful simulation in the Colpach catchment as they helped us to identify the onset of the vegetation (when the trees started to transpire). The comparison is described in detail in *Loritz et al. (2017*; Figure

12). Although we agree that this might be of great interest for the community we would like to avoid discussing this once more to keep the MS as focused as possible.

*AR2: P12 l23 "average distance of each grid cell to the outlet" -> Should it not be the distance along the flow path/flow direction? So it would be possible that the runoff is assumed to stream upwards in some areas of the catchment- Please rephrase or reconsider.*

**RL:** For each grid cell of the precipitation field we calculated the average flow length along the surface topography to the outlet of the catchment using an underlying DEM with a 10 m resolution. We used the averaged flow distances in our lag function. We will explain this in more detail in a revised MS.

*AR2: P12 l30 "3.2.1 to 3.2.3" -> "3.3.1 to 3.3.3"*

**RL:** Following the discussion with the first reviewer Daniel Wright we will restructure section 3 and remove all "subsubsections".

*AR2: P13 l2 "wetness state" Please define this term. It sounds as only soil moisture is included without any additional information, but there is more included, right? If not, why not using the term soil moisture? Section 3.3 and 3.3.1 There are repetitions among the paragraphs, please remove them.*

**RL:** We will remove the term "wetness state" with the term "catchment state" to make clear that we also mean the shallow groundwater table, soil moisture, etc. We will furthermore restructure section 3 and remove the repetitions.

*AR2: P15 l4 & 21 Both thresholds are catchment size-dependent (as the authors state also later).For other applications it would be useful to introduce a catchment size-dependency to derive these thresholds. This is beyond the scope of the study since it demands a multi-catchment analysis, but the authors should add a small sensitivity analysis by e.g. using ΔP >{0.5, 1, 2} mm/hr as thresholds. This is along with a comment I have for the results section later, but I want to state it already here. In the results discussion it is often mentioned, that the number of parameters is reduced between model b and c, there is no figure illustrating it, although I would imagine it would bean impressive plot with y as KGE over x as the summarized number of model parameters per time step (or on average) for one event. Model reference, a, b, c (ΔP>1mm), c(ΔP>0.5mm) and c(ΔP>2mm) would be the points to show in the diagram. I assume model c would represent a break in the curve (KGE not increasing, while number of model parameters do) and the different thresholds would represent the uncertainty of this approach.*

**RL:** Interesting comment. Using a typically physically-based model (CATFLOW) and specifically the setup of our model based on field measurements it is kind of difficult to estimate the number of model parameters in our study. However, we will add a plot with the distributed precipitation binned into different thresholds (0.1, 0.5, 1, 2, 5 mm hr$^{-1}$) to the appendix. Based on this plot we will discuss how the binning will most likely affect our spatial adaptive model. Furthermore, will we discuss that a

sensitivity analysis with different thresholds is needed along your line of arguments. Again thank you for this comment.

*AR2*: *Table 1: As far as I understood the calibration was done only for the reference model, right? Although that seems to be done in a former publication, a brief information about calibration and validation period, objective function and so on is required to interpret the table. For model a, b and c no parameters were changed, so the same parameter set was used throughout the study to enable comparisons? If there was a re-calibration for model c, the reference model and models a and b should be re-calibrated for the events only as well to enable a fair comparison*

**RL:** Exactly the calibration was done in a former publication exclusively for the reference model. All model parameters remain the same. The only differences between the models are the rainfall data which we use to drive the models as well as their spatial resolution. We will add more details about the calibration in a restructured section 3.

*AR2*: *Fig. 5: I'm a bit confused here. The authors state P=12 for t=2, but from counting it is P=13 – please double-check (also the number of entries in the following text refer ring to t=2). Additionally, for t=4 M=3 results from P=2 and S=1 – from my understanding the maximum of model states is max(M)=2 in this case, please double-check.*

**RL:** You are correct. We will check the figure as well as the corresponding text passages. Thank you for checking the figures so carefully.

*AR2*: *P27 l4-22 This paragraph provides already a good overview of related references. However, from my understanding the reference of Nicotina et al. (2008) concluded that spatial patterns of rainfall are only important for large catchments (8000km2 in their study) for hourly time steps, the correct estimation of areal rainfall is sufficient for smaller catchments. The authors should review this reference again and check their implementation in the current manuscript.*

**RL:** Thank you for this comment. We were referring to the following section in *Nicotina et al. (2008)*: "*As noted in section 4, this is because the spatial scales of variability of rainfall are very often much larger than the typical hillslope scale. Whenever infiltration excess mechanisms are important, the spatial distribution of areas of intense rainfall may be an important factor in determining the hydrologic response, ...* ". In the current MS the use of this reference is indeed misleading and a leftover from an earlier version. We will revisit the corresponding sentence.

*AR2*: *Also, Ogden and Julien (1993) state that only for rainfall with durations shorter than the concentration time of a catchment the spatial distribution of the rainfall matters, for longer rainfall events only the temporal distribution matters. To highlight the importance of distributed models the authors could also look at Krajewski et al. (1991), Bardossy & Das (2008) or Müller-Thomy et al. (2018)*

**RL:** Thank you very much for pointing us to these publications we will read them carefully and see if they can help us to improve our argumentation.

---

## Author Comment (AC3) · 18 Sep 2020

Reply to Referee #3 Wouter Knoben:

**Wouter Knoben** *(WK): Summary and Recommendation: The authors develop and test a hydrological model that is able to change its spatial complexity in time. In its most simple state, the model represents the Colpach catchment in Luxembourg as a single representative hillslope. In its most complex state, the model would be able to use 42 hillslope elements to simulate the catchment's response to extremely spatially variable rainfall inputs. The model adds hillslope elements based on the spatial complexity of incoming precipitation and removes hillslope elements based on the change of runoff over time. Both processes use a threshold to decide when upscaling or downscaling the model is needed or possible. The authors show that the adaptive model reaches the same KGE scores as a fully distributed model that uses 42 hillslope elements all the time, while the adaptive model needs 16 representative hillslopes at most. This is shown for two short-duration event that occurred during summer.*

*I have read this paper with much interest and found it generally easy to read and understand. As models grow more complex, computation times go up and studies such as this could open up great opportunities to reduce computation costs by avoiding redundancy in model calculations. However, I have some questions about the tests and metric the authors use to show that the adaptive model is as good as the fully distributed one. These are outlined below. I've provided additional requests for clarification in the line-by-line comments in the hopes that these are helpful.*

**Ralf Loritz (RL):** We would like to thank Wouter Knoben for the interesting discussion on our Manuscript (MS). WK raises a couple of important and well-thought comments and we hope that after this discussion as well as after we have revisited our MS all open issues can be clarified.

Comments:

*1. WK*: *My main concern is the choice of using dQ/dt to reduce the number of model elements. Using the change in discharge over time to measure similarity of states can only work if there is a unique relationship between model state and dQ/dt. Given the equifinality in the fluxes-discharge relation that's typically visible in hydrological models (see e.g. Khatami et al., 2020), I think the section that introduces this concept (P16, l17) is not quite clear about why this dQ/dt assumption can be used together with CATFLOW.*

**RL:** Important comment and also the second reviewer had similar concerns. We will hence add Q as a second variable to group and ungroup model states and improve the discussion on how using a single variable to define similarity between model states will always lead to errors in certain scenarios and following that these variables need to be picked carefully.

*WK*: *Reading further, the authors address this concern to some extent in section 4.4 (P23, l18). This section however seems to show that CATFLOW does not exhibit such a unique relationship and the model reduces the number of model elements before the groundwater states reach similarity. This does*

*apparently not affect the quality of the simulations much, because the KGE scores in Table 1 seem to indicate the adaptive model is as good as the fully distributed model for the two testing events.*

**RL:** In the current MS we did not mention that CATFLOW simulates only shallow subsurface stormflow in the entire summer period. This means that we do have a rather unique relationship between our model states and their function. Furthermore, the example in Fig.8 shows two extreme cases where one model receives much more precipitation than the other exactly intending to show the limits of our approach (section 5.3). As discussed in more detail with the fourth reviewer we will show that there is no difference between *model a* and *model c* (also concerning soil moisture in both depths) already shortly after the rainfall stops and when *model c* represents the entire catchment with a single hillslope. Furthermore, by calculating the Shannon entropy of the 42 hydrographs simulated by the spatially distributed *model b* we can see that there is no reason to assume that two models drift apart in the selected time frame. We will disscuss this in a revisted MS.

*WK*: *Fig. 4 shows that both testing events are selected in the middle of summer, when presumably the catchment is in quite a dry state (catchment state is not mentioned when selection of the two events is discussed on P18, l20 to P19, l6).*

**RL:** We mention the catchment states for event I and II on page 18 line 26 to 28 and page 19 line 5 to 6. Furthermore, do we refer to our former study where we showed 38 time series of soil moisture in the Colpach catchment in various depths and locations for the same hydrological year.

*WK*: *The fact that both events are selected during the dry summer could mean that the model can reset itself to mostly empty between the events and as such the long term (seasonal) impacts of not keeping the groundwater states separate cannot be investigated with the current two testing events.*

*Equally, the events concern high flows so the impact of differences in slow ground-water states probably do not register in the dQ/dt values during the falling limb of the hydrograph (and thus the adaptive model simplifies itself).*

*There is the compounding issue that the KGE scores used to calculate the performance of model c are only calculated during the high flow event and that metrics such as KGE are typically not very sensitive to errors in low flows. This means that the parts of the simulation time series where the differences in groundwater states could be seen are both not included in calculation of the KGE score of model c and if they were, the KGE metric might not be able to pick up on any differences.*

**RL:** Again an important point. As already mentioned above we discussed in section 5.3. "*While this finding is surely constrained by the chosen threshold, the picture is nevertheless quite different in deeper soil layers where the diversity of the rainfall forcing leads even after 24 hrs to increasing differences between the "driest" and "wettest" models. A part of the information about the different meteorological forcings between the two models is hence still stored in the model state after 24*

*hrs and has not yet been dissipated. The importance of those differences likely depends on the dominant runoff generation process. In the present case, they have a minor impact as model ...*"

In a revisited MS we will underpin once more that our approach with the current definition of similarity regarding the model states can have significant impacts for long-term simulations however that there is no reason to expect that in our specific case.

**WK**: *Summarizing the above, I'm not sure that the dQ/dt criterion is entirely appropriate to determine when the adaptive model can reduce its complexity, and I'm equally unsure if the current two testing events would be able to show if the dQ/dt criterion is or is not appropriate. The straightforward solution would be to run model c for the year, add these results to Table 1 and briefly investigate for example the relative contributions of different fluxes to the overall water balance and the model's response to a few precipitation events during winter. Given that the adaptive model should be faster than the fully distributed one, this should not be a large computational burden and it will provide a much more complete impression of the capabilities of the adaptive model.*

**RL:** We hope that an improved discussion of the limits of the dQ/dt criterion (or any other criterion) as well as the addition of a second similarity measure (Q) will clarify the issues WK raises. We would also like to stress once more that we focus exclusively on the summer season as the distributed *model b* outperforms the *reference model* only in this period and because the meteorological boundary conditions change between the winter (frontal) and summer seasons (convective; Fig. 4 b). Furthermore, did we chose two rainfall-runoff events instead of the entire period as it allows us to analyzes the events in great detail (Event I: Fig. 5, 6, 7) and as our main focus in this study is on the rainfall-runoff interaction and not on low flow conditions. We selected event I because it has the highest rainfall intensity and third-highest spatial variability (highest Shannon entropy) in the selected period and event II because we wanted to test our spatial adaptive model at a summer rainfall event with longer duration. We believe that both events represent the state space of the runoff formation of the Colpach in summer well and see no reason to assume that the *model c* would fail at another rainfall event. A test of the spatial adaptive model for the entire hydrological year (or even for a longer period), in a different environment, with more variables and different thresholds to group and ungroup the model states and maybe even with a different type of model, is indeed desirable. However, to keep the already quite elaborated MS as focused as possible we will focus on improving the discussion with respect to the limits of our approach, add Q as second criteria to define similar model states, add a new figure to the appendix where we show how the thresholds impact the number of precipitation groups (please see the discussion with reviewer 2) and finally plot the soil moisture of *model a* and *c* at the end of event I and II to highlight that both models are in a similar state also with respect to their soil moisture.

Line-by-line comments

*WK: P5, l5. This question seems quite strongly related to the contrasting results in the literature that the authors discuss in the first and second paragraph of the introduction, where they conclude that the impact of using a distributed model and/or distributed forcing data is conditional on the catchment under investigation. This research question seems a bit generic in that light, given that only a single model and catchment are being investigated in this work. As is, question 1 seems more like a formality to me (it must be answered with "yes" before Q2 can be investigated) and the main focus of the manuscript seems to be on Q2. Perhaps the manuscript can gain a bit in focus if only the current research question 2 is specified, and the work done to answer the current Q1 is presented as a prerequisite to address the current question 2. For example, "We test this hypothesis by first showing that the model CATFLOW applied to the 19.4 km2Colpach catchment using a gridded radar-based quantitative rainfall estimate improves in performance when it is distributed in space and driven by distributed rainfall. We then address the following research question: "Can adaptive clustering be used to distribute a bottom-up model in space that it is capable to represent relevant spatial differences in the system state and precipitation forcing at the least sufficient resolution to avoid being highly redundant as a fully distributed model?"*

**RL:** Good idea. We will consider rephrasing this section following your lines.

*WK: P5, l14. Assuming that "> 1 m" refers to soil depth, should it be "< 1 m"?*

**RL:** No, soils are rather deep in this area and vary between 1 to 2.7 m according to several drillings and electrical resistivity tomography (ERT) measurements.

*WK: P7, l9. Which numerical scheme is used by CATFLOW?*

**RL:** Darcy-Richards: implicitly solved by a mass conservative modified Picard iteration scheme (Celia et al. 1990); Surface runoff (1d St. Verdant eq.) explicit Euler forward. We will add this information to the MS.

*WK: P7, l20. If possible without using too much space, it might be helpful to the reader to briefly summarize the main findings of Loritz et al. (2017).*

**RL:** The main findings of this study are summarized on page 10 section 3.1.

*WK: P7, l21. What are the outcomes of this quality control?*

**RL:** Manually quality checked by the Luxembourg, Institute of Science and Technology (LIST; no negative values, etc). We will remove the term "quality checked" as it is not necessary here.

*WK: P7, l28. I'm not quite sure I understand why these distances are given as a range if only a single station is concerned. Does this indicate minimum and maximum distance of the catchment bounds to each radar station.*

**RL:** Exactly. These are the distance to the boundaries of the Attert catchment in which the Colpach is located. We will add this information in a revisited MS.

*WK: P9, l13. I find this sentence a bit hard to follow. Is the part from "apart from..." onwards necessary here? This is already discussed in the introduction.*

**RL:** We will remove this part.

*WK: P10, l21. Why is the model tested during two events instead of over the full year? How were these events selected?*

**RL:** Please see the discussion above and the discussion with the second and fourth reviewer.

*WK: P11, l14. The conclusion that a distributed model is needed to account for runoff driven by convective precipitation would be stronger if the authors can (briefly) list which processes are represented at too coarse a scale in the reference model for it to properly deal with convective precipitation.*

*WK: P11, l14. I believe this sentence would be more complete if it also explicitly mentioned that distributed instead of catchment-averaged precipitation data is needed to properly simulate the result of convective precipitation events.*

**RL:** We wrote on page 11 line 14: *"In other words, this entails that a hydrological model, distributed at a sufficiently high spatial resolution, is required to capture the spatial variability of the precipitation field to satisfactorily simulate the runoff generation of the Colpach"*. We believe that our argumentation is well justified here.

*WK: P11, l27. It would be helpful for the reader to repeat that the only difference between reference model and model a is the choice of precipitation data.*

**RL:** We wrote in the sentence before the sentence you mention: *"Model a is identical to the reference model, however, driven by the area-weighted mean of the spatially resolved precipitation data described in section 2.4 (Fig. 2 b)."*

*WK: P12, l3. Are these variables similar or identical to those used in the reference model?*

**RL:** Identical. We will change the word accordingly.

*WK: P12, l4. To clarify, does this mean that model b is run in a gridded fashion with the catchment divided into 42 grids (matching the precipitation grid)? If not, it would be good to clarify this in the text and mention the number of model elements that the precip field similarity approach gives. Line 18 on this page could benefit from a similar clarification.*

**RL:** Yes, this means that model b is "*divided into 42 grids (matching the precipitation grid)".* We will consider rephrasing the corresponding sentences.

*WK: P12, l23. Are there some observations that could help support the choice for 1 m/s?*

**RL:** We will add the reference of *Leopold, (1953)*. Fig. 1 in this reference shows an average relation of flow velocities and discharge in rivers. Correspondingly we picked an average value of 1 m s$^{-1}$ (2 to 3 feet per second).

*WK: P14, l29. It might be good to extend this line of reasoning to soil types and vegetation cover, as these are commonly used as model inputs/parameters.*

**RL:** Agreed. We will rephrase the corresponding sentence.

*WK: P15, l7. This sentence is quite general (referring to humid environments) and could use a reference. However, if the authors chose 1 mm hr-1 based on their expertise and knowledge about this catchment, then I think it's more accurate (and in no way worse) to phrase this decision along those lines, e.g.: "We chose this threshold as a reasonable value upon which we expect differences in hydrologic behavior, based on our collective understanding of the Colpach catchment."*

**RL:** Valuable point, we will rephrase this sentence.

*WK: P17, l10. I think it's import to repeat the similarity condition of dQ/dt here, because for a model that has no unique relation between model state and dQ/dt values this method cannot be applied without accounting for this difference.*

**RL:** Please see the discussion above and in section 5.3 in our MS.

*WK: P20, l6. The authors use KGE values in this section and Table 1. I'm not sure to what extent the aggregated value is a useful metric for events that last only a handful of time step. It would be good to at least disaggregate the KGE into its correlation, variability and bias components (e.g. quantify what can be qualitatively estimated from Figure 7) to see if the total KGE scores of the individual models are generated by (roughly) the same types of errors in the simulations.*

**RL:** Good point. We will add the three components of the KGE in the appendix for each model.

*WK: P21, l25. "acceptable" is somewhat subjective because no standard of acceptability has been defined. It might be cleaner to simply report the correlation component of the KGE to quantify to what extent the hydrograph shape is simulated.*

**RL:** Agreed. We will rephrase this term.

*WK: P21, l26. This trial of a direct runoff component seems somewhat ad-hoc to me. I don't think this adds anything to the manuscript and that it will take more space than is available to properly justify this change. I suggest to remove these sentences.*

**RL:** Thank you. We will consider removing this sentence.

*WK: P30, l4-24. These sentences seem as if they would be better placed in the introduction or methodology sections.*

**RL:** We will rephrase some of these sentences. Please see the discussion with reviewer #1 (Daniel Wright).

---

## Author Comment (AC4) · 18 Sep 2020

Reply to Referee #4 Anna E. Sikorska-Senoner:

**Anna E. Sikorska-Senoner** *(AS): Summary and Recommendation: This paper proposed an adaptive modelling as an alternative to a distributed model for representing spatial variability of the catchment and forcing input (precipitation). Such an adaptive modelling should be able to run faster than a distributed model but should provide a similar model performance as its fully distributed version. The manuscript is generally well written and it is easy to follow. The idea of a spatially adaptive model that dynamically adjusts its spatial structure during runtime is indeed very interesting and has a potential for being applied in many (hydrologic) modelling approaches. Yet, I have few major issues that should be addressed before considering this manuscript for a publication in HESS. Thus, I recommend a major revision.*

**Ralf Loritz (RL):** We would like to thank Anna E. Sikorska-Senoner for her comments and the time she invested to review our Manuscript (MS). We hope that after the discussion as well as after we have revisited our MS all open issues she raises can be clarified.

Comments:

*1. AS: The adaptive model (model c) is tested here only on two rainfall events, which I see as the major weakness of this manuscript. As the strength of this approach should lie in the possibility to apply it to a continuous modelling and not to an event-based modelling. Thus, I think it would be important to demonstrate how the model c works on continuous time series. As this is missing in the current manuscript, we still do not know at the end whether it is a good or a bad option to be used.*

**RL:** *Model a* and *model c* simulate close to identical hydrographs at the end of both rainfall events when *model c* represents the Colpach catchment again by a single hillslope model. This is also true for the soil moisture distributions, which we did not show in the current MS. This means that the information about the spatial organization of a past rainfall event have already been dissipated closely after the spatial adaptive *model c* represents the catchment by a single hillslope. In other words, there is no difference between *model a* and *c* after this point and we would learn not much by letting *model c* run continuously until the next rainfall event.

Furthermore, as rainfall event II is characterized by one of the longest rainfall durations in summer and event I by the highest intensity and third highest spatial variability we see no reason to expect that the spatial adaptive model will fail at other summer rainfall-runoff events. We think that it is not the length of the simulation that matters here but the fraction of the visited state space (or in other words if your training data set is representative). The latter means that we do not assume that the catchment and the model which represents it will function differently at the other untested events. This is underpinned by the fact that also the 42 model elements in the distributed *model b* do not drift apart. The latter reflects the highest complexity *model c* could reach.

However, we agree that we did not well justify the selection of the two events. Following your comment, we will hence plot the soil moisture distribution of *model a* and *c* for event I and II at the time step when the catchment is again represented by a single hillslope. This will show that there is no difference between the *spatially aggregated model a* and *model c* already shortly after the rainfall stopped. Furthermore, will we improve our discussion regarding the choice of our two rainfall-runoff events. Again, we would like to thank AS for her comment.

**2. AS**: *The performance metrics of the calibrated (tuned) models should be provided so that the model ability to predict rainfall events could be assessed*

**RL:** The reference model is the only model which was manual tuned to match the seasonal water balance of the Colpach. This procedure is described in detail in *Loritz et al. (2017)* and in the current MS in section 3.1. The KGE value of the reference model is reported in table 1. We will furthermore add the three components of the KGE as discussed with Wouter Knoben to the appendix.

**3. AS**: *A model set-up between the model a and b could be very didactical, i.e. having a structure as the model b but using the precipitation input as the model a (the same for each grid cell)*

**RL:** The only difference between *model a* and *b* is the precipitation input. Running *model b* with the input of *model a* would mean to produce the same hydrograph as *model a* 42 times.

**4. AS**: *It is not quite clear how the switch between different model setups (i.e. the number of model run in the model c) affect the setup of initial conditions for next runs, which is important to be considered for continuous model simulations but also for simulations of events. More details should be provided on that.*

**RL:** Please see the discussion above.

**5. AS**: *It would be also very didactical to see the comparison of the precipitation records from the ground station with the precipitation fields obtained from the gridded data. This is never done in the manuscript and no reason for not doing that is given.*

**RL**: Agreed. We will add the precipitation from the ground station to Fig. 2b.

*6. AS*: *The (rather) poor model performance of all tested models' set-ups for two selected events requires some discussion. It appears that none of these model can really capture the dynamics of these two events even if using the distributed model and the distributed rainfall information (with KGE<0.3). Hence, it is even more important to verify the model performance (pkt. 2). An addition of other metrics that focus entirely on the flood event such as peak or time to peak could be here very informative. Given a rather poor models' performance, it is difficult to justify the need of developing the adaptive model based on the distributed model if the latter does not provide acceptable simulation results.*

**RL:** Respectfully, the focus of this MS is not to minimize residuals between an observed quantity and a model simulation. The main goal of this study is to introduce an approach with the goal to setup a spatially adaptive model and equally important underpin this approach with a physical meaning. Furthermore, would we like to highlight that we a) discuss the model performance and how it could be improved on page 21 line 26 to 28 and b) would like to reiterate that the *reference model*, which is the basis of this study, was tested against a series of different variables (sap flow, discharge, water balance, soil moisture, etc.), at different hydrological years, in an additional sub-basins as well as mainly setup based on field observations. We believe that such an evaluation and model-building process underpins the quality as well as the ability of a model to mimic the hydrological dynamic of a landscape sufficiently and maybe even more than adding another performance metric.

As the model was setup to simulate the seasonal water balance we think that the annual performance is quite "good" and we are not surprised that if we zoom into a single event that we loss performance. Furthermore would we like to highlight that the performance metric, which is important here is the KGE between *model b* and *c*, which is 0.98. To improve the interpretability of our model scenarios we will add a second table with the three components of the KGE to the appendix as discussed with Wouter Knoben. Furthermore, will we clearly state that the goal of this study is not to perform a best as possible streamflow simulation.

*7. AS*: *A fair comparison of all presented models should involve the same metrics, i.e. computation over the same time at a continuous time scale. In this study, different model setups are compared at different scales that makes it difficult to get an overview of their performance.*

**RL:** We compare and discuss the connection between *model b* and *model c* only for the two events as well as for the corresponding summer period. Respectfully, we do not think that the comparison is unfair.

Detailed comments

**1. AS:** *Abstract: 'a mesoscale catchment'; a 20-km$^2$ catchment appears rather small tome than meso-scale.*

**RL:** Meso-scale: 5 to 1000 km$^2$, we refer here to the work of *Zehe et al. (2014)* and *Dooge, (1986)*.

**2. AS:** *Abstract: 'three hydrological models', the model is actually the same but different set-ups are used that span from the averaged model until the distributed model. Please clarify that here.*

**RL:** This depends on your the definition of the term "model". However, I agree and we will use the term model setups here.

**3. AS:** *L. 20-21 p. 3: It is not always possible and justified to switch from a continuous model to an event based model. Hence, continuous modelling is often required in many applications.*

**RL:** Agreed. Could you provide a reference here?

**4. AS:** *L. 19 p. 4: The tested catchment appears rather small to me. How do you define the cut here for a small/meso-scale catchment?*

**RL:** We refer here to the work of *Zehe et al. (2014)* and *Dooge, (1986)* which is around 5 to 250 km$^2$. The definition of organized complexity is that such systems are too complex that we can tread them exclusively in a mechanistic manner but too organized that we can represent them in a purely statistical manner.

**5. AS:** L. 7-9 p. 5: consider restructuring this sentence.

**RL:** Thank you. We will consider rephrasing it.

**6. AS:** *L. 11-13 p. 7: could you add the location of these meteorological stations to the map in fig 1?*

**RL:** The station "Useldange" is too far away to be added to the map. But its location is provided in the corresponding reference. We will add this information.

**7. AS:** *L. 15 p. 7: it should be 'and measures...'*

**RL:** Thank you. Changed.

**8. AS:** *L. 16 p.7: is there any weighting applied here and what kind of?*

**RL:** No weighting applied.

**9. AS:** *L. 28 p. 7: could the locations of radars be also placed in the fig. 1?*

**RL:** No, they are too far away. However, their location is displayed in the reference provided by *Neuper and Ehret, (2019).* We will add this information.

**10. AS:** *L. 12-14 p. 8: Why do you compare these values with literature and not with the ground station records from your catchment? Is there any reason that you are not using the precipitation records from the ground station?*

**RL:** These values represent the climatic averages of the area. We have only data for about 10 to 15 years.

**11. AS:** *Fig. 2: One would expect that the radar values would be compared with the ground station values as a corresponding mean (Fig. 2b). Could you add these values to the figure?*

**RL:** Agreed. Will be added.

**12. AS:** *L. 10-17 p. 9 till 21 p. 10: I am not quite sure if this text is really helpful. After reading these lines, we still do not know how the reference model and other models look like. Maybe you could merge these lines with the sections 3.1-3.3.*

**RL:** We will restructure section 3. Please see the discussion with Daniel Wright.

**13. AS:** *L. 19-20 p. 10: I would say that the main goal is to test or verify whether similar model performance can be achieved with the adaptive model as compared to the model b. However, by the comparison that you did we still do not know the answer to this question as the comparison is done only based on two pre-selected events both having rather a poor model performance. Please comment on that and also state why these events were chosen for the comparison (and not others)?*

**RL:** Please, see the discussion above.

**14. AS:** *L. 21-22 p. 10: Why do you compare the adaptive model with model b using only these two events and not the entire simulation period? In my opinion, the greatest potential of the adaptive modelling lies in continuous modelling and not in the event-based.*

**RL:** Please, see the discussion above.

**15. AS:** *L. 31 p. 10 – l. 1 p. 11: why do you test the model only based on the annual assessment and not on hourly simulations? It is quite surprising because you use the model for assessing the model performance at an event-based scale in the second step, i.e. when comparing different models. I think it is important to report here how the model behaves at an hourly time scale so that one knows what can be expected from the model.*

**RL:** I am not sure if I have understood that comment correctly. But we tested our models by comparing hourly simulations with hourly observations for one hydrological year.

**16. AS:** *L. 3 p. 11: Which metrics were used here for assessing that the model performance agreed well with the dynamics of observed values? Can you give some more details on that?*

**RL:** We use the Spearman rank correlation, the Nash-Sutcliff eff. and the KGE. We refer to the study of *Loritz et al., (2017)*.

**17. AS:** *L. 7-8 p. 11: It is not surprising that the model performs poorly at time series scale if it was evaluated only on an annual basis. Some insights should be given here; why was the model tested at an annual basis if its intention is to predict events?*

**RL:** Respectfully, the goal of this study is not to perform a best possible streamflow simulation with regards to minimize residuals. If this would be the case we would have picked a more data driven approach.

**18. AS:** *L. 3 p. 12: similar to what?*

**RL:** They are the same in all models.

**19. AS:** *L. 12. p. 12: the model analysis should go after the introduction of all models.*

**RL:** Agreed. We will restructure section 3.

**20. AS:** *L. 16-19 p. 12: an additional model between the models a and b would be here very useful, i.e. a model that has a structure as the model b but uses precipitation as in the model a (so it uses the same precipitation for each grid cell). This inclusion could nicely show the added value (or no value) of including a spatial distribution of i) the model and ii) of the precipitation input data.*

**RL:** Please, see the discussion above.

**21. AS:** *L. 8-10 p. 13: why exactly? In my opinion, the strength of this approach lies in the possibility to apply it to a continuous modelling and not to an event-based modelling. Thus, I think it would be nice to demonstrate how the model works on continuous time series in terms of the model performance and computational efforts. As such a test is missing in the current manuscript, we still do not know at the end whether it is a good or a bad option to use the adaptive modelling approach... Based on the two events selected, we cannot say much about the value of the adaptive approach as the model performance remains poor for these events (as seen from the Table 1 and fig. 7). If the full simulation is not possible(could you give more details why exactly?), already a simple test with shorter but continuous time series of few months or weeks could provide some more insights on how this approach is really working.*

**RL:** Please, see the discussion above.

*22. AS: Tab. 1: as the initial idea is to improve the model performance for rainfall events, it appears from the table that the model c and model b have still rather poor model performance for the event I and II. In addition, all models perform poor for these events. Yet, an inclusion of the spatial variability does not improve much the model performance that is still not so good. Thus, it calls a question of the need of such an adaptive inclusion to this spatially distributed model which performance is rather low.... Could you comment on that? A decomposition of KGE into its components would bring more insights on the models' behaviour.*

**RL:** We will add the three components of the KGE to the appendix.

*23. AS: L. 9-10 p. 15: How many grid cells need to have a difference higher than this threshold to use the model c?*

**RL:** One.

*24. AS: L. 30-31 p. 15 fig.3: is the re-arranged model running with the same initial conditions of the original model or how do you decide on these initial conditions if you want to increase or decrease the number of M in the subsequent time intervals particularly if a continuous simulation is performed?*

**RL:** We aggregate their states.

*25. AS: L. 4-6 p. 18: for a fair comparison of different models, you should use the same metrics and the same time periods for evaluation. It is not clear why this is not the case here.*

**RL:** See discussion above.

*26. AS: Fig. 4: could you add simulations with the model a?*

**RL:** If we add all simulations the figure is hard to read. However, we will consider your comment when we revisit your MS.

*27. AS: L 8. P. 20: The reference Knoben et al. (2019) is missing in the literature list.*

**RL:** Thank you we will add this reference.

*28. AS: L. 6-7 p. 21: the performance of KGE below 0 is still rather very poor, which re-quires some further explanations. According to Knoben et al. (2019), simulations can be considered as behavioral if KGE>0.3 (with KGE≈−0.41 for a mean flow benchmark).*

**RL:** See comments above.

*29. AS: Table 1: the performances (KGE) of all models is rather poor for the events here selected (with KGE between -0.41 and 0.29). As already the model b (distributed)cannot simulate the two events in a*

*good way (as also seen from the fig.7), why would you spend time on developing the adaptive model based on the model b instead if improving the model b or testing different models here? Could you comment or justify that?*

**RL:** See comments above.

***30. AS:** Fig. 7: the simulations with the model a and the reference model should also be added here. Moreover, for both events, all models largely underestimate the events. Could you comment on that?*

**RL:** In fig. 7 we focus on the comparison of *model b* and *c*.

***31. AS:** L 10 p. 26 – l. 2 p. 27: do you have any idea where this large underestimation may come from and how it could be improved?*

**RL:** Discussed in the MS (page 21 line 26 to 28).

***32. AS:** Discussion: I missed some recommendations for other works. When and how would such an adaptive modelling be recommended? How one can set up the adaptive process? And why it is really needed to implement such an adaptive modelling?*

**RL:** Thank you for this comment. We will revisit the discussion of the MS in this regards.

---

## Referee Report (RR1)

The authors have responded well to the four reviews. I am particularly happy with the response to the similarity metrics used to determine when model states are similar and separate model elements can be merged. The authors have updated their choice of similarity metrics and gone to great lengths (e.g. section 4.1, page 23, section 6.3) to impress on the reader that their choice of metrics is catchment and study-dependent, which is a welcome addition. The experiment still relies on just two rainfall events instead of a continuous simulation for a longer period. I understand the authors' justifications for this but I think the paper can be improved further by expanding on this choice a bit more (qualitatively). Details below.

1. Reliance on two rainfall events

The experiment remains constrained to only two rainfall events during summer. The authors have clarified their reasoning for this (there is no benefit from using a spatially distributed model in winter conditions anyway, and the two events are the most extreme conditions available during summer). I find this a reasonable response although I think further testing over longer time scales would still be good to build confidence in this approach. It may be worthwhile to slightly reframe the paper and call this a proof of concept, until a run for a whole year or multiple years is feasible.

I also get the impression that there was a practical element to this choice in the sense that creating an automated version of the spatially adaptive model is "technically difficult" (p13, l10 in the original manuscript) and a "full automation of the proposed adaptive clustering approach [..] is beyond the scope of this study" (p13, l8 in the revised manuscript). I don't think the paper mentions this in so many words, but I get the impression that upscaling and downscaling the number of model elements was a (mostly) manual activity in this experiment.

The cases where this approach would be most useful (long simulations with many catchments) are equally the cases where manually upscaling or downscaling the model setup is least feasible. I think it would be helpful to the community if the authors can provide some context for the decision not to automate this process and describe the practical limitations that make automating this approach difficult. Is there something specific to CATFLOW that makes automation difficult or do the authors foresee similar difficulties for models that are regularly ran across large domains, such as land-surface models? Will there be additional difficulties to overcome, such as issues with routing in larger catchments, where also the location of the rainfall event will start to play a role and a hillslope far away from the outlet cannot be easily merged with one near the outlet on account of their different travel times?

A note on the additional computations needed (binning rainfall field on every time step, binning model states and averaging them in case the number of model elements can be reduced, etc) could be helpful too to determine potential gains in computational efficiency but I would understand that such numbers may currently be difficult to estimate.

A discussion such as this could help the community decide whether this approach is something that can actively be pursued with current models or whether this remains a largely academic exercise until a new generation of models can be developed that are specifically designed to take advantage of spatially adaptive setups. Section 6.2 may be a good place for this.

**Line-by-line comments**

P1, l1. Given the scope of the test cases in this manuscript (only 2 events during 1 year are assessed) and the overall message from the four reviews, one might say that this paper is more about providing a proof of concept for a spatially adaptive model and less about investigating the "role and value if distributed precipitation data for hydrological models". I would suggest that the authors rephrase their title to either emphasize that this paper shows a novel model approach or to be more specific about the limited scope of the investigation of role and value of distributed precip data for modelling.

P3, l22. It's not fully clear to me why moving to the event scale instead of running continuous simulations is one way to achieve dynamical allocation of the model space. Can the authors clarify this by adding the reason(s) why such a shift enables this new type of modelling and/or why this type of modelling is not possible for continuous simulations?

P5, l18. Following up on my earlier comment, I asked whether "> 1 m" should be "< 1 m" because the soils are described as "shallow" in this same sentence. Given the authors response: "No, soils are rather deep in this area and […]" perhaps this description needs to change. This of course doesn't impact the paper's results in any way, but it might avoid some confusion for other readers.

P9, l8. The manuscript mentions four model setups here (reference, a, b and c) but the abstract (p1, l9) mentions only three.

P10, l25. "One goal of … *rather than a result of important structural difference [..] within the Colpach catchment*." Technically, this second part is not tested as such in the paper. The results indicate that with a spatial variable precipitation field modelling results improve but this does not imply that model structure issues do not play a role. It might be good to clarify this sentence.

P21, 27 and further. I still think this addition of a direct runoff component is somewhat ad-hoc and contributes little to the paper. If the main goal is not to accurately match observations in this catchment, this semi-section can be removed to streamline the message in the results section.

P28, l6.It may be more accurate to change "summer season." to "summer season in this catchment."

P28, l22. Should these words be underlined?

P32, l2. Is "constellations" the right word here? Should this be "occurrences" or something similar?

**Editorial**

P3, l22. "one-way" > "one way"

P13, l4.  "The main goal of the model testing of the spatially adaptive …" > "The main goal of testing the spatially adaptive …"?

P14, l7. "are so large, that" > "are so large that"

P14, l11. "elements by" > "elements is by"

P14, l11. "Hydrology" > "hydrology"

P14, l28. "similar" > "similarly"

P15, l18; also l19. "upon we" > "upon which we"

P15, l20. "in humid catchment" > "in this humid catchment"?

P16, l8. "Section 3.3.2" > this needs updating but I'm unsure to what. Section 4.1 maybe?

P17, l3. "models two" > "models to"

P21, l25. "acceptable" > "acceptably"

P23, l29. "is" > "are"

P27, l8. "simulate" > "simulating"

P32, l5. "equifinality" > "as equifinality"

**Misc**

I'll also takes this opportunity to briefly respond to comment #28 by reviewer 4, to clarify a potential misunderstanding. Part of comment #28 states:

"*According to Knoben et al. (2019), simulations can be considered as behavioral if KGE>0.3 (with KGE≈−0.41 for a mean flow benchmark)*".

While we do use KGE = 0.3 in our paper, this is only to illustrate that if arbitrary thresholds are used to determine which models are behavioural, NSE and KGE do not produce consistent results. In fact, later in the paper we try to warn the reader against using such thresholds in an arbitrary manner:

"*Regardless of whether KGE or some other metric is used, the final step in any modelling exercise would be comparing the obtained efficiency score against a certain benchmark that dictates which kind of model performance might be expected (e.g. Seibert et al., 2018) and decide whether the model is truly skillful. These benchmarks should not be specified in an ad hoc manner (e.g. our earlier example where the thresholds are arbitrarily set at NSE=0.5 and KGE=0.3 is decidedly poor practice) but should be based on hydrologically meaningful considerations.*"

I see that the authors also argue against using this threshold in their manuscript and I think is an appropriate response.

---

## Referee Report (RR2)

Dear Editor,

We would like to thank you and the four referees for handling, respectively reviewing, our manuscript (MS). In line with our responses to the reviewers we carefully revised our MS as outlined below:

- We restructured section 3 by splitting it into two sections. In section 3 we now exclusively introduce the model setups and in section 4 we explain the adaptive modeling in detail. We, furthermore, added a table to section 3 where we summarize the properties of the different model setups and added a new subsection where we explain how we test the model setups.

- We carefully revised the discussion of our MS. This means that we now clearly discuss the limitations of our approach regarding the selected test periods and the chosen similarity metric. We, furthermore, added and discuss the proposed literature of the reviewers and finally have improved the connection of our work to the land surface community.

- We added a second variable, namely Q, to group model states.

- We changed Fig. 8 and added the soil moisture distribution after 48 hrs to highlight that different model elements are again in a similar state after 48 hrs of no rainfall. The latter as well as another new figure in the supplement showing the Shannon entropy of the distributed *model b* underpins that the model states of individual hillslopes are not drifting apart in the summer season. This means that there is no reason to assume that the spatially adaptive *model c* would fail if we would run it for the entire summer season in a fully automated manner.

- We updated the KGE values of the reference model for rainfall event I and II in Tab. 2, which were incorrect due to a wrongly selected too long timing period. This does, however, not change the general pattern of the model results.

- Additionally we add:
    o the measured rainfall at the ground station "Roodt" to Fig. 2b
    o a table with the three components of the KGE to the supplement
    o the location of the distrometers and micro rainfall radars to the supplement
    o a new figure to the supplement showing how different binning widths influence the spatial resolution of the adaptive *model c*
    o added information about the numerical schemes used in CATFLOW

- We removed the appendix and added a supplement to further increase the readably and structure of the MS.

Enclosed we added the revised MS with track chances as well as our detailed responses to the four reviewer comments. Again we would like to thank you and the four referees for handling, respectively reviewing, our MS and look forward for your assessment.

Yours sincerely,

Ralf Loritz, on behalf of the Co-Authors

[revised manuscript text omitted]

Reply to Referee #1 Daniel Wright:

**Daniel Wright** *(DW): Summary and Recommendation: The authors present a framework for dynamically adapting the level of spatial detail re-solved within a physics-based rainfall-runoff model depending on the spatial variability in precipitation. I found this the be one of the most interesting manuscripts that I've ever reviewed, and commend the authors on this innovative work. Nonetheless, there are some issues that should be addressed before the manuscript is suitable for publication in HESS, and that could help maximize the impact of the work.*

**Ralf Loritz (RL):** We would like to thank Daniel Wright for his positive comments and the time he invested to review our Manuscript (MS). The revised MS will follow the reviewer's recommendations and include among other things a re-structured section 3 (model introduction) as well as a more extensive discussion about its connection to the land surface modeling community. Furthermore, will we carefully check the references the reviewer recommended and see whether they help us to improve our argumentation.

Major comments:

**1. DW**: *I believe the discussion could be strengthened by deeper consideration of how this approach would "scale up" to larger watersheds or regions. Part of my reason for encouraging this is that the land surface modeling (LSM) community is at least as concerned as the rainfall-runoff community about model computational demands of long-term/ensemble simulations, and are seeking ways of representing fine-scale (e.g. hillslope and below) over continental-to-global domains. In fact, land surface modeling was the focus of the well-known Wood et al. (2011) hyperresolution modeling opinion piece. In addition, there has been relevant progress in LSM development that the authors should cite. I will mention these below. But in terms of scaling up, the key aspects seem to be acknowledgement that heterogeneity of model parameters will increase with modeled area, while the rainfall spatial coverage will, on average, decrease.*

**RL:** Thank you for raising this point. We mentioned in our MS in the method section page 14 line 28 to 31: "*This entails, however, also that if we extend our research area to a catchment that is divided, for instance, into two geological settings that function hydrologically differently (regarding their filter properties) we would always need to run at least two structural different models where each of these models represents one of two geological settings.*"

We agree with the reviewer that this section is rather short and will provide a more extensive discussion in a revisited MS (see also the following points). We will also carefully read the proposed references by Woods et al. (2011).

**2. DW**: *I believe the discussion could also be strengthened by some discussion of how well this approach might fit with specific types of spatial discretizations. It fits quite naturally with hillslope-based models.*

*The fit is less clear with gridded or TIN-based models-or at least with high-resolution gridded models in which individual model grids must "communicate" with each other to transmit water via overland or subsurface flow to channels.*

**RL:** This is an important point. Indeed our approach is limited to hydrological models that are based on a division of the landscape into partly independent spatial units (similar to the work of *Chaney et al. 2016*). However, at least in theory, there is no limit on how complex the interaction between these independent sub-units are as long as there is redundancy/similarity when different model elements "*communicate"* with each other. However, the question at what point of model complexity we would still save computational times by reducing redundancy depends on a series of factors (e.g. model, resolution, no. of processes and state variables). We will discuss this in a revisited MS.

**DW**: *It seems that the computational advantages of the approach might be limited in that case. In addition, models such as GSSHA in which overbank river flow can return to the land surface would have some limits here too. These issues are worth discussing because such models constitute important current directions in physics-based model development.*

**RL:** From our perspective, it makes much sense to divide a landscape into different building blocks such as hillslopes, sub-basins, etc. (*e.g. Zehe et al., 2014*). This is the case as current physically-based models are still constrained to small areas if they are set up on an appropriate grid size. We see hence no way around dividing a landscape into some kind of independent sub-units and either run models in parallel or/and group similar model elements (dynamically or time-invariant) if we want to work on larger scales. That said, we also believe that it would be rather difficult to implement a spatial adaptive modeling approach in a current model like Delft2d or GSSHA.

We wrote in our MS (Pg. 16 line 10): "*While we use CATFLOW as a model here, the proposed approach is not restricted to this model and can be used in any hydrological model that distributes a catchment into independent spatial units.*". To underpin this point we will discuss the limitation of our approach in more detail (please also see the discussion with the second referee and the third). Again we thank DW for this valuable comment.

**3. DW**: *While there may be other relevant LSM developments, the one that I am aware of is Hydroblocks (Chaney et al. 2016). While I recommend reading that paper, the basic approach is similar to this manuscript's in that spatial units are grouped into hydrologically similar clusters to reduce the computational demand.*

**RL:** Thank you very much for pointing us to the study of *Chaney et al. (2016)*. We will examine it carefully.

**DW**: *The difference is that in Hydroblocks, these clusters are not dynamically reassigned according to time-varying characteristics (unless the developers have recently added that capability). So in fact, your*

*approach appears to be superior in some respects. Specifically, within Hydroblocks, since there is no dynamic reassignment, you can never have a cluster that extends beyond the spatial extent of a single precipitation grid cell, which means that their approach loses computational efficiency with higher-resolution precipitation datasets. Your approach thus seems to hold more promise in terms of flexibility to advances in precipitation inputs.*

**RL:** Interesting comment. We would like to highlight that we only showed that our approach is theoretical and practically feasible. It remains an open question if we could actually save more computational times in comparison to HydroBlocks or similar time-invariant approaches. The question is open for discussion as our approach is also more complicated. We will discuss this in a revised discussion section.

*4. DW: More clear description of what each model does and does not do is needed in Section 3. Specifically, I found it confusing the way that the models are briefly introduced at the beginning of the section, and then discussed further in various subsections. I also find it strange that you have text that is not assigned to specific subsections. It isn't clear why section 3.2.1 is needed...convention is that you don't include subsections unless you have at least 2 or 3 (i.e. 3.2.2, 3.2.3). This section structuring needs rethinking. [...] Also, a table that compares the key features and differences of all the models could be effective. I think one think that would really help is to not use "model a", "model b", etc. but some brief descriptive names that actually help the reader understand and recall the differences. More important, I really couldn't figure out from the descriptions what the differences between some models were. I also don't understand the motivation for using a different rainfall dataset for the reference model and model a; this seems unnecessary.*

**RL:** We stated in our MS on page 11 line 26-29: "*We added model a to test if the performance difference between the reference model and our distributed model b is merely a result of quantitative differences between the different precipitation products measured either by a single ground station or by a weather radar.*"

However, we understand the comment of Daniel Wright and we will restructure section 3 entirely and remove the short introduction of the different models. We will also follow your advice and add a table with the key features of each model.

*5. DW: Zhu et al. (2018) and Peleg et al. (2017) both highlight how distributed rainfall structure is really important in determining flood frequency across a range of scales. Though I normally refrain from suggesting that authors cite my own work, in this case it seems appropriate to highlight these studies, since they do show that for extreme events, rainfall space-time structure is extremely important in determining hydrologic response even at very small scales (see Peleg et al. in particular), and that this importance varies with rainfall magnitude and basin size.*

**RL:** We have carefully read both publications and they fit nicely into our revisited discussion. Thank you for pointing us towards these two references.

*DW: Along with this, I disagree with the statement on pg. 27: "it seems that catchment size might not be the best indicator to decide if" a distributed model is needed. It probably is the best single indicator, but is still insufficient. I draw a somewhat different conclusion from your work: that a distributed approach is always needed to reap the full benefit of spatially distributed rainfall (at least in locations in which convective rainfall can occur), and that provides motivation for continued developments such as this into ways of handling this need in computationally-efficient ways.*

**RL:** What we wanted to convey here is that it is the combination of the drainage area and the average size of a typical rainstorm, which is important and not the drainage area alone. For instance, if you wanted to predict the runoff formation in the Colpach catchment in the winter season a spatially aggregated model driven by a single precipitation time series might be sufficient as our results show. This means also that you could invest your limited time and improve for instance the groundwater representation in your model instead of setting up a distributed model. However, if you wanted to make predictions in the summer months our results highlight that you need some sort of a distributed model to be able to capture the spatial variability of the rainfall. This means that only because the Colpach is 20 km$^2$ we cannot decide if we need a spatially distributed model as the catchment size does not explain how variable its meteorological forcing is. *Nicotina et al. (2008)* argued along these lines and stated that the "*total residence time of a water parcel is often controlled by the travel time within hillslopes, we find that when typical hillslope size is smaller than the characteristic size of rainfall structures (say, a correlation length of rainfall intensity), the rainfall pattern effectively samples all possible residence times and the response of the catchment does not depend on the specific rainfall pattern.*" and the second referee pointed us towards the study of *Ogden and Julien (1993)*. The second reviewer also nicely summarized their key finding: "*only for rainfall with durations shorter than the concentration time of a catchment does the spatial distribution of the rainfall matters, for longer rainfall events only the temporal distribution matters.*". Following these two studies and our own results we would argue that our first research question in our MS: "How important are spatial patterns of precipitation for the runoff generation at the catchment scale?" can only be answered if we combine information about the catchment size (e.g. average hillslope length, concentration time) with information about the meteorological forcing (e.g. intensity, correlation length, velocity). In a revisited MS we will rephrase this paragraph and explain in more detail what we meant by this statement.

*DW: Likewise, I disagree with the statement on pg. 30 line 18-19: compressing rainfall into a single time series isn't so important as the ability to only use as much computational power as is truly needed to solve the problem at hand.*

**RL:** Again an interesting point you raise here. In our specific setting, compression of precipitation and saving computational power are the same. By compressing the precipitation field to a single time series we also compress our model, minimize redundant calculations, which again means that we save

computational power. So, we would argue that we first need to understand (test) how far we can compress our rainfall field without losing predictive performance before we can save computational times in a meaningful manner. The data-based / machine learning community most likely would disagree ☺

**6. DW:** *Some discussion of implications for calibration would be interesting. Is it necessary to calibrate using a fully distributed model? This would limit the usefulness of this approach in some respects such as automated calibration procedures.*

**RL:** Typically, one run of the reference model (a single CATLOW hillslope) for a simulation period of one year and hourly printout times takes about 2 - 3 hrs. Assuming that you run your code on a workstation with 32 cores you can run about 400 model setups in 24 hrs. As structurally similar areas are represented by the same model in our approach, testing different model parameters sets should be feasible even in larger areas if the structural properties are not too variable/complex. We will discuss this in a revisited MS.

**7. DW:** *There are a number of minor grammatical issues that nonetheless cause some distraction from the overall high quality of the manuscript. I will point out some of these below, but it could be worthwhile to have a native English speaker perform a careful proofreading.*

**RL:** We will carefully proofread the MS once more and would like to highlight that there will be another professional language check by Copernicus if the MS is accepted for publication.

References:

Nicótina, L., Alessi Celegon, E., Rinaldo, A. and Marani, M.: On the impact of rainfall patterns on the hydrologic response, Water Resour. Res., 44(12), 1–14, doi:10.1029/2007WR006654, 2008.

Ogden, F. L. and Julien, P. Y.: Runoff sensitivity to temporal and spatial rainfall variability at runoff plane and small basin scales, Water Resour. Res., 29(8), 2589–2597, doi:10.1029/93WR00924, 1993.

Zehe, E., Ehret, U., Pfister, L., Blume, T., Schröder, B., Westhoff, M., Jackisch, C., Schymanski, S. J., Weiler, M., Schulz, K., Allroggen, N., Tronicke, J., van Schaik, L., Dietrich, P., Scherer, U., Eccard, J., Wulfmeyer, V. and Kleidon, A.: HESS Opinions: From response units to functional units: a thermodynamic reinterpretation of the HRU concept to link spatial organization and functioning of intermediate scale catchments, Hydrol. Earth Syst. Sci., 18(11), 4635–4655, doi:10.5194/hess-18-4635-2014, 2014.

Reply to Anonymous Referee #2:

**Anonymous Referee #2** *(AR2): Summary and Recommendation: The manuscript introduces an adaptive spatial clustering of hydrologic response units (HRU) to cope with the dynamics of the intermittent rainfall by keeping the model as parameter parsimonious (=model states) as possible in terms of reduction of similar-reacting HRUs. The manuscript is well-written and I enjoyed reading it. The introduced clustering is innovative from and fits into the scope of the journal. I have a few moderate and a number of minor comments, which are stated below. My overall recommendation would be a moderate revision to give the authors enough time to solve the open issues. Since I can only choose between minor and major revision, major revision it is.*

**Ralf Loritz (RL):** We would like to thank the second referee for the time and the effort she/he put into his review. The points she/he raises are relevant and addressing them will help to improve our manuscript. We hope that after this discussion (as well as after we revised our manuscript) all issues she/he raises can be clarified.

Moderate comments:

*1. AR2*: *The manuscript is about the reduction of the spatial model resolution based on the variety of precipitation as input signal. I'm wondering if there is not an adaption of the temporal resolution required as well since scales in space and time are not independent of each other (see Melsen et al. (2015) and references therein)? Maybe it's not an issue for the small catchment studied here …*

**RL:** Important comment. The results of the study of *Zhu et al. (2018;* recommended by the first reviewer) highlight that the timing of the precipitation is more important in smaller catchments while it is the spatial pattern in larger catchments. We will discuss this in a revisited MS and carefully read the study of *Melsen et al. (2015).*

*AR2*: *… but for larger catchments with a small hydrologic variability the numeric stability can be questioned due to the large spatial discretization and the high temporal resolution (e.g. in terms of the Courant-Friedrichs-Lewy condition, Courant et al., 1928). The authors should proof this condition for their model setup and discuss possible issues in the manuscript. An alternative would be to reduce the temporal resolution as well, which would lead to an additional reduction of parameters/computational costs.*

**RL:** CATFLOW uses an adaptive time-stepping, which means that time steps can be reduced down to seconds depending on the numerical solver. In the presented study the Darcy Richards equation is solved implicitly while the surface runoff is solved explicitly (for more details see also *Zehe et al., 2001*). As the horizontal grid resolution of the CATFLOW hillslope (reference model) is below 1 m, the vertical below 10 cm and time steps are small we have no issue to fulfill the Courant criteria in our model. Nevertheless, you mention an important point here and the fulfillment of physical and numerical constraints were the main motivations of our former "representative hillslope" study (*Loritz et al. 2017*).

CATFLOW hillslopes are typically interconnected by a river network and runoff is routed downstream with a diffusion wave approach (explicitly solved) assuming a prismatic river cross-section and roughness that changes with changing Strahler order. However, the combination of a river network with the raster layout of our adaptive model is not straightforward (although not impossible, for instance by linking the centroid of a raster cell to the closest node of the river network). To make things not more complicated as necessary we decided to use a lag function in this study. This lag function is not solved numerically but shifts the simulated hydrographs in time by a constant velocity. Again we have no issue with the Courant criteria here. The latter is different if we would have used an adaptive mesh approach where the numerical grid is changed during runtime. Here we carefully need to check the courant criteria when we increase the size of the grids. We will discuss this in a revisited MS.

*2. AR2*: *The authors have selected two events to show the ability of adaptive clustering. The choice of both events seems to be very arbitrary. From Fig. 4 it seems that the resulting runoff peaks are not representative for runoff mechanisms of the catchment. As far as I understand from P13 l8-10 the clustering is carried out manually and not automatically so far, which is the reason why the authors decided for two small events covering only a few time steps. However, I disagree with the hypothesis that "a test on a longer timescale...would provide only little more scientific inside" (P13 l9-10), which is also not proven by the authors.*

**RL:** Respectfully, we do not agree with the assessment of the reviewer regarding the selection of our rainfall events. We chose event I as it has the highest intensity and third-highest spatial variability in the chosen period. We chose event II because we wanted to test our adaptive model at a rainfall event with a longer duration and lower intensity). From examining the rainfall-runoff events in summer we believe that both events represent the runoff generation in summer well as long as subsurface storm flow is dominant. We agree with the reviewer that our statement is a bit misleading and we will explain better what we mean here. Please see also the discussion with the third and fourth reviewers.

*AR2*: *I rather expect that the reduction of model parameters due to the adaptive spatial resolution is reduced significantly for long-lasting rainfall events causing a direct runoff response over several days as e.g. in Nov 2014, Jan 2014-Mar2014 and Aug 2014.*

**RL:** You are correct. The needed spatial model resolution in winter is much lower compared to the chosen summer rainfall-runoff events. This is indicated by the fact that the distributed *model b* and the *reference model* perform almost identical with respect to simulate the observed discharge in the winter season. We would argue that is is difficult to justify the use of a spatially distributed model over a spatially aggregated model if they perform similarly as long as the focus is on an integral response of a system.

*AR2*: *Another point that can be questioned is snow, which does not cause runoff immediately, but when snow melt begins. How will this be affected/can be incorporated by the adaptive clustering? The impact*

*of more complex events than those analysed in the current study has at least to be discussed sufficiently in the manuscript, although an analysis of more events is encouraged to represent the effect of the adaptive clustering on the variety of runoff responses.*

**RL:** Interesting point. Snow is rare in the Colpach catchment which is fortunate as CATFLOW has no internal snow routine. However, let's assume we would have used a model with a snow routine in an area where snow is a dominant control on the runoff generation. In this specific scenario, we would indeed have to adapt our definition of similarity between the model states. In other words, instead of using the only slope of the simulated hydrograph alone to define similarity, we would also need to check the snow cover before we would group models as functional similar based on their state. Two similar hillslopes would then have the "same" snow cover (given a threshold) as well as the same slope of the hydrograph. We very much like the idea of testing the approach in an area where snow is an important factor. However, for now, we will discuss the limits of choosing a single variable to group model states in a revisited MS.

*3. AR2: The model states are identified by the slope of the resulting runoff curve. However, the slope can be more or less identical for one time step independent of the current runoff situation, e.g. if runoff is reduced in one tile from 25mm to 20mm and in another tile from 10mm to 5mm (which could be the case in a stratiform event with a convective cell inside), the resulting slope is the same, right? So the soil moisture and other storage elements is then "averaged" due to the same model state of both tiles, although both tiles are in completely different hydrologic situations. It would be useful if the authors would comment on that issue or, if I understood it not correctly, clarify the part where I got lost.*

**RL:** You are correct. In an earlier version of our spatial adaptive model, we used the absolute discharge to identify similar model states. The issue here was that two models could produce the same discharge at a given time step but one model would simulate a rising hydrograph while the other a declining. We hence decided to take the slope of the hydrograph assuming that the model differences would be small given the size of the Colpach catchment, the focus on the summer season and because we only simulate shallow subsurface stormflow. In the case of a stratiform event with a convective cell inside or if we have snow in a catchment our assumption might be violated. Thank you for raising this issue and along your lines, we will add another criterion to our spatially adaptive model. In a revisited MS only model elements which share a similar $dQ\ dt^{-1}$ (0.05 mm hr$^{-1}$) as well as Q (0.05 mm hr$^{-1}$) will be grouped together.

Specific comments:

*AR2: P4 l5-8 The difference is not clear formulated at this point. It becomes clearer while reading the manuscript, but should be communicated concisely at this point.*

**RL:** We will rephrase this sentence.

*AR2: P7 l27 Where are the disdrometers located? Can they be used to improve the rainfall input for the reference model to achieve a more realistic uniform areal rainfall? If not, could be an increase of rainfall amounts with altitude improve the areal rainfall estimate? The Roodt station is situated in the raster field with the lowest rainfall amounts (Fig 2) and not representable for the catchment. So any correction has to be done to enable a fair comparison between reference model and model a.*

**RL:** When we were setting up the reference model for our proceeding study (*Loritz et al., 2017*) the only rainfall measurement available at that time was the ground station in "Roodt". As we are aware that the comparison between the *reference model* and *model b* (the distributed model) is not entirely fair as they used different rainfall data we added *model a* to the model ensemble. In a restructure section 3 we will clarify this as well as add the location of the distrometers to the appendix A1.

*AR2: Fig 2 Please add rain gauge data to Fig 2b) to enable a comparison of all rainfall inputs.*

**RL:** We will add the rainfall data from "Roodt" to Fig2b.

*AR2: P10 l2, p11 l26 area-weighted -> As I understand the areal mean is estimated by the arithmetic mean of the satellite data. How do weights for different areas affect this estimation? This is not clear for me, please rephrase/add the explanation.*

**RL:** As not all of the 42 raster cells of the distributed rainfall data are entirely within the borders of the Colpach catchment their weight was reduced when we calculated the average precipitation for *model a*. We will rephrase this sentence accordingly.

*AR2: P11 l2 sap flow -> Do the authors mean by sap flow the flow in plants? I can't imagine at this point how the authors applied observations like that in the current study. If so, please describe a bit more detailed, since it is not a conservative measure for model validation and hence of great interest for the community.*

**RL:** By sap flow we indeed mean sap flow in plants. In our proceeding study (*Loritz et al. 2017*) we compared normalized sap flow velocities against normalized transpiration simulations of CATFLOW to evaluate the transpiration simulation. Sap flow measurements where thereby one of the keys for a successful simulation in the Colpach catchment as they helped us to identify the onset of the vegetation (when the trees started to transpire). The comparison is described in detail in *Loritz et al. (2017*; Figure 12). Although we agree that this might be of great interest for the community we would like to avoid discussing this once more to keep the MS as focused as possible.

*AR2: P12 l23 "average distance of each grid cell to the outlet" -> Should it not be the distance along the flow path/flow direction? So it would be possible that the runoff is assumed to stream upwards in some areas of the catchment- Please rephrase or reconsider.*

**RL:** For each grid cell of the precipitation field we calculated the average flow length along the surface topography to the outlet of the catchment using an underlying DEM with a 10 m resolution. We used the averaged flow distances in our lag function. We will explain this in more detail in a revised MS.

*AR2: P12 l30 "3.2.1 to 3.2.3" -> "3.3.1 to 3.3.3"*

**RL:** Following the discussion with the first reviewer Daniel Wright we will restructure section 3 and remove all "subsubsections".

*AR2: P13 l2 "wetness state" Please define this term. It sounds as only soil moisture is included without any additional information, but there is more included, right? If not, why not using the term soil moisture? Section 3.3 and 3.3.1 There are repetitions among the paragraphs, please remove them.*

**RL:** We will remove the term "wetness state" with the term "catchment state" to make clear that we also mean the shallow groundwater table, soil moisture, etc. We will furthermore restructure section 3 and remove the repetitions.

*AR2: P15 l4 & 21 Both thresholds are catchment size-dependent (as the authors state also later).For other applications it would be useful to introduce a catchment size-dependency to derive these thresholds. This is beyond the scope of the study since it demands a multi-catchment analysis, but the authors should add a small sensitivity analysis by e.g. using ΔP >{0.5, 1, 2} mm/hr as thresholds. This is along with a comment I have for the results section later, but I want to state it already here. In the results discussion it is often mentioned, that the number of parameters is reduced between model b and c, there is no figure illustrating it, although I would imagine it would bean impressive plot with y as KGE over x as the summarized number of model parameters per time step (or on average) for one event. Model reference, a, b, c (ΔP>1mm), c(ΔP>0.5mm) and c(ΔP>2mm) would be the points to show in the diagram. I assume model c would represent a break in the curve (KGE not increasing, while number of model parameters do) and the different thresholds would represent the uncertainty of this approach.*

**RL:** Interesting comment. Using a typically physically-based model (CATFLOW) and specifically the setup of our model based on field measurements it is kind of difficult to estimate the number of model parameters in our study. However, we will add a plot with the distributed precipitation binned into different thresholds (0.1, 0.5, 1, 2, 5 mm hr$^{-1}$) to the appendix. Based on this plot we will discuss how the binning will most likely affect our spatial adaptive model. Furthermore, will we discuss that a sensitivity analysis with different thresholds is needed along your line of arguments. Again thank you for this comment.

*AR2: Table 1: As far as I understood the calibration was done only for the reference model, right? Although that seems to be done in a former publication, a brief information about calibration and validation period, objective function and so on is required to interpret the table. For model a, b and c no parameters were changed, so the same parameter set was used throughout the study to enable comparisons? If there was a re-calibration for model c, the reference model and models a and b should be re-calibrated for the events only as well to enable a fair comparison*

**RL:** Exactly the calibration was done in a former publication exclusively for the reference model. All model parameters remain the same. The only differences between the models are the rainfall data which we use to drive the models as well as their spatial resolution. We will add more details about the calibration in a restructured section 3.

*AR2: Fig. 5: I'm a bit confused here. The authors state P=12 for t=2, but from counting it is P=13 – please double-check (also the number of entries in the following text refer ring to t=2). Additionally, for t=4 M=3 results from P=2 and S=1 – from my understanding the maximum of model states is max(M)=2 in this case, please double-check.*

**RL:** You are correct. We will check the figure as well as the corresponding text passages. Thank you for checking the figures so carefully.

*AR2: P27 l4-22 This paragraph provides already a good overview of related references. However, from my understanding the reference of Nicotina et al. (2008) concluded that spatial patterns of rainfall are only important for large catchments (8000km2 in their study) for hourly time steps, the correct estimation of areal rainfall is sufficient for smaller catchments. The authors should review this reference again and check their implementation in the current manuscript.*

**RL:** Thank you for this comment. We were referring to the following section in *Nicotina et al. (2008)*: "*As noted in section 4, this is because the spatial scales of variability of rainfall are very often much larger than the typical hillslope scale. Whenever infiltration excess mechanisms are important, the spatial distribution of areas of intense rainfall may be an important factor in determining the hydrologic response, … "*. In the current MS the use of this reference is indeed misleading and a leftover from an earlier version. We will revisit the corresponding sentence.

*AR2: Also, Ogden and Julien (1993) state that only for rainfall with durations shorter than the concentration time of a catchment the spatial distribution of the rainfall matters, for longer rainfall events only the temporal distribution matters. To highlight the importance of distributed models the authors could also look at Krajewski et al. (1991), Bardossy & Das (2008) or Müller-Thomy et al. (2018)*

**RL:** Thank you very much for pointing us to these publications we will read them carefully and see if they can help us to improve our argumentation.

Reply to Referee #3 Wouter Knoben:

**Wouter Knoben** *(WK): Summary and Recommendation: The authors develop and test a hydrological model that is able to change its spatial complexity in time. In its most simple state, the model represents the Colpach catchment in Luxembourg as a single representative hillslope. In its most complex state, the model would be able to use 42 hillslope elements to simulate the catchment's response to extremely spatially variable rainfall inputs. The model adds hillslope elements based on the spatial complexity of incoming precipitation and removes hillslope elements based on the change of runoff over time. Both processes use a threshold to decide when upscaling or downscaling the model is needed or possible. The authors show that the adaptive model reaches the same KGE scores as a fully distributed model that uses 42 hillslope elements all the time, while the adaptive model needs 16 representative hillslopes at most. This is shown for two short-duration event that occurred during summer.*

*I have read this paper with much interest and found it generally easy to read and understand. As models grow more complex, computation times go up and studies such as this could open up great opportunities to reduce computation costs by avoiding redundancy in model calculations. However, I have some questions about the tests and metric the authors use to show that the adaptive model is as good as the fully distributed one. These are outlined below. I've provided additional requests for clarification in the line-by-line comments in the hopes that these are helpful.*

**Ralf Loritz (RL):** We would like to thank Wouter Knoben for the interesting discussion on our Manuscript (MS). WK raises a couple of important and well-thought comments and we hope that after this discussion as well as after we have revisited our MS all open issues can be clarified.

Comments:

*1. WK*: *My main concern is the choice of using dQ/dt to reduce the number of model elements. Using the change in discharge over time to measure similarity of states can only work if there is a unique relationship between model state and dQ/dt. Given the equifinality in the fluxes-discharge relation that's typically visible in hydrological models (see e.g. Khatami et al., 2020), I think the section that introduces this concept (P16, l17) is not quite clear about why this dQ/dt assumption can be used together with CATFLOW.*

**RL:** Important comment and also the second reviewer had similar concerns. We will hence add Q as a second variable to group and ungroup model states and improve the discussion on how using a single variable to define similarity between model states will always lead to errors in certain scenarios and following that these variables need to be picked carefully.

*WK*: *Reading further, the authors address this concern to some extent in section 4.4 (P23, l18). This section however seems to show that CATFLOW does not exhibit such a unique relationship and the model reduces the number of model elements before the groundwater states reach similarity. This does apparently not affect the quality of the simulations much, because the KGE scores in Table 1 seem to indicate the adaptive model is as good as the fully distributed model for the two testing events.*

**RL:** In the current MS we did not mention that CATFLOW simulates only shallow subsurface stormflow in the entire summer period. This means that we do have a rather unique relationship between our model states and their function. Furthermore, the example in Fig.8 shows two extreme cases where one model receives much more precipitation than the other exactly intending to show the limits of our approach (section 5.3). As discussed in more detail with the fourth reviewer we will show that there is no difference between *model a* and *model c* (also concerning soil moisture in both depths) already shortly after the rainfall stops and when *model c* represents the entire catchment with a single hillslope. Furthermore, by calculating the Shannon entropy of the 42 hydrographs simulated by the spatially distributed *model b* we can see that there is no reason to assume that two models drift apart in the selected time frame. We will disscuss this in a revisted MS.

**WK**: *Fig. 4 shows that both testing events are selected in the middle of summer, when presumably the catchment is in quite a dry state (catchment state is not mentioned when selection of the two events is discussed on P18, l20 to P19, l6).*

**RL:** We mention the catchment states for event I and II on page 18 line 26 to 28 and page 19 line 5 to 6. Furthermore, do we refer to our former study where we showed 38 time series of soil moisture in the Colpach catchment in various depths and locations for the same hydrological year.

**WK**: *The fact that both events are selected during the dry summer could mean that the model can reset itself to mostly empty between the events and as such the long term (seasonal) impacts of not keeping the groundwater states separate cannot be investigated with the current two testing events.*

*Equally, the events concern high flows so the impact of differences in slow ground-water states probably do not register in the dQ/dt values during the falling limb of the hydrograph (and thus the adaptive model simplifies itself).*

*There is the compounding issue that the KGE scores used to calculate the performance of model c are only calculated during the high flow event and that metrics such as KGE are typically not very sensitive to errors in low flows. This means that the parts of the simulation time series where the differences in groundwater states could be seen are both not included in calculation of the KGE score of model c and if they were, the KGE metric might not be able to pick up on any differences.*

**RL:** Again an important point. As already mentioned above we discussed in section 5.3. "*While this finding is surely constrained by the chosen threshold, the picture is nevertheless quite different in deeper soil layers where the diversity of the rainfall forcing leads even after 24 hrs to increasing differences between the "driest" and "wettest" models. A part of the information about the different meteorological forcings between the two models is hence still stored in the model state after 24 hrs and has not yet been dissipated. The importance of those differences likely depends on the dominant runoff generation process. In the present case, they have a minor impact as model …*"

In a revisited MS we will underpin once more that our approach with the current definition of similarity regarding the model states can have significant impacts for long-term simulations however that there is no reason to expect that in our specific case.

**WK**: *Summarizing the above, I'm not sure that the dQ/dt criterion is entirely appropriate to determine when the adaptive model can reduce its complexity, and I'm equally unsure if the current two testing events would be able to show if the dQ/dt criterion is or is not appropriate. The straightforward solution would be to run model c for the year, add these results to Table 1 and briefly investigate for example the relative contributions of different fluxes to the overall water balance and the model's response to a few precipitation events during winter. Given that the adaptive model should be faster than the fully distributed one, this should not be a large computational burden and it will provide a much more complete impression of the capabilities of the adaptive model.*

**RL:** We hope that an improved discussion of the limits of the dQ/dt criterion (or any other criterion) as well as the addition of a second similarity measure (Q) will clarify the issues WK raises. We would also like to stress once more that we focus exclusively on the summer season as the distributed *model b* outperforms the *reference model* only in this period and because the meteorological boundary conditions change between the winter (frontal) and summer seasons (convective; Fig. 4 b). Furthermore, did we chose two rainfall-runoff events instead of the entire period as it allows us to analyzes the events in great detail (Event I: Fig. 5, 6, 7) and as our main focus in this study is on the rainfall-runoff interaction and not on low flow conditions. We selected event I because it has the highest rainfall intensity and third-highest spatial variability (highest Shannon entropy) in the selected period and event II because we wanted to test our spatial adaptive model at a summer rainfall event with longer duration. We believe that both events represent the state space of the runoff formation of the Colpach in summer well and see no reason to assume that the *model c* would fail at another rainfall event. A test of the spatial adaptive model for the entire hydrological year (or even for a longer period), in a different environment, with more variables and different thresholds to group and ungroup the model states and maybe even with a different type of model, is indeed desirable. However, to keep the already quite elaborated MS as focused as possible we will focus on improving the discussion with respect to the limits of our approach, add Q as second criteria to define similar model states, add a new figure to the appendix where we show how the thresholds impact the number of precipitation groups (please see the discussion with reviewer 2) and finally plot the soil moisture of *model a* and *c* at the end of event I and II to highlight that both models are in a similar state also with respect to their soil moisture.

Line-by-line comments

**WK:** *P5, l5. This question seems quite strongly related to the contrasting results in the literature that the authors discuss in the first and second paragraph of the introduction, where they conclude that*

*the impact of using a distributed model and/or distributed forcing data is conditional on the catchment under investigation. This research question seems a bit generic in that light, given that only a single model and catchment are being investigated in this work. As is, question 1 seems more like a formality to me (it must be answered with "yes" before Q2 can be investigated) and the main focus of the manuscript seems to be on Q2. Perhaps the manuscript can gain a bit in focus if only the current research question 2 is specified, and the work done to answer the current Q1 is presented as a prerequisite to address the current question 2. For example, "We test this hypothesis by first showing that the model CATFLOW applied to the 19.4 km2Colpach catchment using a gridded radar-based quantitative rainfall estimate improves in performance when it is distributed in space and driven by distributed rainfall. We then address the following research question: "Can adaptive clustering be used to distribute a bottom-up model in space that it is capable to represent relevant spatial differences in the system state and precipitation forcing at the least sufficient resolution to avoid being highly redundant as a fully distributed model?"*

**RL:** Good idea. We will consider rephrasing this section following your lines.

*WK: P5, l14. Assuming that "> 1 m" refers to soil depth, should it be "< 1 m"?*

**RL:** No, soils are rather deep in this area and vary between 1 to 2.7 m according to several drillings and electrical resistivity tomography (ERT) measurements.

*WK: P7, l9. Which numerical scheme is used by CATFLOW?*

**RL:** Darcy-Richards: implicitly solved by a mass conservative modified Picard iteration scheme (Celia et al. 1990); Surface runoff (1d St. Verdant eq.) explicit Euler forward. We will add this information to the MS.

*WK: P7, l20. If possible without using too much space, it might be helpful to the reader to briefly summarize the main findings of Loritz et al. (2017).*

**RL:** The main findings of this study are summarized on page 10 section 3.1.

*WK: P7, l21. What are the outcomes of this quality control?*

**RL:** Manually quality checked by the Luxembourg, Institute of Science and Technology (LIST; no negative values, etc). We will remove the term "quality checked" as it is not necessary here.

*WK: P7, l28. I'm not quite sure I understand why these distances are given as a range if only a single station is concerned. Does this indicate minimum and maximum distance of the catchment bounds to each radar station.*

**RL:** Exactly. These are the distance to the boundaries of the Attert catchment in which the Colpach is located. We will add this information in a revisited MS.

*WK: P9, l13. I find this sentence a bit hard to follow. Is the part from "apart from..." onwards necessary here? This is already discussed in the introduction.*

**RL:** We will remove this part.

*WK: P10, l21. Why is the model tested during two events instead of over the full year? How were these events selected?*

**RL:** Please see the discussion above and the discussion with the second and fourth reviewer.

*WK: P11, l14. The conclusion that a distributed model is needed to account for runoff driven by convective precipitation would be stronger if the authors can (briefly) list which processes are represented at too coarse a scale in the reference model for it to properly deal with convective precipitation.*

*WK: P11, l14. I believe this sentence would be more complete if it also explicitly mentioned that distributed instead of catchment-averaged precipitation data is needed to properly simulate the result of convective precipitation events.*

**RL:** We wrote on page 11 line 14: *"In other words, this entails that a hydrological model, distributed at a sufficiently high spatial resolution, is required to capture the spatial variability of the precipitation field to satisfactorily simulate the runoff generation of the Colpach"*. We believe that our argumentation is well justified here.

*WK: P11, l27. It would be helpful for the reader to repeat that the only difference between reference model and model a is the choice of precipitation data.*

**RL:** We wrote in the sentence before the sentence you mention: *"Model a is identical to the reference model, however, driven by the area-weighted mean of the spatially resolved precipitation data described in section 2.4 (Fig. 2 b)."*

*WK: P12, l3. Are these variables similar or identical to those used in the reference model?*

**RL:** Identical. We will change the word accordingly.

*WK: P12, l4. To clarify, does this mean that model b is run in a gridded fashion with the catchment divided into 42 grids (matching the precipitation grid)? If not, it would be good to clarify this in the text and mention the number of model elements that the precip field similarity approach gives. Line 18 on this page could benefit from a similar clarification.*

**RL:** Yes, this means that model b is *"divided into 42 grids (matching the precipitation grid)"*. We will consider rephrasing the corresponding sentences.

*WK: P12, l23. Are there some observations that could help support the choice for 1 m/s?*

**RL:** We will add the reference of *Leopold, (1953)*. Fig. 1 in this reference shows an average relation of flow velocities and discharge in rivers. Correspondingly we picked an average value of 1 m s$^{-1}$ (2 to 3 feet per second).

*WK: P14, l29. It might be good to extend this line of reasoning to soil types and vegetation cover, as these are commonly used as model inputs/parameters.*

**RL:** Agreed. We will rephrase the corresponding sentence.

*WK: P15, l7. This sentence is quite general (referring to humid environments) and could use a reference. However, if the authors chose 1 mm hr-1 based on their expertise and knowledge about this catchment, then I think it's more accurate (and in no way worse) to phrase this decision along those lines, e.g.: "We chose this threshold as a reasonable value upon which we expect differences in hydrologic behavior, based on our collective understanding of the Colpach catchment."*

**RL:** Valuable point, we will rephrase this sentence.

*WK: P17, l10. I think it's import to repeat the similarity condition of dQ/dt here, because for a model that has no unique relation between model state and dQ/dt values this method cannot be applied without accounting for this difference.*

**RL:** Please see the discussion above and in section 5.3 in our MS.

*WK: P20, l6. The authors use KGE values in this section and Table 1. I'm not sure to what extent the aggregated value is a useful metric for events that last only a handful of time step. It would be good to at least disaggregate the KGE into its correlation, variability and bias components (e.g. quantify what can be qualitatively estimated from Figure 7) to see if the total KGE scores of the individual models are generated by (roughly) the same types of errors in the simulations.*

**RL:** Good point. We will add the three components of the KGE in the appendix for each model.

*WK: P21, l25. "acceptable" is somewhat subjective because no standard of acceptability has been defined. It might be cleaner to simply report the correlation component of the KGE to quantify to what extent the hydrograph shape is simulated.*

**RL:** Agreed. We will rephrase this term.

*WK: P21, l26. This trial of a direct runoff component seems somewhat ad-hoc to me. I don't think this adds anything to the manuscript and that it will take more space than is available to properly justify this change. I suggest to remove these sentences.*

**RL:** Thank you. We will consider removing this sentence.

*WK: P30, l4-24. These sentences seem as if they would be better placed in the introduction or methodology sections.*

**RL:** We will rephrase some of these sentences. Please see the discussion with reviewer #1 (Daniel Wright).

Reply to Referee #4 Anna E. Sikorska-Senoner:

**Anna E. Sikorska-Senoner** *(AS): Summary and Recommendation: This paper proposed an adaptive modelling as an alternative to a distributed model for representing spatial variability of the catchment and forcing input (precipitation). Such an adaptive modelling should be able to run faster than a distributed model but should provide a similar model performance as its fully distributed version. The manuscript is generally well written and it is easy to follow. The idea of a spatially adaptive model that dynamically adjusts its spatial structure during runtime is indeed very interesting and has a potential for being applied in many (hydrologic) modelling approaches. Yet, I have few major issues that should be addressed before considering this manuscript for a publication in HESS. Thus, I recommend a major revision.*

**Ralf Loritz (RL):** We would like to thank Anna E. Sikorska-Senoner for her comments and the time she invested to review our Manuscript (MS). We hope that after the discussion as well as after we have revisited our MS all open issues she raises can be clarified.

Comments:

*1. AS*: *The adaptive model (model c) is tested here only on two rainfall events, which I see as the major weakness of this manuscript. As the strength of this approach should lie in the possibility to apply it to a continuous modelling and not to an event-based modelling. Thus, I think it would be important to demonstrate how the model c works on continuous time series. As this is missing in the current manuscript, we still do not know at the end whether it is a good or a bad option to be used.*

**RL:** *Model a* and *model c* simulate close to identical hydrographs at the end of both rainfall events when *model c* represents the Colpach catchment again by a single hillslope model. This is also true for the soil moisture distributions, which we did not show in the current MS. This means that the information about the spatial organization of a past rainfall event have already been dissipated closely after the spatial adaptive *model c* represents the catchment by a single hillslope. In other words, there is no difference between *model a* and *c* after this point and we would learn not much by letting *model c* run continuously until the next rainfall event.

Furthermore, as rainfall event II is characterized by one of the longest rainfall durations in summer and event I by the highest intensity and third highest spatial variability we see no reason to expect that the spatial adaptive model will fail at other summer rainfall-runoff events. We think that it is not the length of the simulation that matters here but the fraction of the visited state space (or in other words if your training data set is representative). The latter means that we do not assume that the catchment and the model which represents it will function differently at the other untested events. This is underpinned by the fact that also the 42 model elements in the distributed *model b* do not drift apart. The latter reflects the highest complexity *model c* could reach.

However, we agree that we did not well justify the selection of the two events. Following your comment, we will hence plot the soil moisture distribution of *model a* and *c* for event I and II at the time step when

the catchment is again represented by a single hillslope. This will show that there is no difference between the *spatially aggregated model a* and *model c* already shortly after the rainfall stopped. Furthermore, will we improve our discussion regarding the choice of our two rainfall-runoff events. Again, we would like to thank AS for her comment.

**2. AS**: *The performance metrics of the calibrated (tuned) models should be provided so that the model ability to predict rainfall events could be assessed*

**RL:** The reference model is the only model which was manual tuned to match the seasonal water balance of the Colpach. This procedure is described in detail in *Loritz et al. (2017)* and in the current MS in section 3.1. The KGE value of the reference model is reported in table 1. We will furthermore add the three components of the KGE as discussed with Wouter Knoben to the appendix.

**3. AS**: *A model set-up between the model a and b could be very didactical, i.e. having a structure as the model b but using the precipitation input as the model a (the same for each grid cell)*

**RL:** The only difference between *model a* and *b* is the precipitation input. Running *model b* with the input of *model a* would mean to produce the same hydrograph as *model a* 42 times.

**4. AS**: *It is not quite clear how the switch between different model setups (i.e. the number of model run in the model c) affect the setup of initial conditions for next runs, which is important to be considered for continuous model simulations but also for simulations of events. More details should be provided on that.*

**RL:** Please see the discussion above.

**5. AS**: *It would be also very didactical to see the comparison of the precipitation records from the ground station with the precipitation fields obtained from the gridded data. This is never done in the manuscript and no reason for not doing that is given.*

**RL**: Agreed. We will add the precipitation from the ground station to Fig. 2b.

**6. AS**: *The (rather) poor model performance of all tested models' set-ups for two selected events requires some discussion. It appears that none of these model can really capture the dynamics of these two events even if using the distributed model and the distributed rainfall information (with KGE<0.3). Hence, it is even more important to verify the model performance (pkt. 2). An addition of other metrics that focus entirely on the flood event such as peak or time to peak could be here very informative. Given a rather*

*poor models' performance, it is difficult to justify the need of developing the adaptive model based on the distributed model if the latter does not provide acceptable simulation results.*

**RL:** Respectfully, the focus of this MS is not to minimize residuals between an observed quantity and a model simulation. The main goal of this study is to introduce an approach with the goal to setup a spatially adaptive model and equally important underpin this approach with a physical meaning. Furthermore, would we like to highlight that we a) discuss the model performance and how it could be improved on page 21 line 26 to 28 and b) would like to reiterate that the *reference model*, which is the basis of this study, was tested against a series of different variables (sap flow, discharge, water balance, soil moisture, etc.), at different hydrological years, in an additional sub-basins as well as mainly setup based on field observations. We believe that such an evaluation and model-building process underpins the quality as well as the ability of a model to mimic the hydrological dynamic of a landscape sufficiently and maybe even more than adding another performance metric.

As the model was setup to simulate the seasonal water balance we think that the annual performance is quite "good" and we are not surprised that if we zoom into a single event that we loss performance. Furthermore would we like to highlight that the performance metric, which is important here is the KGE between *model b* and *c*, which is 0.98. To improve the interpretability of our model scenarios we will add a second table with the three components of the KGE to the appendix as discussed with Wouter Knoben. Furthermore, will we clearly state that the goal of this study is not to perform a best as possible streamflow simulation.

*7. AS*: *A fair comparison of all presented models should involve the same metrics, i.e. computation over the same time at a continuous time scale. In this study, different model setups are compared at different scales that makes it difficult to get an overview of their performance.*

**RL:** We compare and discuss the connection between *model b* and *model c* only for the two events as well as for the corresponding summer period. Respectfully, we do not think that the comparison is unfair.

Detailed comments

*1. AS: Abstract: 'a mesoscale catchment'; a 20-km² catchment appears rather small tome than meso-scale.*

**RL:** Meso-scale: 5 to 1000 km², we refer here to the work of *Zehe et al. (2014)* and *Dooge, (1986)*.

*2. AS: Abstract: 'three hydrological models', the model is actually the same but different set-ups are used that span from the averaged model until the distributed model. Please clarify that here.*

**RL:** This depends on your the definition of the term "model". However, I agree and we will use the term model setups here.

**3. AS:** *L. 20-21 p. 3: It is not always possible and justified to switch from a continuous model to an event based model. Hence, continuous modelling is often required in many applications.*

**RL:** Agreed. Could you provide a reference here?

**4. AS:** *L. 19 p. 4: The tested catchment appears rather small to me. How do you define the cut here for a small/meso-scale catchment?*

**RL:** We refer here to the work of *Zehe et al. (2014)* and *Dooge, (1986)* which is around 5 to 250 km$^2$. The definition of organized complexity is that such systems are too complex that we can tread them exclusively in a mechanistic manner but too organized that we can represent them in a purely statistical manner.

**5. AS:** L. 7-9 p. 5: consider restructuring this sentence.

**RL:** Thank you. We will consider rephrasing it.

**6. AS:** *L. 11-13 p. 7: could you add the location of these meteorological stations to the map in fig 1?*

**RL:** The station "Useldange" is too far away to be added to the map. But its location is provided in the corresponding reference. We will add this information.

**7. AS:** *L. 15 p. 7: it should be 'and measures...'*

**RL:** Thank you. Changed.

**8. AS:** *L. 16 p.7: is there any weighting applied here and what kind of?*

**RL:** No weighting applied.

**9. AS:** *L. 28 p. 7: could the locations of radars be also placed in the fig. 1?*

**RL:** No, they are too far away. However, their location is displayed in the reference provided by *Neuper and Ehret, (2019).* We will add this information.

**10. AS:** *L. 12-14 p. 8: Why do you compare these values with literature and not with the ground station records from your catchment? Is there any reason that you are not using the precipitation records from the ground station?*

**RL:** These values represent the climatic averages of the area. We have only data for about 10 to 15 years.

**11. AS:** *Fig. 2: One would expect that the radar values would be compared with the ground station values as a corresponding mean (Fig. 2b). Could you add these values to the figure?*

**RL:** Agreed. Will be added.

***12. AS:*** *L. 10-17 p. 9 till 21 p. 10: I am not quite sure if this text is really helpful. After reading these lines, we still do not know how the reference model and other models look like. Maybe you could merge these lines with the sections 3.1-3.3.*

**RL:** We will restructure section 3. Please see the discussion with Daniel Wright.

***13. AS:*** *L. 19-20 p. 10: I would say that the main goal is to test or verify whether similar model performance can be achieved with the adaptive model as compared to the model b. However, by the comparison that you did we still do not know the answer to this question as the comparison is done only based on two pre-selected events both having rather a poor model performance. Please comment on that and also state why these events were chosen for the comparison (and not others)?*

**RL:** Please, see the discussion above.

***14. AS:*** *L. 21-22 p. 10: Why do you compare the adaptive model with model b using only these two events and not the entire simulation period? In my opinion, the greatest potential of the adaptive modelling lies in continuous modelling and not in the event-based.*

**RL:** Please, see the discussion above.

***15. AS:*** *L. 31 p. 10 – l. 1 p. 11: why do you test the model only based on the annual assessment and not on hourly simulations? It is quite surprising because you use the model for assessing the model performance at an event-based scale in the second step, i.e. when comparing different models. I think it is important to report here how the model behaves at an hourly time scale so that one knows what can be expected from the model.*

**RL:** I am not sure if I have understood that comment correctly. But we tested our models by comparing hourly simulations with hourly observations for one hydrological year.

***16. AS:*** *L. 3 p. 11: Which metrics were used here for assessing that the model performance agreed well with the dynamics of observed values? Can you give some more details on that?*

**RL:** We use the Spearman rank correlation, the Nash-Sutcliff eff. and the KGE. We refer to the study of *Loritz et al., (2017).*

***17. AS:*** *L. 7-8 p. 11: It is not surprising that the model performs poorly at time series scale if it was evaluated only on an annual basis. Some insights should be given here; why was the model tested at an annual basis if its intention is to predict events?*

**RL:** Respectfully, the goal of this study is not to perform a best possible streamflow simulation with regards to minimize residuals. If this would be the case we would have picked a more data driven approach.

***18. AS:*** *L. 3 p. 12: similar to what?*

**RL:** They are the same in all models.

**19. AS:** *L. 12. p. 12: the model analysis should go after the introduction of all models.*

**RL:** Agreed. We will restructure section 3.

**20. AS:** *L. 16-19 p. 12: an additional model between the models a and b would be here very useful, i.e. a model that has a structure as the model b but uses precipitation as in the model a (so it uses the same precipitation for each grid cell). This inclusion could nicely show the added value (or no value) of including a spatial distribution of i) the model and ii) of the precipitation input data.*

**RL:** Please, see the discussion above.

**21. AS:** *L. 8-10 p. 13: why exactly? In my opinion, the strength of this approach lies in the possibility to apply it to a continuous modelling and not to an event-based modelling. Thus, I think it would be nice to demonstrate how the model works on continuous time series in terms of the model performance and computational efforts. As such a test is missing in the current manuscript, we still do not know at the end whether it is a good or a bad option to use the adaptive modelling approach... Based on the two events selected, we cannot say much about the value of the adaptive approach as the model performance remains poor for these events (as seen from the Table 1 and fig. 7). If the full simulation is not possible(could you give more details why exactly?), already a simple test with shorter but continuous time series of few months or weeks could provide some more insights on how this approach is really working.*

**RL:** Please, see the discussion above.

**22. AS:** *Tab. 1: as the initial idea is to improve the model performance for rainfall events, it appears from the table that the model c and model b have still rather poor model performance for the event I and II. In addition, all models perform poor for these events. Yet, an inclusion of the spatial variability does not improve much the model performance that is still not so good. Thus, it calls a question of the need of such an adaptive inclusion to this spatially distributed model which performance is rather low.... Could you comment on that? A decomposition of KGE into its components would bring more insights on the models' behaviour.*

**RL:** We will add the three components of the KGE to the appendix.

**23. AS:** *L. 9-10 p. 15: How many grid cells need to have a difference higher than this threshold to use the model c?*

**RL:** One.

**24. AS:** *L. 30-31 p. 15 fig.3: is the re-arranged model running with the same initial conditions of the original model or how do you decide on these initial conditions if you want to increase or decrease the number of M in the subsequent time intervals particularly if a continuous simulation is performed?*

**RL:** We aggregate their states.

**25. AS:** *L. 4-6 p. 18: for a fair comparison of different models, you should use the same metrics and the same time periods for evaluation. It is not clear why this is not the case here.*

**RL:** See discussion above.

**26. AS:** *Fig. 4: could you add simulations with the model a?*

**RL:** If we add all simulations the figure is hard to read. However, we will consider your comment when we revisit your MS.

**27. AS:** *L 8. P. 20: The reference Knoben et al. (2019) is missing in the literature list.*

**RL:** Thank you we will add this reference.

**28. AS:** *L. 6-7 p. 21: the performance of KGE below 0 is still rather very poor, which re-quires some further explanations. According to Knoben et al. (2019), simulations can be considered as behavioral if KGE>0.3 (with KGE≈−0.41 for a mean flow benchmark).*

**RL:** See comments above.

**29. AS:** *Table 1: the performances (KGE) of all models is rather poor for the events here selected (with KGE between -0.41 and 0.29). As already the model b (distributed)cannot simulate the two events in a good way (as also seen from the fig.7), why would you spend time on developing the adaptive model based on the model b instead if improving the model b or testing different models here? Could you comment or justify that?*

**RL:** See comments above.

**30. AS:** *Fig. 7: the simulations with the model a and the reference model should also be added here. Moreover, for both events, all models largely underestimate the events. Could you comment on that?*

**RL:** In fig. 7 we focus on the comparison of *model b* and *c*.

**31. AS:** *L 10 p. 26 – l. 2 p. 27: do you have any idea where this large underestimation may come from and how it could be improved?*

**RL:** Discussed in the MS (page 21 line 26 to 28).

**32. AS:** *Discussion: I missed some recommendations for other works. When and how would such an adaptive modelling be recommended? How one can set up the adaptive process? And why it is really needed to implement such an adaptive modelling?*

**RL:** Thank you for this comment. We will revisit the discussion of the MS in this regards.

---

## Author Response (AR2)

Dear Editor,

We would like to thank you and the four referees once more for reviewing our manuscript (MS). In accordance to the recommendations of the four referees we have revisited our MS.

- We corrected all line by line comments and carefully read the MS once more.
- We clearly emphasis (two times in the MS) that a full automation of the proposed adaptive modeling is beyond the scope of this study and mention in which models our approach could be implemented (P16 L24-26).
- We remove the term "wetness state" throughout the MS.
- We finally respectfully do not agree with Wouter Knoben and think that the title of our MS suits our study well.
  - We start our MS discussing that the literature provides little general guidance when it comes to making decisions regarding the choice of the rainfall distribution when setting up a hydrological model.
  - We discuss, from a physical point of view, why this might be the case and that the relevance of the distribution of rainfall is changing in time.
  - We further develop an approach to show how we could potentially overcome this shortcoming and show that this approach seems to work at least at two rainfall runoff events.

Enclosed you can find the MS with track changes. Again we would like to thank you and the four referees for the valuable comments which helped us to improve the MS significantly.

Yours sincerely,

Ralf Loritz, on behalf of the Co-Authors

[revised manuscript text omitted]